# Skill dependencies uncover nested human capital

Moh Hosseinioun [1,2], Frank Neffke [3], Letian Zhang[4] & Hyejin Youn [1,2,5,6] ✉

Modern economies require increasingly diverse and specialized skills, many of which depend on the acquisition of other skills first. Here we analyse US survey data to reveal a nested structure within skill portfolios, where the direction of dependency is inferred from asymmetrical conditional probabilities—occupations require one skill conditional on another. This directional nature suggests that advanced, specific skills and knowledge are often built upon broader, fundamental ones. We examine 70 million job transitions to show that human capital development and career progression follow this structured pathway in which skills more aligned with the nested structure command higher wage premiums, require longer education and are less likely to be automated. These disparities are evident across genders and racial/ethnic groups, explaining long-term wage penalties. Finally, we find that this nested structure has become even more pronounced over the past two decades, indicating increased barriers to upward job mobility.

Modern economies demand a broad spectrum of specialized skills and knowledge[1–4]. As these economies grow more complex, the assessment of human capital has evolved from traditional metrics of educational attainment—such as years of schooling or degrees—to a detailed analysis of the specific attributes workers need, including social, cognitive and technical skills[5–15]. In addition, recent research indicates that the value of these skills often emerges from their combinations and interactions, especially when they are synergistic[16–18]. This insight has led to a new way of conceptualizing human capital, viewed as a network of interconnected skills rather than simply a collection of individual abilities[3,16,18–26].

Building on this framework, we explore an important yet often overlooked aspect of how skills relate to one another: their dependencies. Just as mastering calculus requires a prior understanding of algebra and geometry, education and career paths are both cumulative and sequential, with each step building upon the previous one. This dependency structure adds depth to the complementary nature of skills. Workers acquire and develop skills not only because they complement each other but also because they are learned in a specific sequence through educational and professional experiences[27,28]. These dependencies are fundamental to the structure of human capital, shaping career paths in ways that go beyond the simple combination of individual abilities.

Such a perspective leads us to our central research question: what do these skill interdependencies look like, and what broader implications do they hold? One might naturally expect a hierarchically nested structure where specific roles branch off from a central foundation, as in the way specialization typically progresses from a few broader, general contexts to more specific areas. That said, we also expect that not every skill will fit neatly within such a nested structure; some may depend on others, while some may stand alone. This variation suggests that certain skill dependencies, especially those requiring extensive education or training, may be more deeply nested than others within the overall hierarchy. If this is the case, we ask: how might these varying structural properties provide new insights into economic structures, including differential wage premiums and persistent wage disparities observed across different demographic groups?

To explore these questions, we analyse skill portfolios using occupational surveys and resumes to construct the hierarchical skill dependencies that make up human capital[29]. Our findings reveal that these dependencies not only align with familiar dichotomies in job categories—such as blue collar versus white collar; low skill versus high skill and physical roles versus cognitive roles—but also enable us to systematically parameterize these categories using consistent criteria. In this way, we raise deeper inquiry into the origins of these discrete

[1]Kellogg School of Management, Northwestern University, Evanston, IL, USA. [2]Northwestern Institute on Complex Systems, Evanston, IL, USA. [3]Complexity Science Hub, Vienna, Austria. [4]Harvard Business School, Harvard University, Cambridge, MA, USA. [5]Graduate School of Business, Seoul National University, Seoul, South Korea. [6]Santa Fe Institute, Santa Fe, NM, USA. ✉e-mail: h.youn@snu.ac.kr

categories that are often taken for granted in human capital. Furthermore, by incorporating the hierarchical structure of skill dependencies, we integrate one of the core concepts of traditional human capital theory—human capital specificity—into the network-based complexity approach to understanding workforce capabilities and development.

We begin our analysis by differentiating between specific skills, which are required by only a few specific occupations, and general skills, which are applicable across a wide range of occupations (Fig. 1). Since the inception of human capital theory, the distinction between them has been one of its hallmarks, explaining why market economies typically underinvest in general skills[30], why acquiring specific skills creates hold-up problems[31] and why workers often face earning losses when they are displaced from their jobs[32]. In this study, the distinction matters for another reason: general skills serve as a foundational layer of an individual's human capital, upon which more specific skills build, thereby illustrating their nested dependencies within human capital.

Next, we identify and quantify these dependencies between pairs of skills by calculating the conditional probabilities that a skill is required for an occupation, given that another skill is also required for the same occupation (Fig. 2)[33]. Consistent with our common understanding of educational and occupational learning, the skill dependencies manifested in job requirements indeed follow a hierarchically nested structure such that specialized expertise is embedded within broader, general skill layers[27]. Within this framework, we further distinguish between skills that align with this overall nested structure and those that deviate from it. Our analysis indicates that skills that align with this overall structure are associated with longer education, higher wage premiums and a reduced risk of displacement by automation (Fig. 3)[9,34,35]. This dependency structure provides an alternative explanation of why cognitive and technical skills—primarily those deeply embedded within our nested dependencies—along with managerial and social skills (classified here as general skills) tend to command higher returns in the labour market.

Lastly, we examine how the nestedness of skill requirements in jobs recapitulates the course of individual workers' careers[36]. Our longitudinal analysis using three datasets—median occupational ages, synthetic birth cohorts of individuals, and millions of job transitions in resumes—reinforces our initial findings (Fig. 4). First, wage premiums are strongly tied to high proficiency in specialized skills that individuals increasingly acquire and apply as they advance in their careers. But, here is the catch: advancing these high-wage specialized skills nonetheless relies heavily on a strong foundation in general, prerequisite skills (Fig. 5). Specializations that lack these nested dependencies face disadvantages over time, echoing the cumulative and sequential nature of human capital (Supplementary Fig. 31).

By identifying, quantifying and classifying the nested structure of human capital, our analysis provides a comprehensive and dynamic view of workforce skills to offer new insights into career trajectories, wage dynamics and persistent earnings disparities. The nested architecture, for example, provides a structural perspective on long-standing occupational disparities across racial, ethnic and gender lines (Fig. 6). As detailed data on skills, knowledge and tasks become more available[3,19,24], this approach serves as a useful tool for understanding labour market shifts. With the emergence of new skills and the obsolescence of older ones with new technologies[37–39], it becomes even more valuable for systematically assessing the evolving skill landscape and its implications for career development and socio-economic inequalities. Notably, we find that these nested dependencies have grown more pronounced in the past decade, potentially raising new barriers to upward mobility (Fig. 7).

## Results
### Skill generality
The distinction between general and specialized skills is widely acknowledged but rarely empirically quantified in human capital theory[40–49].

In this study, we quantify, analyse and classify the generality of skills on the basis of their breadth of application across occupations by analysing publicly available survey data from the US Bureau of Labor Statistics (BLS). These surveys provide detailed observations on job requirements for nearly a thousand occupations, including the importance and required level of each skill, knowledge or ability necessary for workers to perform their tasks. Each is numerically rated: skill's importance is rated on a scale from 1 to 5 to indicate 'not important' to extremely important, while the required level ranges from 0 to 7, indicating the low to high level of expertise needed for a given occupation. For instance, while speaking is essential for both lawyers and paralegals, lawyers who argue in court require a higher level of proficiency than paralegals, who need only an average level[29]. We chose skill level as our primary indicator of skill demand—measuring how many occupations require each skill at varying levels of proficiency—because advanced skills typically develop through specialized training and education rather than being immediately applied, and thus, it better captures the sequential nature of skill acquisition and development. Nevertheless, these two measures are empirically correlated[50], and our results remain robust even when considering skill importance.

Figure 1a–c shows the average skill level demand, from which we infer skill generality by their distribution profiles. Some distributions are skewed (blue on the left panel), while others are more centred (red on the right panel). Based on these shapes, we group skills into three categories—specific, intermediate and general—and arrange them along a specificity gradient, represented by the arrow at the top. The insets offer examples of these categories to aid our understanding. Specific skills (for example, dynamic flexibility and programming) are characterized as skewed demand distributions that peak near zero with a long tail. This skewed distribution shape indicates that most jobs require little or no proficiency in these skills, with only a few specialized roles requiring higher levels at the distribution tail. Meanwhile, general skills (for example, English language and oral expression) are needed at elevated levels (3–4) across a wide range of jobs owing to their broad applicability.

These demand distributions lay the groundwork for empirical definitions of skill generality, independent of their broader socio-economic contexts. General skills are those required by the majority of jobs at proficiency level 3 or higher, while specific skills are needed at this level in only a limited number of specialized roles, as most jobs need almost no proficiency. In line with this approach, we introduce two additional measures of skill generality—median skill levels across occupations and later-defined network-based metrics. All three approaches demonstrate strong consistency. Median skill levels group skills into three categories: general skills (median level 3.34), intermediate skills (2.37) and specific skills (0.87). In the following section, we introduce network-based measures such as local reaching centrality and nestedness contributions for generality and dependencies, respectively[51,52].

We streamline our analysis by consistently colour-coding skills according to their generality: general (red), intermediate (grey) and specific (blue). Supplementary Table 1 provides the full list of skills—31 general, 43 intermediate and 46 specific, accompanied by various grouping methods. Our findings remain robust across various group sizes, clustering methods and unit choices (Supplementary Section 1).

### Skill dependencies
The uneven distributions of skill demand that we observe create the conditions for a second-order structure beyond individual job requirements: a nested skill–occupation hierarchy. In this structure, jobs requiring specific skills are often subsets of those that demand general skills, indicating that specific skills tend to complement general ones rather than stand alone, while general skills remain largely independent. These asymmetric co-occurrences suggest that the demand for specific skills often depends on the presence of general skills, but not

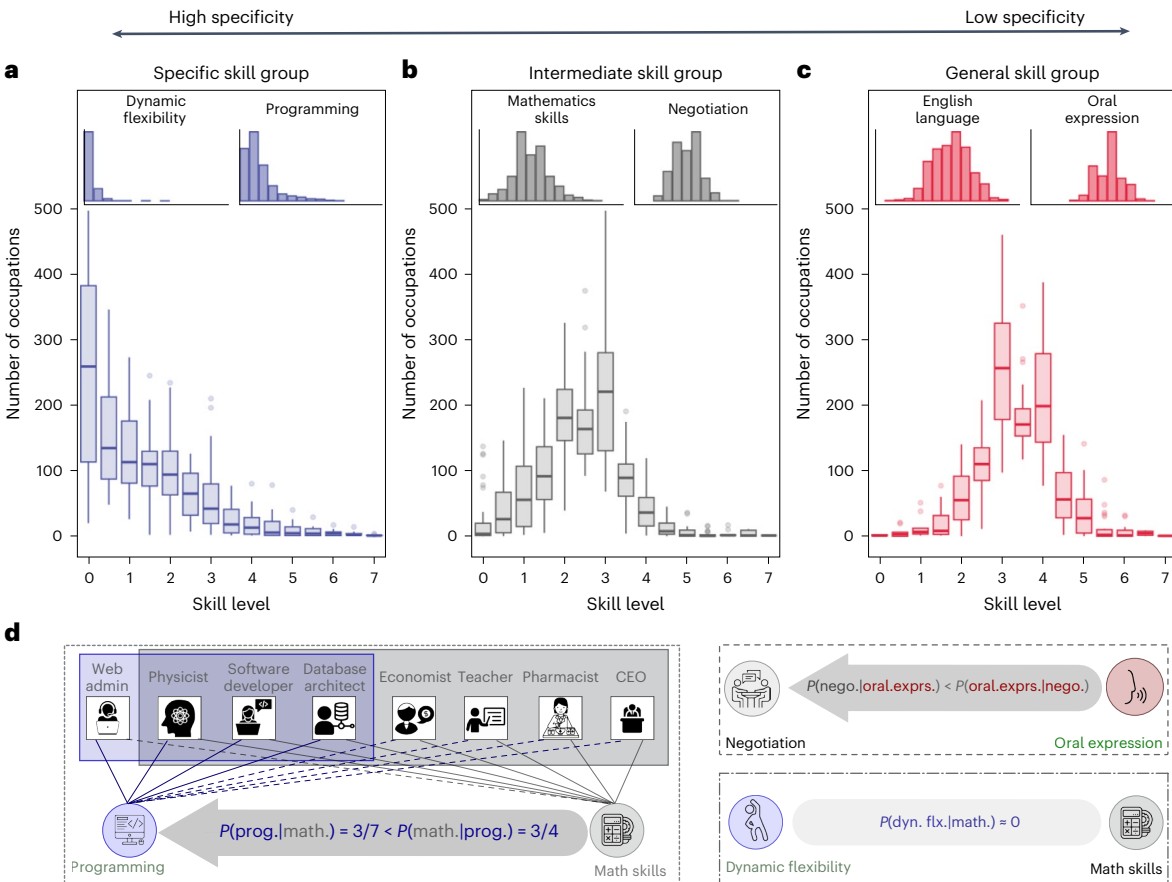

**Fig. 1 | Skill level demand distributions and dependencies. a–c**, The number of occupations requiring given skill levels within each of the three skill groups (Supplementary Section 1): specific (**a**), intermediate (**b**) and general (**c**). The box plots display the data range, with the boxes representing the interquartile range, the central line indicating the median and the whiskers extending to the minimum and maximum data points within the interquartile range. Skills are categorized on the basis of the characteristic shapes of their level distribution across occupations, exemplified by the insets. These categories are labelled as general (31 skills), intermediate (43 skills) and specific (46 skills), with an arrow at the top indicating increasing specificity from right to left. The skewed distribution shapes (blue on the left) peak at zero; that is, most occupations require little or no proficiency in these skills, and only a few require advanced levels. As we move general skills (red on the right), the distribution shifts towards the higher levels, indicating that a wide range of jobs require high proficiency in those skills. **d**, A schematic illustrating our inference method for dependency between skill pairs using the asymmetric conditional probability in job requirements—one skill being required given that another is. For example, if math skill is more likely needed given the presence of programming in occupations (compared with the reverse), $p(\text{skill}_{math}|\text{skill}_{program}) \gg p(\text{skill}_{program}|\text{skill}_{math})$, we infer a directional dependency: math → programming, weighted by the level of asymmetry. Similarly, oral expression → negotiation, but math ↛ dynamic flexibility, as rare and independent events are filtered out (Methods).

necessarily the reverse. This nested skill–occupation hierarchy, manifested in job co-requirements, reveals a hierarchy of skill dependencies that shape career trajectories.

Most jobs require a proficiency level of at least 2 in general skills covering a broad range of roles, including entry-level positions. As a result, workers typically begin their careers in roles that rely on broadly applicable abilities before transitioning directly into more specialized positions. In these transitions, certain skills—most often general ones—serve as prerequisites for more specialized roles. For example, a software engineer may need advanced programming skills (specific) in addition to written comprehension and problem-solving and communication skills (general), whereas a customer service representative who relies on communication and problem-solving skills (general) does not necessarily need programming skills (specific). These skill hierarchies are evident in job co-requirements.

Therefore, we calculate conditional probabilities between different skills in occupation requirements: how often occupations that require one skill also demand another. As previously discussed, the relationship is not necessarily reciprocal. If one skill is a prerequisite for another—similar to how foundational courses precede advanced ones in educational curricula—it is more likely that an occupation will require the prerequisite given that the advanced skill is needed than vice versa. This asymmetric conditional probability suggests a directional dependency between the two. We operationalize our methods by comparing $p(\text{skill}_A|\text{skill}_B)$, the conditional probability of requiring skill$_A$ given the presence of another skill$_B$, with the reverse condition, $p(\text{skill}_B|\text{skill}_A)$. Only when $p(\text{skill}_A|\text{skill}_B) \gg p(\text{skill}_B|\text{skill}_A)$ do we assign a hierarchical direction from skill$_A$ to skill$_B$ as a substantive conditional direction to indicate that application or acquisition of skill $B$ is more likely to be contingent on skill $A$ (ref. 33).

Figure 1d illustrates our inference method with a few examples. For instance, the conditional probability of requiring math skills (whose distribution peaks at 2.5), given the need for programming skills (which skews towards zero, indicating that only few jobs require programming at this level), is greater than the reverse, that is, $p(\text{skill}_{math}|\text{skill}_{prog}) \gg p(\text{skill}_{prog}|\text{skill}_{math})$. Accordingly, we assign a direction math → programming. Note that not all specific skills necessarily depend on general ones. In some cases, such as the pair between dynamic flexibility and math skills in Fig. 1d, skills may appear independently and yet still co-occur by random chance. We apply a $z$-score threshold to remove the random noise and retain only meaningful skill dependencies in Fig. 2 (Methods and Supplementary Section 3).

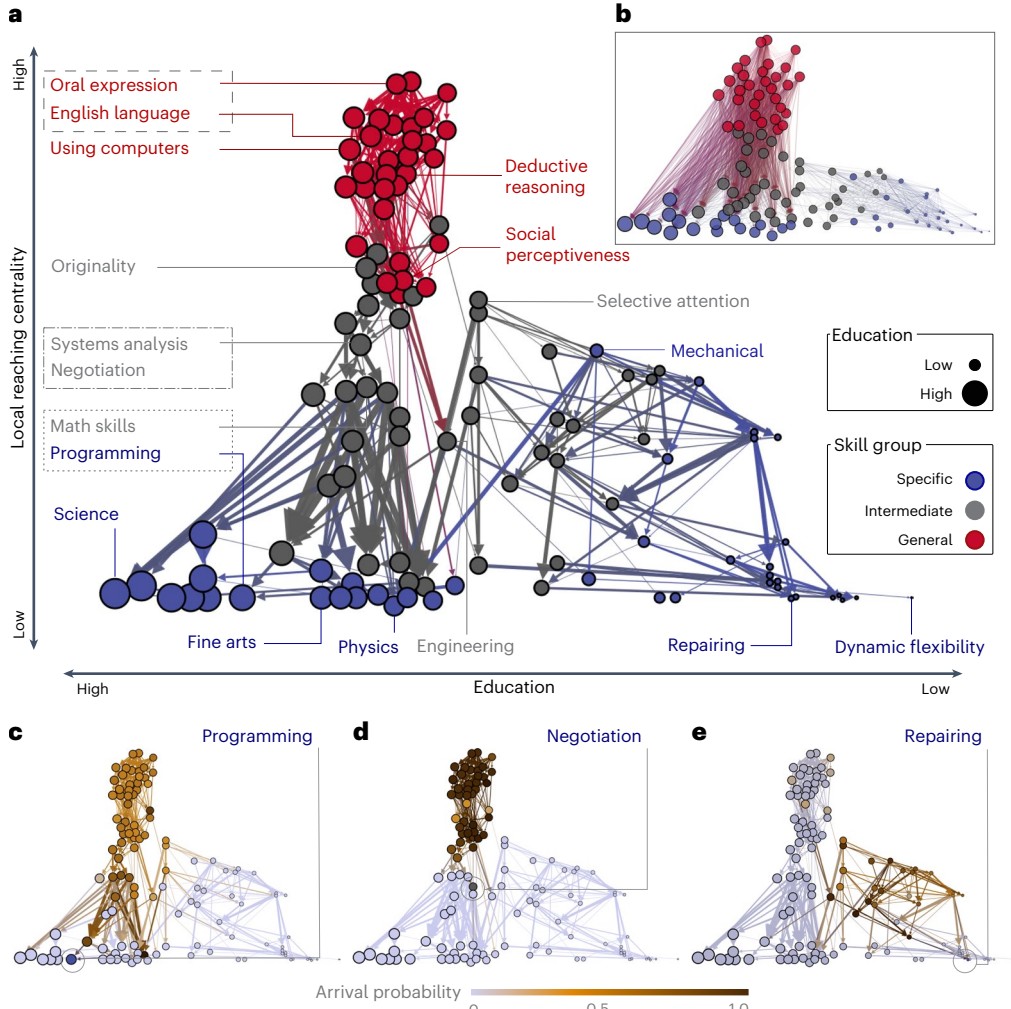

**Fig. 2 | Skill dependency hierarchy. a,b**, The hierarchy is constructed from aggregating weighted directed dependencies for all skill pairs in its backbone connections (**a**) and its full connections (**b**). Node sizes are proportional to education levels and coloured according to the skill generality groups, and embedded in its educational attainment ($x$ axis) and local reaching centrality ($y$ axis)[51]. **c–e**, Reachability (that is, arrival probability) from each skill to programming (**c**), negotiation (**d**) and repairing (**e**) (highlighted)[58]. Dark hues indicate a higher likelihood of arriving at the focal skill (Methods). Contrary to the well-nested programming and negotiation, repairing does not predominantly rely on general skills, indicating its unnested nature.

Figure 2a,b show the resulting dependency network in its backbone and the full connections, respectively (see Supplementary Figs. 15 and 16 for labels). Nodes are coloured by generality group as in Fig. 1 and positioned on the basis of educational requirements ($x$ axis) and the local reaching centrality ($y$ axis), which indicates the number of other skills reachable from the focal skill[51] (Methods). Reaching centrality in this network quantifies the volume of dependent nodes (skills) connected to the focal skill and, thus, can indicate its hierarchy order (skill generality). Indeed, we find reaching centrality strongly correlates with the skill's median level measure ($\rho \approx 0.71$).

Overall, the hierarchical structure in Fig. 2a aligns with our common understanding of specialization. For instance, to develop programming skills (blue node at the bottom), one must first have a general knowledge of math and systems analysis (grey in the middle), which themselves rely on deductive/inductive reasoning (red at the top). Similarly, negotiation skills (grey in the middle) are contingent on systems analysis (grey in the middle) and oral expression (red at the top). These relations constitute the nested dependency chains running from top to bottom in the hierarchy direction in Fig. 2. In the following sections, we will provide a more technical definition of the nested structure, originally developed in ecology[52–54], and explore the implications of these nested dependency chains for wage premiums, career trajectories and skill entrapment.

It is possible that the inferred directionality does not fully account for the underlying microprocesses and mechanisms driving skill acquisition. These dependencies may arise not only from the natural progression of individuals' learning, but also from skill requirements imposed by firms as employees advance in job seniority. Separating these factors would require further microlevel analyses or field studies. However, our additional analysis suggests that job seniority alone does not explain our findings. To test this, we excluded skills and jobs typically associated with seniority, such as social and management skills, as well as occupations with management titles, and found that our results remained robust even with these exclusions (Supplementary Section 9.1). Therefore, the observed nested structure in career trajectories is probably driven by broader acquisition dependencies. This reasoning is especially so in today's complex economy. With its diverse range of specialized skill demands, individuals would not pursue advanced skills—often requiring non-zero effort—unless those skills were necessary. Thus, the inferred directionality from the nested structure reflects acquisition dependencies that shape career development.

Our detailed analysis further supports the manifestation of acquisition dependency in individual career progression. First, we examine the career paths of registered nurses who transitioned to nurse practitioners and find that skill and wage differences in their resume data

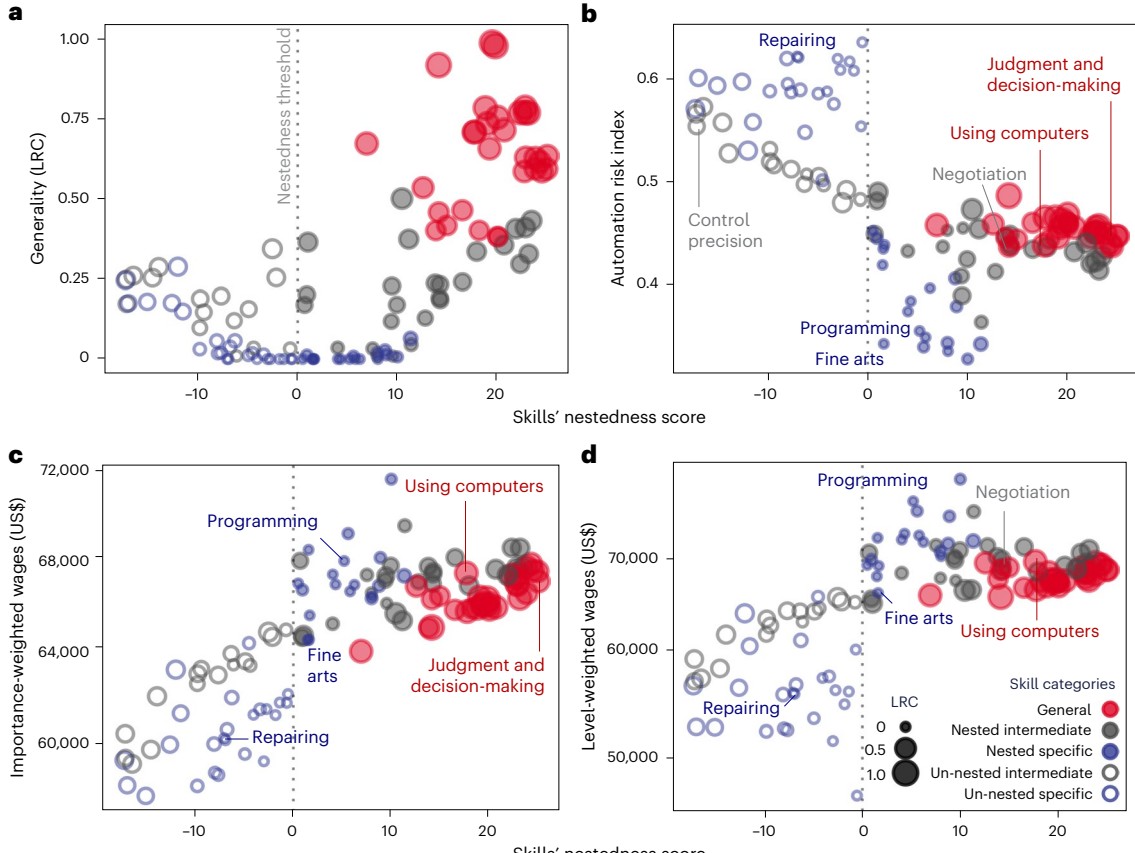

**Fig. 3 | Skill nestedness contributions score $c_s$. a–d,** A skill's nestedness score with its generality (**a**), risk of automation (**b**), importance (**c**) and level values (**d**). Skill nestedness contributions are measured following ref. 52. Generality is measured as local reaching centrality (LRC), which indicates the number of nodes reachable from the focal node through outgoing edges (i.e. in the direction of dependency) in the network of Fig. 2. Automation risk index and value for each skill are calculated following refs. 35,73. We define specific skills with ($c_s > 0$) nested and those with ($c_s < 0$) un-nested.

align with nested dependency chains in the human capital structure (Supplementary Sections 3.4). Furthermore, our analysis of the career trajectories of Hispanic immigrants shows that limited proficiency in certain general skills (for example, English language) limits the development of dependent specialized skills, resulting in demographic skill entrapment (Supplementary Section 3.5). Lastly, we analyse sequential datasets, including occupational ages, survey participants' ages and job sequences in resumes, to further support our findings in the next section.

## Skills' structural alignment with nested architecture

We now quantify the degree to which the overarching skill–occupation structure is nested and how individual skills align with this nested architecture. We first use nestedness ($N$), originally developed in ecology, to quantify to what extent specialists preferentially engage with generalists, which we translate into interactions between specific and general skills[52–57]. Basically, $N$ measures how often specific skills are demanded in occupations that also require general counterparts more than by random chance. There are a number of different definitions of $N$, such as the overlap index ($N_c$), checkerboard score, temperature and nestedness metric based on overlap and decreasing fill (NODF), and we test them to ensure the robustness of our analysis in Supplementary Section 2.

We then assess each skill's alignment with the observed nestedness by its contribution score ($c_s$). The contribution score ($c_s$) for each skill $s$ is calculated as $(N - \langle N_s^* \rangle)/\sigma_{N_s^*}$ (ref. 52), where $\langle N_s^* \rangle$ and $\sigma_{N_s^*}$ are the average and standard deviation of nestedness across an ensemble of random counterparts, respectively. For each skill $s$, we generated 5,000

counterparts ($\{N'\}$) in which its dependencies are randomized (that is, skills are randomly distributed across occupations as if there were no conditional probability structure) while its generality (demand distribution shape) remains constant (Methods).

Therefore, the contribution score ($c_s$) serves to differentiate skills that align or misalign with the observed nested dependency structure. A positive $c_s$ indicates that the dependencies of skill $s$ are aligned with and, thus, positively reinforce the overall nested hierarchy. Conversely, a negative $c_s$ suggests that the skill's dependencies are misaligned, diminishing the nested structure ($N$). For example, skills such as negotiation, programming and fine arts are embedded in strongly interdependent branches of the hierarchy in Fig. 2 (represented by blue and grey nodes on the left), which explains their positive contribution scores ($c_s > 0$) in Fig. 3. Meanwhile, skills such as mechanical, repairing and dynamic flexibility appear in loosely dependent branches (blue nodes on the right), and thus their dependency structure diminishes the overall nestedness ($c_s < 0$). In this way, $c_s$ quantifies how much a skill's dependencies reinforce or diminish the overall hierarchical structure.

Figure 3 and Supplementary Fig. 10 show how the structural attribute $c_s$ correlates with key socio-economic attributes such as generality, education, wages and even automation risk. First, general and foundational skills strongly align with the hierarchical structure of human capital in Fig. 3a. Second, the greater a skill's alignment with this nested structure (that is, the higher the $c_s$), the higher the required education level and expected wages in Fig. 3c,d and Supplementary Fig. 10. To illustrate the implications for career development and reskilling, we present skills' arrival probabilities in Fig. 2c,d (ref. 58). For repairing

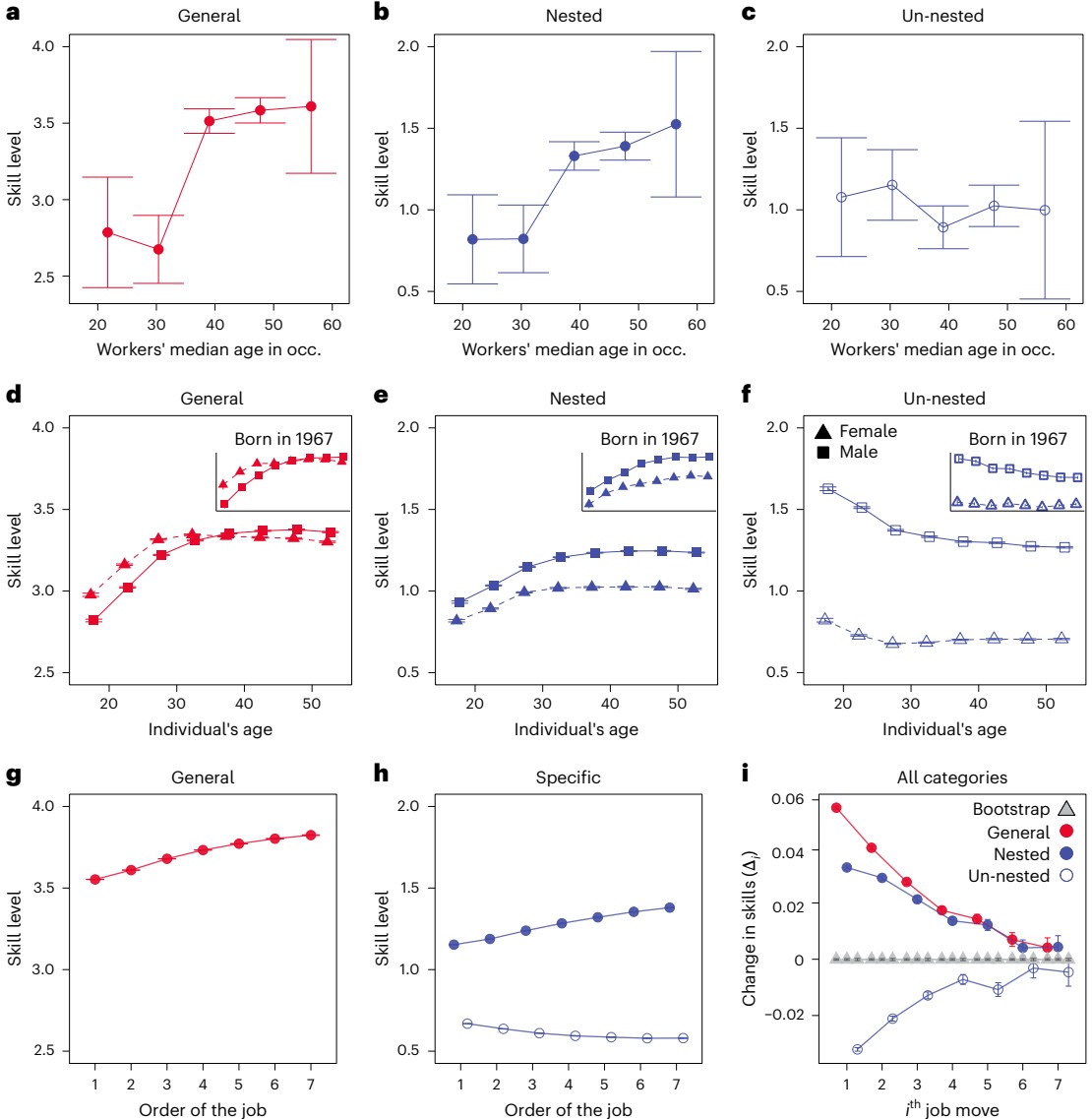

**Fig. 4 | Skill compositions with occupational ages and career trajectory.**
**a**–**c**, Average general (**a**), nested (**b**) and un-nested (**c**) skill levels of occupations
(occ.) (with 95% CI), segmented by the median ages of employees ($n_{occ}$ = 542).
Levels of general and nested skills rise with an occupation's median age, while
unnested skills show no notable correlations with median age groups.
**d**–**f**, Average general (**d**), nested (**e**) and un-nested (**f**) skill levels (with 95%
CI) against age in synthetic birth cohorts ($n_{survey}$ = 1,493,142). The insets
isolate cohorts born in 1967, while the main figures average across all cohorts
(1980–2022). Notably, general and nested skills rise markedly while unnested

skills decline until around age 30, at which point gender gaps become more
pronounced. **g**,**h**, Average general (**g**) and nested and un-nested (**h**) skill levels
(with 95% CI) over job sequences in resumes ($n_{moves}$ = 12,561,319), which favour
more nested job roles, and thus general and nested skill levels are elevated.
**i**, Changes in skill levels between *i*th job transitions in observed resumes (circles)
and bootstrapped resumes with randomized job sequences (grey triangles).
Skill profiles stabilize within the initial five jobs. The 95% CI is not immediately
discernible due to the large sample size.

($c_s$ < 0), there are fewer nodes that reach it through nested dependent
pathways, compared with skills such as programming and negotia-
tion, which have positive $c_s$. With significant implications for wages
and education, these nested pathways, therefore, may contribute to
disparities in demographics and opportunities[59].

Our analysis underscores the importance of considering both
skill generality and its structural alignment with the nested hierarchy
of human capital ($c_s$) when assessing socio-economic implications.
As previously discussed, career paths often begin with jobs requir-
ing general skills and transition into more specialized roles that
demand additional specific skills. These specialized skills can be
divided into two categories: nested-specific skills ($c_s$ > 0) that align
with the nested hierarchy and typically require higher education
with greater economic rewards, and un-nested-specific skills ($c_s$ < 0),

which fall outside the structure and tend to miss out on increasing
economic benefits as careers progress. To streamline our analysis,
we have categorized skills accordingly. In the next section, we will
explore how nested skills unfold along career trajectories and how
they interact with factors such as occupational age, wage premiums
and skill entrapment.

## Skill categories in career trajectories
We analyse three datasets of occupational sequences to examine how
skill levels evolve across various categories: occupation sequences by
their median ages, by following synthesized birth cohorts from indi-
vidual surveys, and by job transitions in resumes. Each dataset offers
unique strengths and limitations, which, when combined, provide
a coherent picture of both nested and unnested career trajectories.

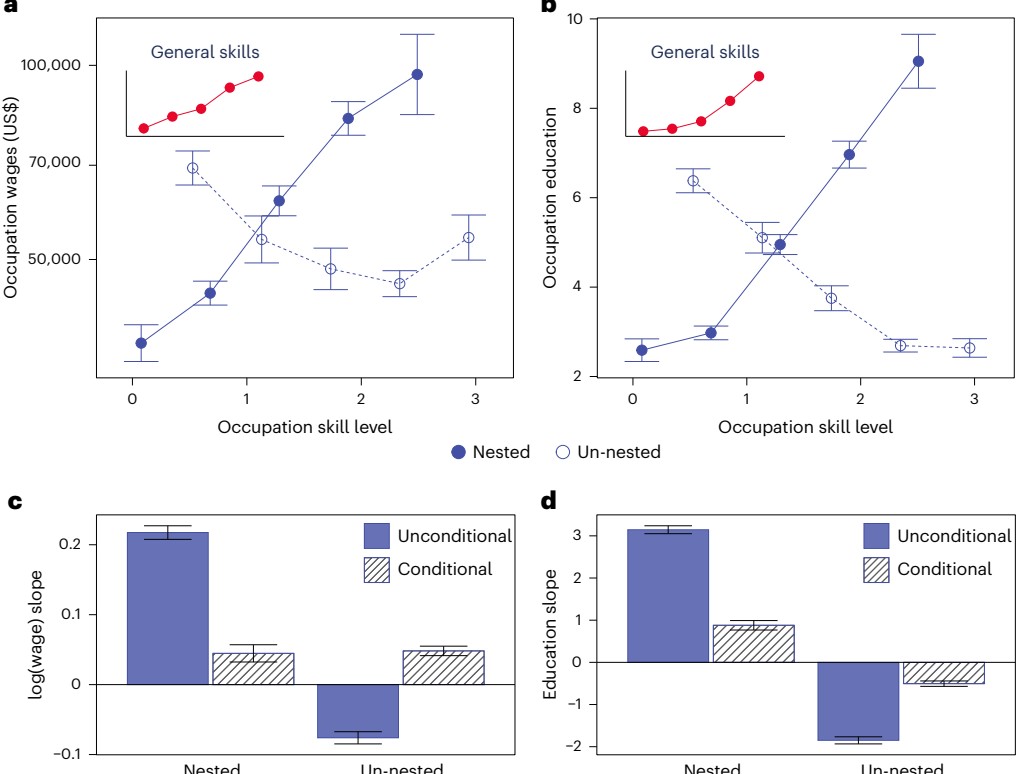

**Fig. 5 | Skill wage premiums and educational requirements. a–d**, Occupations' average annual wage (**a**) and required education levels (**b**) plotted against skill levels (with 95% CI), and their respective regression coefficients (blue bars) and standard errors in (**c** and **d**, respectively). The substantial wage premiums ($n = 774$) and higher educational requirements ($n = 968$) associated with nested specializations are much reduced (shaded bars) after controlling for general skill levels (insets), implying that the bulk of investments in and returns to specialization are conditional on accumulating general skills. The initial wage penalty for unnested specializations turns into a wage premium once general skill levels are controlled for. Full statistics are reported in Supplementary Tables 6 and 7.

We begin our analysis with the median ages of occupations, assuming that skill progression and development are closely correlated with the median age of workers in those occupation groups[28,60,61]. Figure 4a–c shows the proficiency levels of general, nested and unnested skills across occupations, segmented and arranged by the median age of workers in those roles (Methods). Consistent with previous findings, the data indicate that skill demand shifts towards nested skills as individuals advance in their careers: occupations with older workers (median age over 30 years) tend to require higher levels of both general and nested skills compared with those with younger workers. It also suggests that, as workers age and progress in their careers, these skills become increasingly essential to meet job demands. Meanwhile, the demand for unnested skills remains relatively stable across all age groups, indicating that their necessity is not influenced by worker age.

Our results hold when examining career trajectories constructed from synthetic birth cohorts of full-time respondents aged 17–55 years in the Current Population Survey (CPS) (Fig. 4d–f). Because the CPS provides yearly cross-sectional data rather than long-term longitudinal tracking, we connected birth cohorts across surveys to emulate career trajectories. For example, we created a 1967 cohort for the insets and repeated this process for all birth cohorts in the surveys between 1980 and 2022 (Methods)[10,11]. These results are consistent with our earlier findings with a more detailed pattern. Age 30 emerged as a marked turning point. Up to this age, there is a sharp increase in both general and nested skill levels, while unnested skills show a moderate decline. After age 30, the overall changes in skill levels tend to stabilize.

The second dataset includes additional demographic information about the respondents, allowing us to decompose our findings by gender and, later, by race. For example, analysing skill trends by

gender reveals a gap in specializations that emerges around age 30, as shown in Fig. 4d–f. While men continue to grow their general and nested skills until their 50s, for women, the growth in these skills plateaus in their early 30s, which coincides with the typical age range for first-time mothers in the USA. We further analyse the influence of parenthood on male and female workers with and without children, as well as working schedules and hours in Supplementary Sections 7.1 and 7.2. Our findings are robust to controlling for yearly economic conditions (Supplementary Fig. 28) and educational attainment (Supplementary Figs. 29 and 30).

Lastly, we analyse over 70 million individual job transitions recorded in 20 million resumes. Figure 4g,h shows that the results of these direct observations are again consistent with our two earlier findings. The baselines for general and nested skill levels are shifted because the resume dataset favours more nested job roles. Once again, career trajectories (the $i$th job transition) show a steady accumulation of both general and nested skills ($\Delta_i > 0$) during the first five transitions. In addition, Fig. 4i indicates that general skills advance faster than nested skills early in a career ($\Delta_{i<3}^{\text{general}} \gg \Delta_{i<3}^{\text{nested}}$), and even later, their growth remains comparable to specific skills ($\Delta_{i>3}^{\text{general}} \approx \Delta_{i>3}^{\text{nested}}$). We provide skill growth along randomized sequences (grey triangles), $\Delta_i^{\text{random}} \approx 0$, to confirm that the observed job sequences explain their corresponding skill growth curves (Supplementary Section 4.1).

All three empirical observations consistently depict career paths where both general and nested skills ($c_s > 0$) grow, while unnested ones ($c_s < 0$) remain relatively underdeveloped or even decline. Moreover, the results prompt interesting recurring patterns. First, skill advancement extends well beyond formal education, suggesting that nested specialization pathways persist throughout a career, beyond the

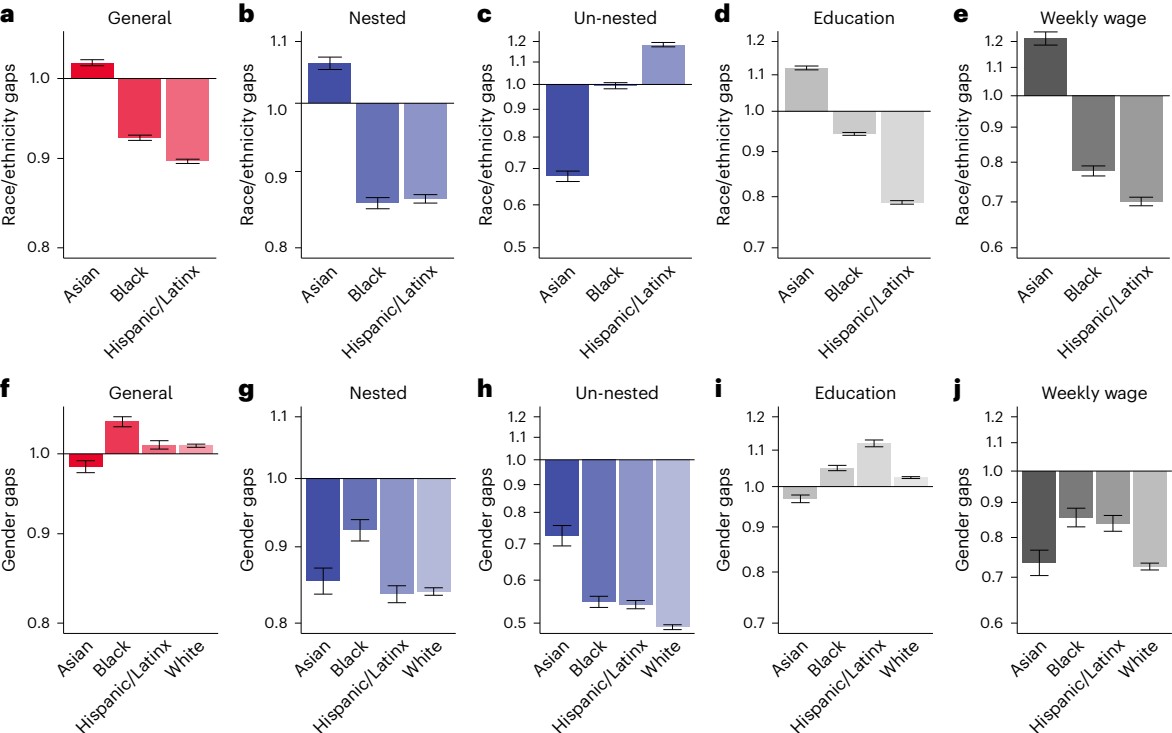

**Fig. 6 | Skill disparity in demographic distribution of race/ethnicity and gender. a–e**, The relative average general (**a**), nested (**b**) and un-nested (**c**) skill level, education level (**d**) and weekly wages (**e**) for Asian ($n = 86,055$), Black ($n = 180,304$) and Hispanic/Latinx ($n = 291,493$) workers compared with white ($n = 1,375,617$) workers (expressed as ratios). **f–j**, The relative average general (**f**), nested (**g**) and un-nested (**h**) skill level, education level (**i**) and weekly wages (**j**) for female workers ($n_{\text{Asian}}^{\text{Fem}} = 34,586$, $n_{\text{Black}}^{\text{Fem}} = 89,230$, $n_{\text{Hisp.}}^{\text{Fem}} = 105,371$ and $n_{\text{White}}^{\text{Fem}} = 518,688$) compared with male workers. The 95% CI around each ratio is calculated by bootstrapping subsamples (Methods). These differentials are robust to measurement (Supplementary Fig. 49) and to time-variant economic factors (Supplementary Fig. 54) and follow age trends similar to those seen in Fig. 4. Supplementary Figs. 51 and 52 show that the gaps have narrowed over time.

schooling phase (Supplementary Figs. 29 and 30). This observation challenges the traditional view that education is the primary driver of human capital development[11,40,62–64].

Second, the findings question the simple linear progression model, which assumes that basic general skills precede advanced specialized ones, as is often portrayed in economics, sociology and psychology and even in our study's initial assumptions[28,65]. Instead, we observe that general skills continue to develop alongside specific ones, even after workers enter specialized roles. For example, critical thinking is a prerequisite for transitioning into roles requiring new skills such as negotiation, as indicated by the asymmetric conditional probabilities in Fig. 2. However, even within a negotiation role, further advancement depends on honing higher levels of critical thinking. Thus, career trajectories unfold through nested specializations, where the ongoing development of general skills supports their dependent, nested counterparts, thereby echoing the cumulative and sequential nature of skill development.

Next, we explore the connection between each specialization pathway and wage premiums. Figure 5a,b shows that both educational requirements and average annual wages rise with proficiency in nested specific skills. However, as shown in Fig. 5c, these observed wage premiums associated with nested skills almost fully disappear when controlling for an occupation's general skill requirements. This finding underscores the previous argument that specific skills complement general ones rather than stand alone, while general skills remain largely independent. Therefore, even in specialized roles, a strong foundation in general skills remains essential. That said, the wage penalties associated with unnested skills ($c_s < 0$) in Fig. 5 turn into wage premiums of a magnitude comparable to those of nested skills once general skill requirements are factored in. These findings suggest that, although unnested skills do hold value

in the labour market, their potential wage premiums are diminished by a lack of dependency structure, almost like a penalty for missing that foundation. This finding provides structural insights into career development pathways and the broader evolution of workforce skills.

We present a series of tests to demonstrate that our findings remain robust even when controlling for education, training and workplace experience, as well as holding across subsamples of major occupational groups (Supplementary Section 5, Supplementary Table 5 and Supplementary Figs. 32–34 and 39). We also confirm our findings hold even when excluding managerial roles and social skills (Supplementary Sections 9.1 and 9.2 and Supplementary Figs. 62 and 64).

### Disparity, skill entrapment and long-run wage penalties across demographic groups

We now examine how our skill taxonomy helps in understanding salient features of the labour market. Specifically, we analyse skill categories across demographic groups to determine whether the nested architecture in human capital explains socio-demographic inequalities, skill entrapment and persistent wage penalties. Figure 6a–e compares skill, education and wage differences between racial/ethnic groups and their white peers. The results show, first of all, notable wage gaps between Black and Hispanic workers on the one hand, and Asian workers and the baseline of white workers, on the other hand. These wage gaps are accompanied by employment in jobs with lower requirements for nested skills for Black and Hispanic workers. Notably, for Hispanic workers in particular, another critical factor emerges: an elevated requirement for unnested skills, which may contribute to skill entrapment.

To understand this finding in more detail, we examine a case study of how language-skill requirements may limit access to jobs requiring certain nested skills (Supplementary Section 3.5). We first distinguish

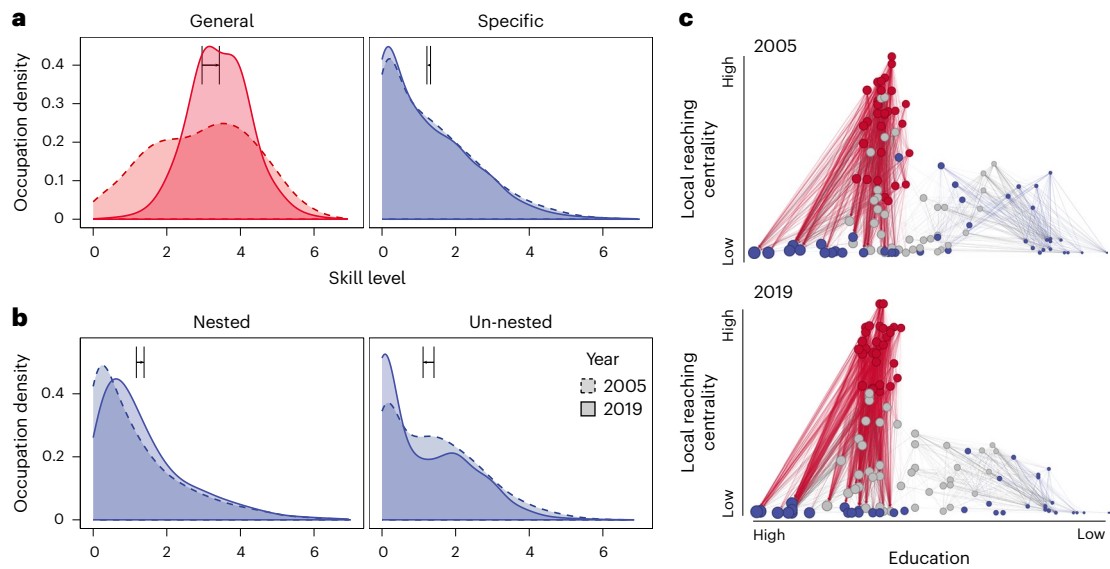

**Fig. 7 | Historical changes in the skill structure between 2005 and 2019. a**, The distribution of skill levels for different skill groups. The arrow indicates the shift in average skill levels over time. Unlike the positive shifts in general skills, the shift in specific skills is less noticeable. **b**, The distribution of skill levels for nested and unnested skills. The arrow shows the shift in average skill levels over time. While nested skills follow the shift in general skills, the demand for unnested skills has decreased. **c**, The skill structure has become more nested, as indicated by a lower checkerboard score (from 438.67 to 356.4) and temperature (from 40.07 to 31.89), alongside higher NODF (from 39.06 to 41.72) and $N_c$ (from 573,873 to 651,030) in addition to a growing divide between nested and unnested specific skills (blue clusters on the left and right).

between nested skills that depend on language skills (general) and those that do not, and find substantial gaps in language-dependent nested skills among Hispanic workers, particularly those who recently arrived in the USA. This disparity often leads to the development of unnested skills, creating skill traps associated with long-term wage penalties (Supplementary Fig. 31)[23]. Taken together, these findings suggest that closing wage gaps for Black workers and Hispanic workers may require tailored solutions.

Next, Fig. 6f–j shows skill gaps between men and women across social groups. The most pronounced disparities occur for specific skills: Except in the Asian subsample, women generally work in occupations requiring more education and higher levels of general skills than men do. However, women's jobs do not require similar levels of nested skills. These disparities probably contribute to the well-known gender wage gap observed in the right-most panel. This gap has narrowed over time, as shown in Supplementary Fig. 52, but the disconnect between education and general skills on the one hand and wages and nested skills on the other remains puzzling.

To understand this in more detail, we examine another case study on parenthood and women's tendency to work in jobs with more regular and predictable schedules that impact both wages and skill development in Supplementary Sections 7.1 and 7.2 (refs. 66–68). Our analysis shows that having children is associated with reduced general and nested skills for women. By contrast, men with children tend to have higher levels of these skills compared with men without children. A key factor appears to be work schedules: when we control for irregular hours and overtime, the gender gap in nested skill requirements decreases by more than a third.

Finally, we examine the geographic distribution of skills across categories. Our analysis shows that general skills tend to concentrate in densely populated urban areas (Supplementary Section 6), a finding consistent with previous research on the diversity and complexity of economic activity in large urban economies[4,36,69–73]. Moreover, differences in general skills account for about one-third of the well-documented urban wage premium[74], which is associated with employment in large cities.

### Increasing nested structure in human capital

Figure 7 shows the historical changes in the hierarchical skill structure between 2005 and 2019. Over this period, the skill structure has become increasingly nested, as indicated by a decrease in the checkerboard score (from 438.67 to 356.4) and temperature (from 40.07 to 31.89), alongside increases in NODF (from 39.06 to 41.72) and $N_c$ (from 573,873 to 651,030)[55,56]. This trend towards a more nested structure is attributed to increasingly uneven job requirements, with growth in high-dependency branches and a decline in low-dependency branches. Workers with broad skill sets have seen higher wage premiums, probably driven by demands for nested specialization skills (general and nested specific skills with $c_s > 0$), while demand for unnested skills has declined (Supplementary Fig. 40). These trends are perhaps attributed to the growing complexity and interdependence of the economy (Supplementary Fig. 57)[7,75].

The growing divide between nested and unnested specializations raises concerns, echoing ongoing discussions on job polarization[20,22,76–78] (Fig. 7c). Given the important role of nested skills in career progression and wage expectations, the growing divide potentially further reinforces demographic and regional disparities. Workers with insufficient foundational general skills often find themselves stuck in unnested specialization paths with limited opportunities for upward mobility (Supplementary Figs. 20 and 31)[21–23,77]. Thus, policymakers and educators must prioritize fundamental skill development across all demographic groups and regions to mitigate these structural disparities. In other words, investing in education and training programmes that cultivate general skills and unlock specialized pathways is essential to addressing this growing divide and improving economic mobility.

### Discussion

We present a structural framework for the nested hierarchy in human capital manifested in job requirments. Our findings show that skills aligned with the nested hierarchy tend to yield greater economic rewards, while those outside this structure often miss out on the growing benefits that come with career progression. This structural classification of skills explains traditional dichotomies without directly

relying on socio-economic contexts. Notably, skills with positive alignment scores ($c_s > 0$)—such as cognitive abilities common in white-collar roles—are rewarded more substantially and better aligned with educational systems. Indeed, the structural properties of skills unveil socio-economic contexts.

As society continues to evolve, mastering a universal set of skills is no longer feasible for individuals across all fields[1,2]. Our analysis shows that the skill structure is becoming more nested with the modern economy's growing demand for specific sets of skills, knowledge and abilities that require extensive educational and work trajectories[17,40,62,79–81]. In response, the process of selecting and acquiring the right set of skills has become increasingly crucial, but the most valuable skills are often embedded within specific domains and require not only cumulative learning but also prerequisites to unlock. This hierarchical skill structure is evident in both education and career trajectories, shaping broader social and economic systems.

As such, our findings contribute to the field of economic complexity by illustrating how connections within skill networks are conditional and how structures become increasingly nested as complexity and specialization grow. The directional dependencies we identify challenge the symmetric relation in traditional co-occurrence networks[36,82–85]. This added depth sheds light on structural changes in economic complexity, showing how knowledge, capabilities and technologies are accumulated within populations and how precedence relations between activities shape economic outputs at the firm, city, regional or national level[25,70,71,83,86–92].

In this way, we bridge economic theories of career progression and wage premiums through hierarchical structures[40,62] and economic complexity models that describe development through interdependencies between skills and capabilities[25]. The hierarchical organization of skills, long assumed to shape developmental trajectories[16,36,69], is revealed in our analysis to be directional rather than mutual through a dependency structure. These structured pathways systematically shape professional development and the socio-economic landscape, driving differences in rewards and career accessibility based on early choice of skill acquisition[36,63,76,93–95]. As complexity and specialization increase, we observe the skill dependency structure becoming more nested. This deeper nested structure imposes greater constraints on individual career paths, amplifies disparities and has macroeconomic implications, affecting the resilience and stability of the entire system[24,54,96–98].

We acknowledge the limitations of this approach. Hierarchical dependencies within human capital can be analysed using different methodologies and data sources. Our framework offers just one perspective. Furthermore, our analysis is centred on the US labour market, which has its own unique characteristics in its own education systems, industry composition and urban structure. Generalizing these findings to other settings, such as entrepreneurship[99] or economies at different stages of development[100], remains an open question for future research.

## Methods

### O*NET

Occupational Information Network (O*NET) provides occupation-specific descriptors on almost 1,000 occupations across the US economy[29]. This publicly accessible database is regularly updated with input from workers across various occupations. For our analysis, we use the 2019 version to avoid distortions from the coronavirus disease 2019 pandemic and compare it with the 2005 version for historical changes, as it is the earliest dataset that offers consistent skill and education categories across many occupations. We define skills broadly, covering skills, knowledge and abilities (collectively referred to as skills), covering a total of 120 items. Each occupation lists the importance and required proficiency level, as well as educational requirements for each skill. Importance is rated from 1 to 5 as

not important to extremely important, while proficiency level ranges from 0 to 7, low to high. This numerical rating creates an occupation–skill matrix, with each entry representing the skill's importance or required level. Our main analysis focuses on skill levels to construct demand profiles for skill generality and skill growth along individual career trajectories. However, because skill importance and level are highly correlated (0.94), our findings remain consistent regardless of the measure used[50].

### OEWS

Occupational Employment and Wage Statistics (OEWS) provides wage and employment data at both national and regional levels for each occupation. For consistency with the O*NET dataset, we use the 2005 and 2019 versions, although we find that including and aggregating data from several years before and after these dates does not alter our results. While both O*NET and OEWS conveniently use the Standard Occupational Classification (SOC) system, OEWS data are available at a more aggregated level (774 unique titles with 6-digit SOC), whereas O*NET provides more detailed data (968 unique occupations with 8-digit SOC), requiring us to aggregate them for alignment.

### CPS

CPS is a monthly survey conducted by the Census Bureau for the BLS[101]. For our analysis, we use the median age of workers in occupations for 2019, along with detailed demographic data from 1980 to 2022, including respondents' occupation titles, wages, hours worked, gender and race/ethnicity. As with other datasets, matching occupational units requires the use of a crosswalk, detailed in the corresponding section.

### Resume dataset

The resume dataset contains 20 million anonymized resumes spanning from 2007 to 2020, detailing 70 million job sequences classified using the 8-digit SOC system. Unlike the CPS dataset, this dataset lacks demographic information such as age, gender and race, as all identifying information has been removed.

### Skill generality groups

O*NET provides the required proficiency levels for each skill across various occupations. To measure a skill's generality—how broadly it applies across occupations—we define the skill demand profile as the distribution of occupations requiring that skill at different proficiency levels. We then group skills by the shape of their demand profiles (Fig. 1a–c) using a $k$-means clustering algorithm. Based on our statistical tests, the optimal number of groups is three ($k = 3$), which we use in our main analysis, focusing on the effects of general and specific skills, while downplaying intermediate skills. However, our findings remain robust even when using $k = 2$ or $k = 4$ (Supplementary Section 1).

### Local reaching centrality

Local reaching centrality provides an alternative measure of skill generality once the skill dependency network is defined. It quantifies how many skills depend on a focal skill by calculating the number of skills reachable from it through outgoing edges (that is, in the direction of dependency)[51]. A higher reaching centrality in the hierarchical structure indicates a skill with more dependencies (thus, a more general skill), offering an additional metric for assessing skill generality.

### Skill dependency using asymmetric conditional probability

To calculate the conditional probability of one skill being included in job requirements given the presence of another skill, we first convert the continuous [0, 7] skill levels ($\text{Level}_{o,s}$) into binary variables indicating the presence/absence of each skill for each occupation ($m_{o,s}$), using the widely used disparity filter[102]. This algorithm identifies the statistically significant disparities of a skill's presence in an occupation, comparing it against random expectation. We set the algorithm's

parameters to satisfy two conditions: first, the ranking of skills by strength in the binary representation should be preserved (that is, skills' ranking by $\sum_o \text{Level}_{o,s}$ aligning closely with those by $\sum_o m_{o,s}$). Second, the ranking of occupations by skill levels should be maintained in a binary representation (the ranking by $\sum_s \text{Level}_{o,s}$ matches those by $\sum_s m_{o,s}$). Basically, these conditions ensure that high-skilled jobs remain high-skilled, and widely used skills retain their ranking in both representations (Supplementary Section 3.1).

Once the presence/absence is set, we next calculate conditional probabilities for every skill pair to infer their dependence directions, following ref. 33. Two key thresholds are used to filter out random noise: $z_{\text{th}}$, which sets the threshold for significant co-occurrences to be considered meaningful, and $\alpha_{\text{th}}$, which sets the threshold for asymmetry to determine a meaningful directional dependency. After filtering out non-significant co-occurrences using $z_{\text{th}}$, we compute the conditional probabilities $P(u|v)$ and $P(v|u)$. A directional dependency $v \to u$ is assigned only when $P(u|v)$ is substantially greater than $P(v|u)$, based on the threshold $\alpha_{\text{th}}$. Note that $\alpha_{\text{th}}$ is differentially weighted for each pair of skills so that it accounts for heterogeneous skill's node degrees, and we test various levels of $\alpha$ (see equation (6) in Supplementary Section 3). The dependency strength/weight is then calculated as a function of the difference between these conditional probabilities, adjusted for the null model that accounts for the expected number of shared occupations between skills $u$ and $v$, given their respective degrees (see equation (7) in Supplementary Section 3).

### Reachability with arrival probability

Reachability with arrival probability quantifies the chances of getting to the focal skill given the prerequisite connections based on arrival probability, a version of hitting probability, from node $i$ to $j$ by random walks[58]. For different source and target skills $i \neq j$, this is numerically equivalent to first deriving the probability of random walks of length $l$ by raising the weighted-directed adjacency matrix (skill dependency network in Fig. 2), $M$, to power $l$, and then calculating $R_{i,j} = \sum_l M_{i,j}^l$. We obtain the final arrival probability by summing over a sufficient number of path lengths until reaching the saturation points. To compute arrival probabilities for focal skills (such as programming, negotiation and repairing) in Fig. 2c–e, we apply the R package markovchain[103].

### Nested skills

Nestedness ($N$) is a structural characteristic originally developed for ecological systems to assess how species interactions are organized, particularly in a way where the interactions of specialists (species with fewer interactions) are subsets of those of generalists (species with more interactions)[53,98,104,105]. For example, in a plant–pollinator interaction, if a specialist pollinator visits only a few plant species, those plants are also visited by more generalist pollinators, creating a hierarchically nested structure. This hierarchical arrangement is often visualized as an upper-triangular or pyramid-like shape in an interaction matrix, where rows and columns are ordered by the total interactions. However, there is no consensus on how to precisely measure deviations from this ideal upper-triangular structure, so different metrics are used, including the overlap index ($N_c$), checkerboard score, temperature and NODF. We test them to ensure the robustness of our findings (Supplementary Section 2).

Nestedness has been applied beyond ecology to explore socio-economic structures, such as occupations and technological capabilities across nations, regions, urban areas and companies[52,54,90,106]. In our study, nestedness $N$ quantifies the extent to which the presence of narrowly applicable skills (specialists) in job requirements is consistently contained within the broader application ranges of skills (generalist).

The skill–occupation matrix ($m_{s,o}$) in Supplementary Fig. 6 shows an imperfect upper-left triangle when occupations and skills are sorted by their marginal totals. We assess whether skill $s$ could be present in

different job requirements to improve the overall nested structure, following[52]. We calculate the observed nestedness ($N$) and the counterfactual nestedness ($N^*$) by randomizing the presence of skill $s$ across job requirements. The deviation between these values, normalized by the standard deviation of this counterfactual $N^*$, indicates the contribution of skill $s$ to the overall nestedness: $c_s = (N - \langle N_s^* \rangle)/\sigma_{N_s^*}$. Here, $N$ is a nestedness score of the empirically observed interactions while $\langle N_s^* \rangle$ and $\sigma_{N_s^*}$ are the means and standard deviation derived from the null model to create counterfactuals. For each focal skill $s$, we run 5,000 iterations to measure its mean and standard deviation[107]. Again, we use the overlap index, checkerboard score, temperature and NODF to quantify nestedness $N$ (refs. 55–57,108). We then categorize specific skills with $c_s > 0$ as 'nested' and those with $c_s < 0$ as 'un-nested'. Supplementary Table 2 provides the list of skills in these categories, and further details and robustness checks are in Supplementary Section 2.

The imperfect nested structure with negative $c_s$ may be attributed to human capital constraints within occupations, similar to the concept of limited carrying capacity in systems ecology. Unlike skills, which show a broad range of distribution profiles (Fig. 1), Supplementary Fig. 5 shows that the scope of occupations remains mostly constant, suggesting that the total amount of skill levels embodied in occupations does not differ much as if constrained. Supplementary Section 2 shows that these constraints persist regardless of how well the occupations are paid and how advanced an education they require. It seems that the limited cognitive and physiological capacity of individual workers imposes a natural boundary on how much one person can learn and perform in a single job[2,109]. These constraints drive specialization in complex jobs, resulting in modular structures within skill–occupation interactions where certain skills become mutually exclusive within an occupation in addition to the nestedness, creating nested-modular[110,111], explaining the uneven nestedness observed in our skill hierarchy in Fig. 2.

### Expected education level for skills and occupations

The education variable in O*NET is categorized into 12 discrete grades ($\text{edu}_e$), ranging from below high school (1) to post-doctorate (12). Each occupation ($o$) consists of the percentage of employees who require each grade of education ($f_{o,e}$). We calculated the expected education grade for each occupation as an average education grade weighted by each employee's fraction, $\langle \text{edu} \rangle_o = \sum_e f_{e,o} \times \text{edu}_e$. For instance, 6.05% of Chief Executives require a minimum of some college courses (4), 4.23%, an associate's degree (5) and 21.61%, a bachelor's degree (6), among other levels. Then, the expected education grade $\langle \text{edu} \rangle_o = \dots 0.0605 \times 4 + 0.0423 \times 5 + 0.2161 \times 6 + \dots$. We then use $\langle \text{edu} \rangle_o$ to estimate the educational requirement of a skill $s$, expressed as $\langle \text{edu} \rangle_s = \frac{\sum_o \langle \text{edu} \rangle_o \times \text{Level}_{o,s}}{\sum_o \text{Level}_{o,s}}$, where Level denotes the level of the skill $s$ required by occupation $o$.

### Synthetic birth cohorts

The median ages of workers in each occupation are derived from the 2019 CPS, and synthetic birth cohorts are constructed using individual survey data jointly conducted by the US Census Bureau and the BLS[101]. The different occupational taxonomies in the two datasets are mapped by the BLS crosswalk. Although the CPS provides monthly surveys for each household, these surveys are not long enough to cover long-term trends. Therefore, we create synthetic birth cohorts by stitching together snapshots of individuals born in the same year across different survey rounds. This method, widely used in the literature, allows us to gain insights into the evolution of various social, economic and demographic characteristics over time when longitudinal data are unavailable[10,11,112,113].

For example, to construct a synthetic cohort for those born in 1970, we identify individuals whose birth year was 1970 in CPS surveys conducted in 1995, 1996, 1997 and so on, up to 2015. We then compile this data as if we were following the same group of individuals throughout their lives, as shown in the inset of Fig. 4. This approach is referred to as

a synthetic birth cohort because it is not a real cohort in the traditional sense; rather, it is constructed by aggregating data from different individuals born in the same year. The individuals surveyed in each CPS round vary, even though they all share the same birth year. By tracking individuals born in the same year across multiple survey rounds, nevertheless, we can observe changes in behaviours or characteristics of interest as people age, albeit with different individuals representing the cohort at each point in time.

### Demographic analysis
CPS microdata provide information on gender and race/ethnicity demographics. We chose four race/ethnicity categories—white, Black, Asian and Hispanic—as they make up the bulk of the sample in Fig. 6. To identify individuals of Hispanic background, we overrode the information of the race (RACE) variable with the Hispanic (HISPAN) variable in the CPS. Wages are adjusted for inflation, and the number of hours worked is accounted for to calculate an adjusted weekly wage, allowing comparability across the population. We include only full-time workers aged 18–55 who were employed at the time of the survey and earn at least US$10,000 annually to minimize the impact of attrition and early retirement. For each demographic category, the average skill level is calculated for its occupational composition OCC, that is, $\mathrm{Level}_{occ,s} = \langle \mathrm{Level}_{o,s} \rangle_{o \in OCC}$.

The race/ethnic disparities in Fig. 6 are shown as ratios of each demographic measure (average skill levels, education level and weekly wages) relative to those of white workers. Likewise, the gender gap within each race/ethnicity is measured as the ratio of these quantities compared with male workers within that group. Due to the absence of a matched sample, 95% confidence intervals (CIs) are derived using random subsampling. In each iteration, we take 10% of the subpopulation of interest (for example, Asian male and Asian female workers) and estimate all corresponding measures. This sampling and estimation process is repeated 10,000 times, generating a distribution for each measure from which the 95% CIs are derived. The skill, education and wage estimations of Fig. 6 are averaged over the years (1980–2022). Supplementary Figs. 51 and 52 capture temporal patterns of these factors, exhibiting the gaps have narrowed over time. In addition, Supplementary Figs. 53 and 54 show that the skill differentials between male and female workers, which begin around the age of 30 (Fig. 4), are evident across all racial and ethnic groups.

### Skill level changes in career trajectories
The resume dataset comprises 20 million individual resumes collected between 2007 and 2020, with over 70 million job transition sequences (classified as 8-digit SOC). For our primary analysis, we exclude job transitions shorter than 1 year or those occurring within the same occupation (for example, moving between companies without a change in job role). This exclusion is due to anomalies observed in such cases, where individuals in roles like janitor or model appeared to transition directly to CEO positions, sometimes with overlapping timelines. We confirm that our findings remain robust despite this exclusion (see Supplementary Section 4 for further details).

We then calculate the average skill levels of those occupations appearing in $i$th sequences in each resume, and their increase/decrease compared with the next job requirements ($\Delta_i$). To verify whether the observed trends are genuinely related to career trajectories, we randomize the $i$th sequences in resumes and compare these randomized patterns with the empirically observed skill changes. This confirms that the trends observed are indeed specific to actual career trajectories.

### Temporal evolution of skill structure
We analyse temporal changes in the skill structure from O*NET[29]. We choose two sufficiently temporally distant snapshots of the data to capture the structural differences: version 9.0 in 2005, as it is the first version comparable to the most recent version while providing satisfactory coverage of occupational information (such as education and wage), and version 24.1 in 2019, as it is the most recent version without the potential irregular patterns introduced by the pandemic. A key empirical challenge is that the classification system is continuously updated to reflect technological advancements, economic shifts and social changes[36,38,39].

We created a crosswalk between occupation classifications in 2005 and 2019 that is not immediately available other than between two consecutive years. Occupation codes in 2005 are matched to those in 2006, and then those in 2006 to 2009, and so on to 2019. Our crosswalk automatically matches 968 occupations in 2019 skill data and 941 unique occupations in the 2005 skill data, and the rest are manually matched[36]. Using these occupations and their skill levels in 2005, we construct the skill structure for 2005 in Fig. 7c, using comparable parameters and layouts for both years to make the networks as comparable as possible (Supplementary Section 8).

### Reporting summary
Further information on research design is available in the Nature Portfolio Reporting Summary linked to this article.

### Data availability
This study has used three publicly available data sources: O*NET (https://www.onetonline.org), CPS (https://www.bls.gov/cps) and occupational information, including wages, employment and age demographic prepared by the US BLS (https://www.bls.gov/oes). Anonymized resumes in Fig. 4g–i are based on proprietary data that can be purchased from Burning Glass Technologies (https://lightcast.io).

### Code availability
All algorithms previously developed and used in processing and preparing data and analysis have been cited. The code used for data processing and analysis is available via GitHub at https://github.com/mohhoss/Nested-Skills-in-Labor-Ecosystems.

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

## Acknowledgements

H.Y. and M.H. acknowledge support of the National Science Foundation grant award number EF-2133863. We are grateful to Y.-Y. Ahn, I. Hong, B. Lengyel, M. Yildirim, J. McNerney, M. Frank, C. Esposito and B. Uzzi for their valuable discussions and feedback. H.Y. thanks the participants of the ANN-SONIC-NICO workshop at Northwestern, the Growth Lab seminar at Harvard, the KAIST College of Business seminar and the ANET seminar at ELKH for their valuable feedback during the initial stages of the manuscript. F.N. acknowledges financial support from the Austrian Research Agency (FFG), project no. 873927 (ESSENCSE). H.Y. acknowledges the NRF Global Humanities and Social Sciences Convergence Research Program (2024S1A5C3A02042671) and the support from the Institute of Management Research at Seoul National University. The funders had no role in study design, data collection and analysis, decision to publish or preparation of the manuscript.

## Author contributions

M.H., H.Y. and F.N. conceived the project and designed the analytical framework; M.H. performed empirical analyses with help from H.Y. and F.N.; M.H., H.Y., F.N. and L.Z. discussed and interpreted results; M.H. and H.Y. wrote the manuscript; M.H., H.Y., F.N. and L.Z. edited the manuscript.

## Competing interests

The authors declare no competing interests.

## Additional information

**Correspondence and requests for materials** should be addressed to Hyejin Youn.

# Reporting Summary

## Statistics

For all statistical analyses, confirm that the following items are present in the figure legend, table legend, main text, or Methods section.

| n/a | Confirmed | |
|---|---|---|
| ☐ | ☒ | The exact sample size (*n*) for each experimental group/condition, given as a discrete number and unit of measurement |
| ☒ | ☐ | A statement on whether measurements were taken from distinct samples or whether the same sample was measured repeatedly |
| ☒ | ☐ | The statistical test(s) used AND whether they are one- or two-sided *Only common tests should be described solely by name; describe more complex techniques in the Methods section.* |
| ☐ | ☒ | A description of all covariates tested |
| ☒ | ☐ | A description of any assumptions or corrections, such as tests of normality and adjustment for multiple comparisons |
| ☐ | ☒ | A full description of the statistical parameters including central tendency (e.g. means) or other basic estimates (e.g. regression coefficient) AND variation (e.g. standard deviation) or associated estimates of uncertainty (e.g. confidence intervals) |
| ☒ | ☐ | For null hypothesis testing, the test statistic (e.g. *F*, *t*, *r*) with confidence intervals, effect sizes, degrees of freedom and *P* value noted *Give P values as exact values whenever suitable.* |
| ☒ | ☐ | For Bayesian analysis, information on the choice of priors and Markov chain Monte Carlo settings |
| ☒ | ☐ | For hierarchical and complex designs, identification of the appropriate level for tests and full reporting of outcomes |
| ☒ | ☐ | Estimates of effect sizes (e.g. Cohen's *d*, Pearson's *r*), indicating how they were calculated |

*Our web collection on statistics for biologists contains articles on many of the points above.*

## Software and code

Policy information about availability of computer code

| Data collection | This study has used three publicly available data sources, Occupational Information Network (O*NET), Current Population Survey (CPS), occupational information, including wages, employment and age demographic prepared by the US Bureau of Labor Statistics (BLS). We have also utilized a proprietary source containing a database of anonymized resume, widely used in the literature. |
|---|---|
| Data analysis | All algorithms previously developed and used in processing and preparing data and analysis have been cited. The code used for data processing and analysis is available at https://github.com/mohhoss/Nested-Skills-in-Labor-Ecosystems. |

For manuscripts utilizing custom algorithms or software that are central to the research but not yet described in published literature, software must be made available to editors and reviewers. We strongly encourage code deposition in a community repository (e.g. GitHub). See the Nature Portfolio guidelines for submitting code & software for further information.

## Data

Policy information about availability of data

All manuscripts must include a data availability statement. This statement should provide the following information, where applicable:
- Accession codes, unique identifiers, or web links for publicly available datasets
- A description of any restrictions on data availability
- For clinical datasets or third party data, please ensure that the statement adheres to our policy

This study has used three publicly available data sources: Occupational Information Network (O*NET): https://www.onetonline.org, Current Population Survey

(CPS): https://www.bls.gov/cps, occupational information, including wages, employment and age demographic prepared by the US Bureau of Labor Statistics (BLS): https://www.bls.gov/oes. Anonymized resumes in Fig. 4 (g-i) are proprietary data.

# Research involving human participants, their data, or biological material

Policy information about studies with human participants or human data. See also policy information about sex, gender (identity/presentation), and sexual orientation and race, ethnicity and racism.

| | |
|---|---|
| Reporting on sex and gender | Studying genders is a not a central theme of this study. However, we have utilized the richness of Current Population Survey (publicly available) to uncover possible sources of social and economic gender disparities as implied by our framework. In consistent with the Nature portfolio policy, we have adopted the term "gender", given the social and cultural context of our study. Where applicable, we have articulated how our findings differ as it pertains to different genders. To be clear, Current Population Survey is the only data source used where (merely binary) gender information is made available. |
| Reporting on race, ethnicity, or other socially relevant groupings | Studying race and ethnicity is a not a central theme of this study. However, Current Population Survey (publicly available) includes self-declared information on race/ethnicity of surveyed individuals as per federal guidelines (https://www.bls.gov/cps/rvcps03.pdf). We have utilized such information to uncover possible sources of social and economic race/ethnic disparities as implied by our framework. Given the limitations of the dataset, we have explicitly examined a subset of such social groups, and have articulated such decisions in the Method section. |
| Population characteristics | NA |
| Recruitment | NA |
| Ethics oversight | NA |

Note that full information on the approval of the study protocol must also be provided in the manuscript.

# Field-specific reporting

Please select the one below that is the best fit for your research. If you are not sure, read the appropriate sections before making your selection.

☐ Life sciences   ☒ Behavioural & social sciences   ☐ Ecological, evolutionary & environmental sciences

For a reference copy of the document with all sections, see nature.com/documents/nr-reporting-summary-flat.pdf

# Behavioural & social sciences study design

All studies must disclose on these points even when the disclosure is negative.

| | |
|---|---|
| Study description | We study the structure of skill dependencies in workplace skills, reframing classic topics in human capital. The study is quantitative, combining network analysis, data and regression analysis. |
| Research sample | We include brief descriptions of the four datasets used in this study to supplement in-depth description offered the Method section: 1. The Occupational Information Network (O*NET) includes survey records of job-oriented attributes and worker-oriented descriptors that pertains to a representative sample of individuals and establishment in the United States. 2. Current Population Survey (CPS) is a primary source of monthly labor force statistics. We have utilized CPS micro-data that contains surveyed demographic and work-related information, as well as aggregate-level estimates on occupational age information. 3. Occupational Employment and Wage Statistics (OEWS) from the Bureau of Labor Statistics (BLS). OEWS program produces employment and wage estimates annually for approximately 830 occupations. These estimates are available for the nation as a whole, for individual states, and for metropolitan and non-metropolitan areas; national occupational estimates for specific industries are also available. We have utilized national-level and metropolitan and non-metropolitan information. 4. Proprietary resume data offers information on over 70 million jobs organized in 20 million individual resumes between 2007 and 2020. |
| Sampling strategy | We used the full set of publication record of countries. No sampling strategy is used. |
| Data collection | No primary data is collected. |
| Timing | 2005-2022 |
| Data exclusions | We used the complete publication records of countries. |
| Non-participation | NA |
| Randomization | Our study doesn't involve random-control design. |

# Reporting for specific materials, systems and methods

We require information from authors about some types of materials, experimental systems and methods used in many studies. Here, indicate whether each material, system or method listed is relevant to your study. If you are not sure if a list item applies to your research, read the appropriate section before selecting a response.

## Materials & experimental systems

| n/a | Involved in the study |
|---|---|
| ☒ | ☐ Antibodies |
| ☒ | ☐ Eukaryotic cell lines |
| ☒ | ☐ Palaeontology and archaeology |
| ☒ | ☐ Animals and other organisms |
| ☒ | ☐ Clinical data |
| ☒ | ☐ Dual use research of concern |
| ☒ | ☐ Plants |

## Methods

| n/a | Involved in the study |
|---|---|
| ☒ | ☐ ChIP-seq |
| ☒ | ☐ Flow cytometry |
| ☒ | ☐ MRI-based neuroimaging |

## Plants

Seed stocks — *Report on the source of all seed stocks or other plant material used. If applicable, state the seed stock centre and catalogue number. If plant specimens were collected from the field, describe the collection location, date and sampling procedures.*

Novel plant genotypes — *Describe the methods by which all novel plant genotypes were produced. This includes those generated by transgenic approaches, gene editing, chemical/radiation-based mutagenesis and hybridization. For transgenic lines, describe the transformation method, the number of independent lines analyzed and the generation upon which experiments were performed. For gene-edited lines, describe the editor used, the endogenous sequence targeted for editing, the targeting guide RNA sequence (if applicable) and how the editor was applied.*

Authentication — *Describe any authentication procedures for each seed stock used or novel genotype generated. Describe any experiments used to assess the effect of a mutation and, where applicable, how potential secondary effects (e.g. second site T-DNA insertions, mosiacism, off-target gene editing) were examined.*

