## [Peer Review File · Nature Human Behaviour]

Skill Dependencies Uncover Nested Human Capital

Corresponding Author: Professor Hyejin Youn

This manuscript has been previously reviewed at another journal. This document only contains information relating to versions considered at Nature Human Behaviour.

Version 0:

Decision Letter:

23rd October 2023

Dear Professor Youn,

Thank you once again for your manuscript, entitled "Nested Skills in Labor Ecosystems: A Hidden Dimension of Human Capital", and for your patience during the peer review process.

Your Article has now been evaluated by 3 referees. You will see from their comments copied below that, although they find your work of potential interest, they have raised quite substantial concerns. In light of these comments, we cannot accept the manuscript for publication, but would be interested in considering a revised version if you are willing and able to fully address reviewer and editorial concerns.

We hope you will find the referees' comments useful as you decide how to proceed. If you wish to submit a substantially revised manuscript, please bear in mind that we will be reluctant to approach the referees again in the absence of major revisions. We are committed to providing a fair and constructive peer-review process. Do not hesitate to contact us if there are specific requests from the reviewers that you believe are technically impossible or unlikely to yield a meaningful outcome.

Finally, your revised manuscript must comply fully with our editorial policies and formatting requirements. Failure to do so will result in your manuscript being returned to you, which will delay its consideration.

If you wish to submit a suitably revised manuscript, we would hope to receive it within 4 months. I would be grateful if you could contact us as soon as possible if you foresee difficulties with meeting this target resubmission date.

- Include a "Response to the editors and reviewers" document detailing, point-by-point, how you addressed each editor and referee comment. If no action was taken to address a point, you must provide a compelling argument. When formatting this document, please respond to each reviewer comment individually, including the full text of the reviewer comment verbatim followed by your response to the individual point. This response will be used by the editors to evaluate your revision and sent back to the reviewers along with the revised manuscript.
- Highlight all changes made to your manuscript or provide us with a version that tracks changes.

Link Redacted

Thank you for the opportunity to review your work. Please do not hesitate to contact me if you have any questions or would like to discuss the required revisions further.

Sincerely,

[REDACTED]

REVIEWER COMMENTS:

Reviewer #1:

Remarks to the Author:

This article uses asymmetry in the conditional probabilities of the co-occurrences of skills within occupational descriptions in BLS occupational survey data to construct a hierarchical rendering of skills, where lower-level “basic” skills are seen as prerequisites for the development of higher-level skills. Across three validation exercises using cross-sectional data, synthetic panel data, and panel data from digital resumes, the authors show that older workers tend to work in occupations that require higher-level skills, that workers tend to shift to occupations with higher-level skills as they grow older, and that workers move into occupations demanding higher-order skills with each career transition. Occupation-level regressions show that occupations that require expertise in nested skills pay higher wages and require more education than unnested skills, and a further analysis suggests that wage and education premium associated with nested skills is driven by their interaction with fundamental basic knowledge skills on which they build. However, a potential (?) typo in the legend for Figure 4C makes it unclear which bars in the chart correspond to the wage and education premiums with and without the fundamental skill controls, and thus make it difficult to evaluate this final result. Breaking out the results by gender, race/ethnicity, geographical region, and time, the authors generate key insights that women deepen their skills less in nested areas at a slower rate than men as they grow older, that Hispanic/Latinx workers disproportionately have un-nested skills, and that greater expertise in nested skills (and less in un-nested ones) was demanded in 2019 than in 2005.

Overall, I am very intrigued by this paper. I believe that it makes a very unique and important theoretical contribution to our understanding of human capital and how it is generated over time, and they explore this theory using relevant and informative empirics. The paper does not introduce a new data set, but that is not essential as its novelty lies in the development of a new and important way of processing existing occupational data to learn from it more deeply. It is also evident that the authors have undertaken a lot of work to produce a very thorough investigation and a (nearly) polished paper. With these points in mind, I have several comments for the authors as they work to improve their paper.

My primary comment is that there is a mismatch between the paper’s theoretical setup and its empirical exercise. The authors frame their skill hierarchy as a reflection of the sequential order in which workers develop skills. However, the conditional probabilities that they use to develop the hierarchy are from occupational data. Therefore, the produced hierarchy reflects not the order in which skills are learned, but applied. The authors have a brief acknowledgement of this point, on line 107, where they say that it would be ideal to have data on individual skill acquisition, but they do not note that their data measure a very different process. Moreover, language throughout the text suggests that the hierarchy is constructed from learning sequence data; for example, line 66 says that “The prerequisites for understanding calculus, for instance, are grounded in the knowledge of algebra”. To be exact, the authors do not demonstrate that algebra is necessary to learn calculus, but that algebra is necessary to apply calculus in more advanced occupations.

Ultimately, I do not think the use of occupational (skills in application) instead of learning (skills in development) data sinks the paper. I think there is still a ton we can learn from this paper and the sequential order through which workers apply skills. In particular, the later analyses showing that workers move into higher levels of the skill hierarchy over the course of their careers is very good, and strongly supports their arguments. I see these analyses as a sort of validation test, and the results are encouraging. However, the authors need to be much more upfront about what raw input data measure, and what they can say with it. The language of the paper needs to make it clear that they identify the sequential order in which workers learn, but the sequential order in which workers apply their knowledge (They do not show that algebra is a prereq to learn calculus, but that algebra is a prereq to use calculus in useful settings). To make this correction, the paper needs a subtle but crucial theoretical reframing.

More generally, the exposition of the paper can be improved. In particular, the description of the methods to produce Figures 2A and 2B are not clear. As I understand it, you have a bipartite graph that links skills to occupations. You infer directed edges between the skills based on the asymmetrical conditional probabilities between those skills, using subtraction. Thus, you end up with a directed graph that contains only one type of node, representing skills. This language and a figure to depict it would be helpful. I would also want to know (and see depicted) the idea that there is an edge recording the weight of the difference flowing between the skills. (In the programming and math example, I think the weight is 0.32?)

A related point on exposition of methods is that it is quite opaque how skill levels are used to measure the “skills demanded by each occupation”. My understanding is that the raw data lists an occupation, the skills that are required to perform the occupation, and a skill level for each of those skills, with low scores indicating basic proficiency and high scores indicating expertise. The authors leverage these skill levels as indicators of the extent to which each skill is “demanded” by each occupation. Thus, the authors use these “quality” quality to measure the “quantity” of each skill demanded by each occupation.

I thus think that the authors are conflating skill quality with skill quantity in their use of this data. As before, I do not think that this issue sinks the paper, but again it needs to be described thoroughly and I would like to see the theoretical implications discussed.

A final point on exposition is that I could not figure out, from the main text and the supplement, how the synthetic panel is constructed.

In terms of materials, I have a few comments on the first results section, beginning on Line 102. My first comment is whether Figure 1A is needed? Case examples are embedded in the top of Figure 1B already. My second comment is a question: is this first results section necessary? As far as I can tell, the authors do not use these measures of skill generality in the subsequent analyses. They are introduced almost to demonstrate the existing approach to conceptualizing the breadth of skills, a measure on which the authors aim to improve using their hierarchical approach. If the Skill Generality section is to remain in the paper, I suggest that the authors present it as a way to contrast to the hierarchical approach that they later develop. Moreover, it lends me to wonder to what extent the hierarchical approach improves on the skill generality/ubiquity framework. In this regard, it would be helpful to see a scatterplot of skills by generality (measured as per Figure 1) and nestedness (measured as per Figure 2). This would allow readers to evaluate the value added of the hierarchy.

In some areas, the paper is in a position to contribute more deeply to existing literatures than it currently does. I have a number of suggestions here. First, the cities analysis has a lot of potential, but is currently underdeveloped. To some extent, I think that the authors could withhold their analysis of the geography of nested skills from this paper, and write a separate paper about how their nestedness framework helps to weigh in on the diversity vs. specialization debate in urban economics and economic geography – that debate is still stuck on defining and measuring specialization and diversity, and I believe that the skill hierarchy produced in this paper would lend a lot of clarity to that literature. As it currently stands, the cities analysis currently confirms much of what urbanists already know about the skill composition of cities, and raises (but does not answer) a handful of interesting questions about cities, such as whether the high level of general skills in big cities is driven by a deep local division of labor, or that the professionals in the tradable sectors in big cities also have developed foundational, general skills. Moreover, I was curious to what extent the big-city results were driven by within-individual or across-individual variation. I would like to see a decomposition by cities based on their raw material / natural resource extraction. This has a nice feature because natural resources are the commodities that generally require the least human capital input to enhance their value. Again, I think it would take a full-length paper to bring out all of the contributions that this paper can make to our understanding of skill compositions in cities.

The demographic analyses (gender, race, ethnicity) are very interesting, and without individual level on these attributes (they are crosswalked by occupation), I think it makes sense to keep these analyses within the current paper (for example, the authors do not have individual level data on women workers, so they cannot test whether the slow adoption of nested skills by women beyond age 30 is associated with shifting to motherhood).

In the demographic analyses, the gap between the skill level for men and women in unnested occupations was really remarkable (Figure 3F). This makes sense to me, in that we often think of applied/mechanical jobs that are high-paying to be “mens’ work” (think of oil well operators or deep sea fisherman). But I would like to see more discussion and analysis of this finding. I am also interested in seeing the authors interpret the higher attainment of Hispanic/Latinx workers in unnested skills. Is there a tradeoff for unnested skills, in that they can be attained relatively quickly and at lower cost, but offer fewer avenues toward entering high-income occupations; moreover, by being involved in primarily unnested skills, are Latinx and many male workers stuck in a sort of “middle income trap”, where they could reach moderate income levels relatively easily but have not built the foundational skills needed to achieve high income?

Rhetorically, I found the introduction to not fit the paper exactly. I appreciated the writing on the historical growth of complexity of the economy over time, but the actual lead of the article does not appear until line 62, when the authors (finally) state that “certain skills can be gained without any prerequisites . . . while other skills are not immediately attainable”. This is a great sentence and a good point, should be featured in the first paragraph. Moreover, first two paragraphs of the introduction currently present the paper as if it is about growth, but it is not.

Finally, in attempting to get a sense of the extent of the contribution, I wondered if we can distinguish, using your framework, between the skills like Fine Art and Programming. I raise these skills because they are both are grouped as “nested skills” because they draw on a large range of more basic fundamental skills. But wages are strikingly different across these two skills. Ultimately, human capital theory is intended to explain things like productivity and wages. How can your framework help us to understand why programmers are paid more than people with expertise in the fine arts? Rhetorically speaking, what is your model’s R-squared in predicting wages?

-- Smaller points --

In the beginning of the results section, I found the language of how occupations “demand” skills to be confusing. For me, this evoked the concept of demand in the economic sense, so my mind immediately started to think about a market for skills. This is not what you are describing --you’re just describing skill requirements in occupations – so I suggest simplifying the language.

Some of the core writing of the intro is unclear. For example, the final sentence of the intro (line 100) refers to a widening gap between the two, but I do not know what “the two” refers to.

For median ages for occupations, the figure should say “the median age of workers in each occupation”. The current text suggests to me that you might be counting the numbers of years elapsed since the BLS introduced each occupation code to its classification system.

Why in the maps of skills by county (Figure 6 and in the supplement) do there appear to be state-level effects? For example, Wyoming is almost all the same shade of blue, as is Nevada? Is the occupational info classified uniquely by each state? I am trying to figure out why there would be any state-level grouping, given that the economies are porous between adjacent counties across state lines.

Reviewer #2:

Remarks to the Author:

This paper presents a detailed study of how different occupational skills precede and depend on one another within a network of dependencies. Its main data source is a database of occupations and the skills that each needs. By observing the conditional probability for an occupation to require some skill A, given that it requires skill B, the paper infers a network of skill dependencies showing which skills (e.g. "Mathematics") tend to precede which others (e.g. "Programming"). From this network and other analyses, the paper present findings about skill acquisition and its relationship to variables such as wages and the geographic distribution of employment.

The strong positives of this paper are that it offers new ways of thinking about and analyzing human capital that are potentially

richer than traditional conceptualizations, and that it presents intriguing findings on the nature of skills that appear to be enabled by this new approach. The results and methods are of interest to a broad array of researchers.

Some key claims include the following:

- Many skills are "nested". These skills come from a tree-like portion of the skill network that extends from root nodes (which can be understood as foundational skills) to leaf nodes (which represent specialized skills).
- Other skills fall in an "un-nested" portion of the network. Such skills (e.g. "Repairing") do not require foundational skills (to the same degree?) as the specialized skills in the nested portion.
- The analyses focuses on differences between these 3 large categories of skills above: (1) the un-nested skills, and the 2 types of nested skills: (2)"general" skills (which I call here "foundational" skills), and (3) "specific" or specialized skills.
- Over the course of a career, workers use more of *both* foundational skills and specialized skills, and less of un-nested skills. As the authors note, the increase in foundational skills is interesting because it suggests that these skills need to be "continuously enhanced regardless of how advanced we are in our career journeys". (Also interestingly, the authors find that this increased used of foundational skills is not driven by workers' tendency to move into management jobs over time, nor is it specific to those foundational skills that could be considered "social skills". Evidently foundational skills in general get enhanced as workers progress.)
- One dramatic result is the correlation with wages: Without controlling for other variables, wages rise with an occupation's skill level in specialized skills, but not with skill level in un-nested skills. However, this difference largely vanishes when an occupation's level of foundational skills is taken into account. Apparently, without strengthening foundational skills, higher level of ability in specialized skills doesn't yield higher wages.
- Economic demand may be increasing over time for general skills over specialized and un-nested skills.

The results and methods are interesting and the manuscript warrants publication in a revised form.

My two main concerns are below.

(1) Analysis of un-nested skills / focus on nestedness

I feel unsatisfied by the analysis of un-nested skills, and more generally by the strong focus on the concept of "nestedness". From what I can tell, "un-nested" nodes are not un-nested - they are less deeply nested, with a different set of root nodes/foundational skills. The authors have "backboned" the network in Figure 2, i.e. what appears there drops many links for the sake of visualization. This is not necessarily a problem, but after seeing the detailed representations of this network in the SI, I am not sure why they do this, and feel the one in the main text leaves out useful information. Fig S21-S22, for example, make it clear that the nested skills are overwhelmingly non-physical ones (e.g. thinking, social interaction) and that un-nested skills overwhelmingly emphasize physical activities (dexterity, depth perception, hearing sensitivity, arm-hand steadiness). Among the un-nested skills, one can see plausible nests / sequences of skills that build on one another, from general skills to specific ones, e.g. Visualization -> Perceptual Speed -> Quality Control Analysis -> Equipment selection. The data seems to distinguish many fewer skills in this "physical-and-related" category than in the "non-physical" category. Don't the "un-nested" skills just represent a different nest, for physical and related skills? (And that it is shallower than the other nest, because there are just fewer skills to construct dependencies between?)

The labels "nested" and "un-nested" themselves are a bit abstract and it matters for interpreting how novel the results are. For example, one finding is about how cities with different levels of manufacturing intensity differ in skill emphasis:

"Upon grouping cities by manufacturing employment... we find that cities highly specialized in manufacturing tend to exhibit lower levels of nested specialization but higher level of un-nested specializations."

Written this way, the statement sounds more novel than if one had said

"Upon grouping cities by manufacturing employment... we find that cities highly specialized in manufacturing tend to exhibit lower emphasis on cognitive skills but higher emphasis on physical skills."

I do not mean to convey that the emphasis on nestedness is unjustified and that findings like the one above are not interesting or novel. It's rather that I find I'm left struggling to decide on my own whether the focus on nestedness teaches me something new (or an interesting new angle on something old). It would be great for the paper to provide more guidance.

From what I can tell, the results just hinge on there being "dependencies" between skills - to do "negotiation", a worker needs a sufficiently strong basis in a general skill such as "oral expression". Why the strong focus on "nestedness"? This is a specific phenomenon where observational units - a company or a location say - have portfolios of activities that are subsets of one another. Diversified companies not only make more products than un-diversified ones; they make supersets of the portfolios of un-diversified companies. As the authors note, this phenomenon is seen in ecosystems, e.g. a diverse continental ecosystem may contain all the species of less diverse island ecosystems, plus others. The text and the title call out this ecosystem analogy. But it is only mildly developed in the paper, and I'm ultimately not sure what the significance of nestedness is here over the simpler idea that some skills depend on other skills.

On a separate note: In Figure 4c, after controlling for the level of general skills, the slope flips and wages rise with the level of un-nested skills. I am just curious whether the authors think this clashes with the idea that un-nested skills do not depend on general skills.

This paper has a number of methods and findings worth communicating to the audience of this journal, and I do not feel all questions of interpretation should or can be resolved here. But I would like in the main text a better discussion of why "nestedness" is important and a more complete / nuanced discussion about the un-nested skills.

(2) Presentation

Overall I was not thrilled with how this material has been presented. The paper contains a lot of passages and phrases that seems needlessly complex or vague:

- "Modern economies... operate through globally interconnect networks. As economies become more complex, so do these networks..." (Are we discussing social networks? Trade networks? What are these sentences ultimately telling me? Could they be reduced to just "As economies become more complex, they coordinate more diverse portfolios of knowledge"?)
 - "Nevertheless, the role of foundational skills for such ascent remains pivotal; without reinforcing them, the anticipated wage premiums may vanish." (Is this referring to a prior result, or are the authors referring to their own results on wages?)
 - "By differentiating general human capital into a structured spectrum, from the most foundational and general to the most specialized, human capital are comparable at different scales of organizations, which is essential for policy implications." (What does "comparable at different scales of organizations" mean?)
 - "The hierarchical structure and its inherent directionalities add a new dimension to the rising field of economic complexity, providing a deeper understanding of how knowledge is accumulated within a population and how it is expressed in the economic activities of a firm, city, region, or country." (This paper is not about adding "hierarchy" and "directionality" in general to economic complexity. This feels over-wrought. The paragraph it appears within generally seems like it could be reduced to a sentence or two.)
 - The comparison to ecosystems is possibly interesting but again too vague for me to gain anything from it.
- These examples focus on individually confusing sentences but the problem is more pervasive; the text could use a work over. The abstract is particularly convoluted. I spent much more time parsing the paper than should be necessary. The authors have interesting results and they could present them much more simply.

The paper comes with a huge SI - 90 pages - that is difficult to map to the main text. References to the SI are a bit chaotic; SI sections/figures/tables are not arranged in an obvious way (e.g. sequentially) with where these components are needed, and quite a number of figures do not seem to be referenced at all. The SI doesn't compensate with its own separate intro or summary to offer guidance. The result is that it is very hard for anyone to practically review all that has been done, or for interested specialists to find things they are curious about. It also triggers the worry that some SI analyses are really essential for the core arguments and may be more appropriate for the Methods section of the main text.

It is a little hard to find information about the datasets. It is there but dispersed. It would help to have a clearly-labeled places in the main text where the 3 datasets / data-derived objects are covered:

- The database of occupations based on US BLS surveys 2005-2019.
- The pseudo data of cohorts constructed with data from the Current Population Survey.
- The corpus of resumes from Burning Glass (70 million job transitions).

Reviewer #3:

Remarks to the Author:

Peer Review for "Nested Skills in Labor Ecosystems: A Hidden Dimension of Human Capital" in Nature Human Behaviour

The study presented in this paper is intriguing and utilizes a unique dataset of 70 million CVs and career transitions. The methodological approach employed in the research is well-developed, and the derivation of relevant metrics is executed with precision. Furthermore, the concept of nested and un-nested skills is a novel and insightful perspective on human capital, and the organization of skills in a directed network based on their pair-wise dependencies is a powerful approach with significant implications for this and future investigations.

However, I am compelled to advocate for a major revision before this work can be considered for publication. My primary criticism is the absence of a clear theoretical research question that is addressed by the findings of this study. The results explore various distributional issues related to skills across geographical locations, gender disparities, and life trajectories, but none of these insights seem to be driven by a theory-based research question. The paper reads more like a methodologically-savvy data expedition, covering diverse societal angles without offering a conclusive framework to close gaps in existing theory.

One potential way to address this deficiency would be to develop a research question grounded in theory. For instance, the paper could investigate whether earnings differentials between men and women can be explained by systematic differences in the distribution of nested versus un-nested skills. The data presented in Figure 3 could prompt an exploration into whether women experience disruptions in the accumulation of nested skills, possibly due to motherhood, and whether these imbalances subsequently contribute to wage differences. This would not only provide a stronger theoretical foundation but also enhance the relevance and impact of the research.

As a minor point, I noted that some findings related to skill nestedness and wages appear to overlap with a previous study (<https://doi.org/10.101/j.respol.2023.104898>) that is not cited in the present work.

In conclusion, while "Nested Skills in Labor Ecosystems: A Hidden Dimension of Human Capital" is an engaging and methodologically sound study with novel concepts and an impressive dataset, it is crucial to address the lack of a clear theoretical research question. The addition of a well-defined theoretical framework and the exploration of questions such as gender disparities in skill accumulation could significantly strengthen this work.

Version 1:

Decision Letter:

Our ref: NATHUMBEHAV-23082510A

18th June 2024

Dear Dr Youn,

Thank you for submitting your revised manuscript "Nested Skills in Labor Ecosystems: A Hidden Dimension of Human Capital" (NATHUMBEHAV-23082510A). It has now been seen by the original referees and their comments are below. As you can see, the reviewers find that the paper has improved in revision. We will therefore be happy in principle to publish it in Nature Human Behaviour, pending remaining revisions to satisfy the referees' final requests and to comply with our editorial and formatting guidelines.

We are now performing detailed checks on your paper and will send you a checklist detailing our editorial and formatting requirements within two weeks. Please do not upload the final materials and make any revisions until you receive this additional information from us.

Sincerely,

[REDACTED]

Reviewer #1 (Remarks to the Author):

I am deeply impressed by the efforts the authors have taken to respond to my comments and improve their manuscript. The paper is now much more legible and its contributions are more clear. However, I have a few minor outstanding comments:

1. The complexity framing still does not work for me. The contribution of the paper is that it shows that the value of specialized knowledge is contingent on the existence of foundational knowledge. However, the first two sentences of the abstract and the first two paragraphs of the main text are about another topic entirely. I suggest these 2 sentences and these 2 paragraphs should be stricken. The intro section is excellent from the third paragraph onward, and is sufficient to begin the article. I would make a similar comment about the second paragraph of the Discussion section. That paragraph distracts from the actual contributions to of the paper to human capital theory (which I think are very substantial!)
2. The text in the discussion section about the issue with observing the order skills are performed, as opposed to the order they are learned, is very good. However, the text on this issue in the main section introducing this method is still not clear. To reiterate, my main concern is that the order of learning skills may differ from the order that they are applied. Your text in the discussion I think addresses this issue, but I would like to see it succinctly summarized in the section introducing the method.
3. Figure 1, I think the figure should be bumped back a page. Also, the headers for Figure 1A are confusing: there are specific skills, general skills, and "intermediate" skills, but it is unclear what intermediate refers to. How about High Specificity, Moderate Specificity, and Low Specificity? That would allow you to strike the term "General" from the paper, reducing the number of concepts your readers have to remember.
4. Page 8, the term "foundational" in the sentence "Skills with a high nestedness contribution (*c.s*) are foundational to . . ." is confusing, because foundational/bedrock knowledge is used to refer to the general skills at the bottom of these skill pyramids, but now it is being used to refer to the skills at the top of the pyramids. Also, in the discussion, some terms ("skills like Programming exhibit a positive impact on nestedness", imply causation, but you have no way to determine whether nested skills emerged on their own (and those drove the increase in nestedness), or were anticipated by the existence of the lower-level more fundamental skills.
5. I like the new Figure 3, but it seemed a bit out of order and that it broke down the nicely-flowing organization of the manuscript. I wonder if it can be moved around or if the sections organization can be redesigned.
6. With regard to the additional analyses you performed in response to my earlier comments, the results on all of these are very interesting, especially with regard to family dynamics (Figure S49 and Table S7) and race/ethnicity (Figure S18). Thank you for engaging so thoroughly with my comments!

Reviewer #2 (Remarks to the Author):

Overall, I am happy about the changes in the new version. My concerns about presentation have largely been addressed. The new text is clearer, and better explains the logic behind the various analyses. The Methods and SI gained helpful text to understand the datasets and navigate what is a large SI.

On the analysis of un-nested skills / focus on nestedness, the authors have also responded in a couple ways to concerns I raised. They argue that the labels "nested" and "un-nested", while a bit of a simplification, are useful expositionally, and derive from a clear quantitative criterion in the paper. In addition, they acknowledge that "nested" and "un-nested" skills closely correlate with cognitive and physical skills, which for me raised the question whether we learn something new from statements like (for example) un-nested

skills are associated with lower wages. The authors note that "cognitive" and "physical" skills are arbitrary categories applied ex post. In contrast, the network quantities they observe are completely derived from the skill dependency network. I agree on all the above points.

In general, I remained convinced that the paper's results are fundamentally sound and represent an important contribution.

I have a small technical concern from text that appears to be new. In addition, I still have concerns about the discussion of nestedness. Many of these are about how this concept is discussed in the paper versus ecology and economic complexity, fields this paper is connecting itself with. More generally, I am also still trying to understand why nestedness and "ecosystems" are used in framing this paper at all.

****Independence of skills A and B****

At the top of page 6, the authors discuss what it means for the presence of two different skills in an occupation to be independent events. The text says if skill A is independent of skill B, we should see

$$p(\text{skill_A} \mid \text{skill_B}) = p(\text{skill_A}) p(\text{skill_B}).$$

But this seems to confuse conditional and joint probabilities. Presumably, the argument needed here is

$$p(\text{skill_A} \mid \text{skill_B}) = p(\text{skill_A}).$$

If I am correct then this worries me (slightly) how this assumption might be entering the analysis.

*****"Nestedness" and connections to ecology, economic complexity****

I greatly appreciate the effort the authors put into responding on these issues. However, in spite of significant changes, I am still asking myself, "Why frame these results in terms of nestedness and ecosystems at all? Aren't these results just about 'dependencies' between skills and how they matter for career and economic outcomes?"

It's evident from lived experience that some skills serve as prerequisite foundation for others. Yet (as the authors rightly hone in on) there is nowhere one can look for a comprehensive quantitative description of these dependencies. The paper offers this, making many interesting observations about both the network structure itself, and crucial correlations with economic variables and career trajectories.

Now, somehow, these interesting observations get entangled with an ecosystem analogy. This begins from the title, and is repeated throughout, explicitly, and is also reflected in the focus on the phenomenon of "nestedness" (which also occurs in ecology). The analogy originally struck me as a connection that could readily be dropped. Yet the new draft and the author responses seem to elevate its importance. In addition, there are also connections to the field of economic complexity that somehow a part of all this.

I think the connections to these fields are confusing. I first review what I think I know about other literatures (on which the authors should freely correct me). In ecology, nestedness is a property of a set of ecosystems. It can be computed from a matrix that has the dimensions

species x places [ecology].

"Nestedness" here is the observation that less diverse ecosystems are subsets of ("nested within") more diverse ones. So, large island "A" has all the (generalist) species of small island "B", plus some extra specialist species.

In economic complexity, the corresponding matrices have dimensions of economic activities x places. The current paper deals with "occupations", so let's take these as the activities:

occupations x places [economics]

"Nestedness" also comes up in economic complexity, where it has a nearly identical meaning, as the observation that less-diverse places not only have fewer occupations (by definition), but the occupations they have are subsets of those of more diverse places.

In the current paper, the relevant matrix has these dimensions:

skills x occupations [this paper].

Here, nestedness is posed as a property of a skill. So, a more nested skill here is one that is deeper along the branches that stem from root, foundational skills.

In all these contexts, there is discussion about possible mechanisms underlying the nestedness. It sounds like ecologists have a mechanism that involves mutualistic interactions of generalist species with specialists. In economic complexity, it's often thought that nestedness results from economic "capabilities". I.e. some activities need more capabilities and some places have more capabilities, and nestedness of economies occurs (it's thought) because having more capabilities means an economy can do any of the things less complex economies can do plus more. In this paper, the proposed mechanism is not quite fleshed out (nor needs to be as far as I'm concerned) but the authors have in mind that making complex things means combining many skills, but cognitive limits constrain how many skills one human can learn, requiring us to develop careers in which we specialized in particular skill sets.

I find the discussion about ecosystems, economic complexity, and nestedness confusing for a few reasons:

- First, what exactly is the analogy to ecosystems? A "labor ecosystem" at first blush sounds like occupations are the "species". But the paper instead takes skills the "species"; it makes this correspondence:

skill <-> species
occupation <-> place

In the usual analogy between economic complexity and ecology, we have occupations as the species:

occupation <-> species
place <-> place

The usual analogy is natural and "nestedness" has essentially the same meaning in both economic complexity as in ecology. (I.e. not only are some places less occupationally diverse, but the occupations they do have are subsets of the occupations of more diverse economies.) The analogy of the paper feels odd or forced, and it seemingly leads to the issues below.

- "Nestedness" clashes with its meaning elsewhere. If skills are the "species", then given how nestedness is discussed in ecology and economic complexity, nestedness should be a property of an occupation, not a skill. (Or really, a collection of occupations.) I.e. skills should not be the thing that's more or less nested but occupations, with some occupations requiring narrow skill sets, which are subsets of the skills of occupations requiring broad skill sets. (To some extent this issue gets confusing because many metrics of the nestedness of a matrix give the same score whether applied to a matrix or its transpose.)

- Discussion of mechanisms: The paper says nestedness of skills is akin to ecological interactions where specialist species interact mutualistically with generalist species. This comment gets mentioned several times, but it's unclear why the paper is emphasizing this. Is this just reiterating something ecologists think about the cause of nestedness in ecology? If so, why do we care? Is the implication that an analogous mechanism could underly skills and occupations? Is this fleshing out the analogy itself? Is it just a random tidbit?

At the same, economic complexity has another explanation of nestedness that readers might naturally try to apply to this paper. (The capabilities story above.) But again, it is unclear what mapping the authors have in mind here, or what significance it would have. Also, note that, in the economic complexity discussion of nestedness (instead of ecology), one deals with the extra concept of "capabilities". I.e. we don't just have the bipartite network of places and the "species" that express the makeup of these places, but a tripartite network where the expression is mediated by capabilities. Nestedness is something that has been observed in the expression of activities by different places (i.e. in the sectors or products), but conceptually, "skills" correspond much better to "capabilities". And connecting "skills" to "capabilities" (instead of to economic sectors) makes all the more sense here given the authors' point that diverse skills need to be combined to make complex goods, while each human is limited in how many skills they can learn, leading to career specialization.

Making analogies is in principle fine, but the way it is handled currently will cause confusion. I am not quite sure what to suggest, because, while I don't see how these connections as necessary (given the importance and strength of the findings), the authors may regard these connections as important. I ultimately find myself back to my original question- should these results just be discussed as an investigation of inter-dependencies among skills? "Nestedness" and the ecosystem analogy, while interesting topics, don't seem like necessary connections to make the findings important.

****Minor comments****

- Page 8, "Figure 2 (d-f)": No panel f in figure 2.
- Page 9: "translated"  "translate"
- Pages 18-19: "Future research could benefit from surveys targeting employees.... etc." Repeated text with next paragraph.
- Page 21: Reference to Fig. 2 (b-f) but no panel f.

Reviewer #4 (Remarks to the Author):

The paper has advanced quite a bit from my review, and I think it is well on its way to publication in Nature Human Behavior, congratulations to the authors on these changes.

I have two important remaining notes:

1. The biggest one, which I don't know exactly how to say, is that while the authors have answered all my comments directly, the paper is throughout still hard to follow.
 - a. This issue stems from something in the writing that I cannot quite state: it is too wordy, too many long sentences, and too flowery, rather than the clear and concise language that make for good academic writing. Each sentence is long, complex, and jargony, rather than short and to the point.
 - b. The most egregious example of this are the extremely long responses to my review. I found the document very hard to follow.
 - c. And, while the reviewer document is less important, the issues also show up in the paper.

For example, the paper starts with the following sentences:

"Complexity and specialization are foundational to the narrative of economic growth and innovation [3{6}. As society advances, creating and maintaining sophisticated goods, services, and infrastructure, these socio-economic complexities have surpassed what individuals can embody and manage on their own [7, 8]. It is no longer feasible for individuals to master universal expertise across all areas. For economies, this means developing deep divisions of labor and knowledge that first distribute knowledge

across people and then coordinate this distributed knowledge in teams, firms, and value chains [9{12}].”

For an introductory paragraph, is this really jargon heavy and hard to read.

A simpler version of basically the same could be:

Complexity and specialization are central to economic growth and innovation [3{6}. As society advances, socio-economic complexities have surpassed what individuals can embody and manage on their own [7, 8]. Because individuals cannot master expertise across all areas, developing deep divisions of labor and knowledge that both distribute and coordinate this knowledge is important for all economies [9{12}].”

I am not a good writer, but I do think the above is better. To me, the first order of business is getting this paper to be much simpler, potentially working with a copy editor.

2. A big area where this could improve is on the explanation of nestedness. I finally learned clearly what it is from a simple Wikipedia page. I think just slightly more clarity on writing this section will make a big difference.

Great work overall. I hope this helps.

Version 2:

Decision Letter:

Dear Hyejin,

We are pleased to inform you that your Article "Skill Dependencies Uncover Nested Human Capital", has now been accepted for publication in *Nature Human Behaviour*.

Please note that *Nature Human Behaviour* is a Transformative Journal (TJ). Authors may publish their research with us through the traditional subscription access route or make their paper immediately open access through payment of an article-processing charge (APC). Authors will not be required to make a final decision about access to their article until it has been accepted. [Find out more about Transformative Journals](https://www.springernature.com/gp/open-research/transformative-journals)

We welcome the submission of potential cover material (including a short caption of around 40 words) related to your manuscript; suggestions should be sent to *Nature Human Behaviour* as electronic files (the image should be 300 dpi at 210 x 297 mm in either TIFF or JPEG format). Please note that such pictures should be selected more for their aesthetic appeal than for their scientific content, and that colour images work better than black and white or grayscale images. Please do not try to design a cover with the *Nature Human Behaviour* logo etc., and please do not submit composites of images related to your work. I am sure you will understand that we cannot make any promise as to whether any of your suggestions might be selected for the cover of the journal.

To assist our authors in disseminating their research to the broader community, our SharedIt initiative provides you with a unique

shareable link that will allow anyone (with or without a subscription) to read the published article. Recipients of the link with a subscription will also be able to download and print the PDF.

With best regards,
[REDACTED]

P.S. Click on the following link if you would like to recommend Nature Human Behaviour to your librarian
<http://www.nature.com/subscriptions/recommend.html#forms>

** Visit the Springer Nature Editorial and Publishing website at http://editorial-jobs.springernature.com?utm_source=ejp_NHumB_email&utm_medium=ejp_NHumB_email&utm_campaign=ejp_NHumB for more information about our career opportunities. If you have any questions please click [here](mailto:editorial.publishing.jobs@springernature.com).

Dear Reviewers:

Thank you so much for the feedback and the opportunity to improve our paper. We are lucky to receive comments that are constructive and instrumental in improving our manuscript. Below we include (1) revision summary, (2) summary of previous reviews, and (3) point-by-point response for each reviewer.

Previous reviews recap:

1. Editorial comment: [considerable] potential interest, but there are quite substantial concerns to be addressed.
2. Reviewers comments:
 - a. strength: new methods/framework (additional dimension) & technically sounds;
 - b. weakness: theoretical framing, empirical interpretations, theoretical contribution and articulation are needed, especially centered around nestedness and labor ecosystems.

Revision summary:

1. We entirely rewrote the paper: introduction and results to articulate our methods and logic.
2. In particular, we created an additional section (Sec 3) with a new figure of four panels (Fig 3) to strengthen our logic and measures. Instead, we remove the geography section in the main manuscript in order to sharpen and focus our empirical analysis and methodological contributions. The previous analysis is included in SI for the reference for follow-up researchers.
3. We provide additional case studies and additional analysis of (Fig 5) to flesh out the disparity and justify/validate our use of nestedness score and tree structures in human capital.
4. We fleshed out our conceptualization of nestedness in the context of human capital.
5. We added our theoretical contribution throughout the manuscript with more traditional references

Below are point-by-point responses for each reviewer. Once again, thank you so much for your constructive and instrumental suggestions.

1. Reviewer #1

This article uses asymmetry in the conditional probabilities of the co-occurrences of skills within occupational descriptions in BLS occupational survey data to construct a hierarchical rendering of skills, where lower-level “basic” skills are seen as prerequisites for the development of higher-level skills. Across three validation exercises using cross-sectional data, synthetic panel data, and panel data from digital resumes, the authors show that older workers tend to work in occupations that require higher-level skills, that workers tend to shift to occupations with higher-level skills as they grow older, and that workers move into occupations demanding higher-order skills with each career transition. Occupation-level regressions show that occupations that require expertise in nested skills pay higher wages and require more education than unnested skills, and a further analysis suggests that wage and education premium associated with nested skills is driven by their interaction with fundamental basic knowledge skills on which they build. However, a potential (?) typo in the legend for Figure 4C makes it unclear which bars in the chart correspond to the wage and education premiums with and without the fundamental skill controls, and thus make it difficult to evaluate this final result. Breaking out the results by gender, race/ethnicity, geographical region, and time, the authors generate key insights that women deepen their skills less in nested areas at a slower rate than men as they grow older, that Hispanic/Latinx workers disproportionately have un-nested skills, and that greater expertise in nested skills (and less in un-nested ones) was demanded in 2019 than in 2005.

Overall, I am very intrigued by this paper. I believe that it makes a very unique and important theoretical contribution to our understanding of human capital and how it is generated over time, and they explore this theory using relevant and informative empirics. The paper does not introduce a new data set, but that is not essential as its novelty lies in the development of a new and important way of processing existing occupational data to learn from it more deeply. It is also evident that the authors have undertaken a lot work to produce a very thorough investigation and a (nearly) polished paper. With these points in mind, I have several comments for the authors as they work to improve their paper.

Thank you for dedicating your time to providing such a comprehensive summary and constructive feedback on our manuscript. We greatly appreciate your efforts in reviewing our work and acknowledging the merits of our research. We are particularly grateful for your recognition that the novelty of our work lies in the framework rather than the data. Our aim was to make the most out of publicly accessible data instead of relying on proprietary or novel datasets (although we did use resume data to test our theory we tried not to rely on this dataset in our findings). It is encouraging to receive positive feedback on this aspect of our work, as not everyone appreciates this effort.

Furthermore, we want to express our sincere gratitude for your valuable suggestions and constructive criticism. As you will see in the detailed responses below, your input has been instrumental in helping us clarify our thoughts and improve the overall quality of the manuscript.

Regarding the mislabeling issue you pointed out, we deeply apologize, and are even embarrassed, for the confusion caused by our error. Upon reviewing Figs. 4C and 4D, we realized that the legends were indeed mislabeled. The solid blue bars should have been labeled as "Unconditional" instead of "Controlled," and the shaded bars should have been labeled as "Conditional" instead of "the uncontrolled." We sincerely apologize for any confusion this error may have caused.

We have now rectified this issue in both figures, ensuring that the labels accurately reflect the intended meaning. The blue bars are now correctly labeled as "Unconditional," and the shaded bars are labeled as "Conditional." We hope that this correction will provide clarity and eliminate any potential misinterpretation of our results.

Fig. 1.1.1. Updated Figure 4 with legends for parts (c) and (d) correctly labeling the blue bars "Unconditional" and the shaded bars "Conditional."

1.1. My primary comment is that there is a mismatch between the paper's theoretical setup and its empirical exercise. The authors frame their skill hierarchy as a reflection of the sequential order in which workers develop skills. However, the conditional probabilities that they use to develop the hierarchy are from occupational data. Therefore, the produced hierarchy reflects not the order in which skills are learned, but applied. The authors have a brief acknowledgement of this point, on line 107, where they say that it would be ideal to have data on individual skill acquisition, but they do not note that their data measure a very different process. Moreover, language throughout the text suggests that the hierarchy is constructed from learning sequence data; for example, line 66 says that "The prerequisites for understanding calculus, for instance, are grounded in the knowledge of algebra". To be exact, the authors do not demonstrate that algebra is necessary to learn calculus, but that algebra is necessary to apply calculus in more advanced occupations.

Ultimately, I do not think the use of occupational (skills in application) instead of learning (skills in development) data sinks the paper. I think there is still a ton we can learn from this paper and the sequential order through which workers apply skills. In particular, the later analyses showing that workers move into higher levels of the skill hierarchy over the course of their careers is very good, and strongly supports their arguments. I see these analyses as a sort of validation test, and

the results are encouraging. However, the authors need to be much more upfront about what raw input data measure, and what they can say with it. The language of the paper needs to make it clear that they identify the sequential order in which workers learn, but the sequential order in which workers apply their knowledge (They do not show that algebra is a prereq to learn calculus, but that algebra is a prereq to use calculus in useful settings). To make this correction, the paper needs a subtle but crucial theoretical reframing.

Thank you so much for your insightful comments and excellent suggestions. We acknowledge that due to the lack of ideal longitudinal data, we cannot directly observe individual workers' skill acquisition sequences. Instead, we make assumptions and support them with two synthesized longitudinal datasets and one direct observation from resume data. Even then, we recognize that we cannot definitively distinguish between the application and acquisition of skills unless we were able to ask individual workers about their actual state of knowledge. Therefore, what's happening at the individual worker level is inferred rather than directly measured in the current study because our empirical evidence is based on occupational attributes. This introduces a mismatch between what we observe and what we theorize, but we try to fill the gap by theorizing our assumptions with the available data. This is so because we believe that exploring this direction is crucial to shift research attention towards thinking in terms of conditional probabilities laid upon individuals as possible choices. These conditional probabilities come from the structure of society that grows with complexity and specializations, and this structure shapes our job polarizations and labor disparity. We would like to push this direction of the framework as long as it is theoretically plausible. To support this effort, we even conducted a preliminary survey with employees, finding results that align with our premise. However, incorporating a comprehensive survey into an already detailed paper would exceed our current scope due to the distinct methodologies and complexities associated with human subjects research. Thus, we plan to pursue a more thorough survey in future research to provide stronger empirical evidence.

Instead, based on your suggestions, we have focused on clearly describing and explaining our assumptions, predictions, and the type of data that can support our findings in the future. More specifically, in response to your comments, we have made the following changes: (1) being upfront about and articulating our rationale for using cross-sectional data as a proxy for longitudinal data and thus indirectly inferring learning from skill application. This explanation is now prominently featured in the introduction and relevant sections. (2) providing additional empirical evidence to support our assumptions and reasoning. This includes incorporating new analyses and strengthening the existing ones to support our case. (3) explicitly acknowledging the limitations of our assumptions and the potential impact on our findings. We have also discussed the need for future research with more suitable data to validate and extend our conclusions.

Furthermore, your question inspired us to conduct a deeper investigation into what our skill hierarchy truly uncovers. This led us to perform a short case study, which we have included in the Supplementary Information, Section 3.4 and Figure S17, as well as in the main text and Figure 1.2.1 below. Due to the complexity of the dataset and the noise present in the job mobility data, we focused on one specific case to track job mobilities and their associated skill requirement categories. This simple study has generated additional future research ideas that we have started working on, such as building a machine learning model to extract the latent structure of career mobility, which is a more technical and engineering-oriented study.

Finally, we explored the career progression from registered nurses (RNs) to nurse practitioners (NPs), analyzing resume data to understand how skill and wage differences manifest themselves in career trajectories. The results, visualized below, show that skills required for higher-paying NP positions appear in nested paths encompassing common RN skills and specialized ones in medicine, therapy, biology, science, and chemistry. Initially, we attempted to expand this analysis using a simple Markovian matrix of

immediate transitions, but it failed to capture meaningful career paths. The complexity and nonlinearity of career transitions necessitate methodologies that trace long-term trajectories. We are currently developing a machine learning model to uncover these trajectories, and hope to gain insights into the dynamics of labor markets and inform policies promoting worker development and addressing wage disparities. But it will take a year to get a meaningful model to capture what we have here now. Instead, we use this case study to serve as a preliminary exploration using a rudimentary model (dividing each resume into early career and late career).

Figure S17: Transition between Registered Nurses (RNs) and Nurse Practitioners (NPs). (a) uses resume data from Burning Glass Technology to capture the transition statistics between RNs and NPs. We restrict the analysis to individuals with at least five listed occupations in their resume and define their early career occupations as the most appeared occupation in the first three jobs, similarly late career occupations as the most appeared in the fourth jobs and onward. We disregard individuals whose early and late careers are neither RN nor NP. Including these individuals would not change the result but significantly complicate the exposition. One expects that higher wages for NPs would attract RNs (e). Indeed, most NPs were RNs early on. However, only a subset of RNs progresses to NP jobs, suggesting barriers to entry, summarized in higher experience and educational requirements (c-d). (b) captures the skill requirements of RNs and NPs, highlighting the advantage of integrating the conceptual distinction between general and niche skills with a structural network approach to studying skills in revealing pathways of progress (also known as “specialization”). The structure of our skill hierarchy also implies that progress entails co-development in certain niche skills and the prerequisite, often more general skills.

1.2. More generally, the exposition of the paper can be improved. In particular, the description of the methods to produce Figures 2A and 2B are not clear. As I understand it, you have a bipartite graph that links skills to occupations. You infer directed edges between the skills based on the asymmetrical conditional probabilities between those skills, using subtraction. Thus, you end up with a directed graph that contains only one type of node, representing skills. This language and a figure to depict it would be helpful. I would also want to know (and see depicted) the idea that there is an edge recording the weight of the difference flowing between the skills. (In the programming and math example, I think the weight is 0.32?)

A related point on exposition of methods is that it is quite opaque how skill levels are used to measure the “skills demanded by each occupation”. My understanding is that the raw data lists an occupation, the skills that are required to perform the occupation, and a skill level for each of those skills, with low scores indicating basic proficiency and high scores indicating expertise. The authors leverage these skill levels as indicators of the extent to which each skill is “demanded” by each occupation. Thus, the authors use these “quality” quality to measure the “quantity” of each

skill demanded by each occupation.

I thus think that the authors are conflating skill quality with skill quantity in their use of this data. As before, I do not think that this issue sinks the paper, but again it needs to be described thoroughly and I would like to see the theoretical implications discussed.

A final point on exposition is that I could not figure out, from the main text and the supplement, how the synthetic panel is constructed.

In terms of materials, I have a few comments on the first results section, beginning on Line 102. My first comment is whether Figure 1A is needed? Case examples are embedded in the top of Figure 1B already. My second comment is a question: is this first results section necessary? As far as I can tell, the authors do not use these measures of skill generality in the subsequent analyses. They are introduced almost to demonstrate the existing approach to conceptualizing the breadth of skills, a measure on which the authors aim to improve using their hierarchical approach. If the Skill Generality section is to remain in the paper, I suggest that the authors present it as a way to contrast to the hierarchical approach that they later develop. Moreover, it lends me to wonder to what extent the hierarchical approach improves on the skill generality/ubiquity framework. In this regard, it would be helpful to see a scatterplot of skills by generality (measured as per Figure 1) and nestedness (measured as per Figure 2). This would readers to evaluate the value added of the hierarchy.

We are thankful to Reviewer 1 for these suggestions. We combined your suggestions on exposition together to provide a comprehensive answer and summary as they turned out to be all related. First, the methods section was previously not detailed enough, and our rationale for the first section was not elaborated. We have completely revised this section in the manuscript and adjusted two figures accordingly (Fig 1a is out now as cases are already embedded in the insets, as you pointed out). Second, we revise the first and second sections to build logic from skill groups of general and specific to conditional probabilities as cross-group directions as reliances. This was a really useful exercise, and we really enjoyed the process of building logic here. We also found that these generality groups are eventually correlated with other quantities such as the local reaching centrality (the mass of dependencies of each focal skill). In addition, we were not clear enough to guide the measure from 'quality' to 'quantity' for both section 1 and section 2. We have now elaborated on them to incorporate a detailed explanation of the processes used to generate the graph and its weighted edges. For your convenience, we add a few highlights in the revision.

In the first result section, we add the rationale:

.... "The distinction between general and specialized skills is widely acknowledged, but a systematic quantification of this divide has been lacking. Therefore, our study starts with examining, quantifying, and classifying the generality of skills based on their breadth of application across occupations, using publicly available survey data from the U.S. Bureau of Labor Statistics (BLS)... " ... "To systematically classify skills, we group them based on similar distribution shapes, presupposing they reflect skill generality. This methodological approach indicates that general skills are ubiquitously present in occupations compared to their specialized counterparts (see Methods). ..." ... "The allocation of skills to these groups, listed in SI-Table, aligns with our common understanding of general and specialized skill categories."

In the second result section, we add:

"... our empirical generality skill groups suggest a hierarchical structure among skills, with some serving as prerequisites for others. This hierarchy has been a longstanding topic of interest in fields such as labor economics, sociology, and management, but it has not been systematically analyzed. As such, we propose a

method to quantify these relations by calculating how often occupations that require niche skills also require general skills, and compare this to the inverse---how often needing general skills predicts the need for certain niche skills. If general skills are indeed prerequisites for niche skills, much like how most college curricula have fundamental courses preceding specialized ones, we should expect to find an asymmetry in these probabilities....”

....

“...These cross-group dependencies resemble biological mutualistic interactions where specialist species preferentially interact with generalists, suggesting a nested hierarchical integration of skills. However, the result is not always obvious; not every skill may exhibit such dependencies in a linear chain. Some specialized skills, like Dynamic Flexibility, may not be contingent on more general skills like mathematical prowess, which is again consistent with our common understanding. This can be calculated as $p(\text{skill}_{dynamic\ flexibility}|\text{skill}_{math})$ and $P(\text{skill}_{math}|\text{skill}_{dynamic\ flexibility})$. We find these two are independent events in which both directions result in $p(\text{skill}_{dynamic\ flexibility}) \cdot P(\text{skill}_{math})$, resulting in no directionality in our methodological framework...”

...

“...Constructing a network structure from these conditional directions provides a methodologically consistent definition of general and specific skills using reaching centrality as an alternative measure for generality, as this can reflect the mass of interdependent nodes on the focal node...”

As for transforming continuous values to discrete values for the conditional probability of skills, the reviewer is correct that O*NET (the raw data) lists an occupation and two continuous measures about the qualitative proficiency and importance of the skill to the occupation. A *level* score indicates the required proficiency *level* of skill for the occupation [0,7], and an *importance* score indicates the extent to which the skill is critical to the occupation [1,5]. Firstly, we use the skill levels required by each occupation for job performance (the first one) but also examine the importance of skills for each occupation as a part of our robustness checks. In the original manuscript, we referred to the “demand distribution” of each skill to describe how many occupations require the skill across various proficiency levels. Recognizing that this term might be ambiguous, we have updated it to “Level Distribution” in the revised manuscript. Secondly, we have refined our manuscript to better define the concepts of demand, acquisition, and application of skills, explaining the rationale behind these terms. Thirdly, we offer further explanations and robustness tests concerning the transition from continuous (quality) to discrete values (quantity) through the application of thresholds. The results remain consistent when using the “Importance” measure as an alternative. This was not surprising, according to (Handel, 2016), as they are known to be strongly correlated (0.94,) between skill levels and importance in our dataset.

As for continuous to discontinuous measure, we apply a well-established method in network science, (Serrano et al., 2009), which was also used in Jo et al. (2020) paper, whose method we adopt to our analysis. We now elaborate on these methods and contextualize them for our datasets in the revised version (main text, methods part, and supplementary information). We provide a quick summary for your convenience although the details will be found in the manuscript.

Together with your comment below, we recognize this explanation was not enough in the text; and We changed the text accordingly. We have clarified the notion of “Level Distribution” in the Data and Method section: “*We call the distribution of the number of occupations that require a skill at varying levels, the level distribution*” and in the Results section, upon introduction of the concept “...we classify skills into categories of generality based on the shape of their level distributions, the distribution of the number of occupations that require a skill at varying levels,” and have further unpacked it using the examples of the skills in the inset of revised Figure 1.

On the weighted dependence of programming on math, we have added in the Supplementary document (Section 3.2.) the parametric derivations for filtering (Eqs. 5 and 6), and computing the strength of

dependency between skills (Eq. 7). Based on the above derivation, the dependency weights of Programming on Math Skills, which you required, is: $W_{Math \rightarrow Programming} = 0.2013$.

Finally, we are grateful to Reviewer 1 for highlighting the need for a clearer explanation of how synthetic “birth cohorts” are constructed in our manuscript. To clarify the term and the approach, we have revised the text as follows. We first renamed the method, referring to it as “Synthetic birth cohorts” or “birth cohorts” across the text, a term more broadly accepted in the literature. Basically, our underlying assumption is that attrition in the sequence of survey does not mess things up much, and thus individuals’ long-run career progression can be inferred from the population-level career progression, which is a common practice across various literature (Acemoglu & Autor, 2011, Kambourov & Mansovskii, 2013, Hermo, et al., 2022, Aeppli & Wilmers, 2022). Accordingly, we have also revised the Data and Method section to include the following description:

The Current Population Survey (CPS) conducts monthly surveys to obtain a representative sample of the population in each round (Flood et al., 2022). However, this longitudinal survey does not span over a long period of time, which presents a challenge when attempting to analyze long-term trends. To address this issue, we employ the concept of synthetic cohorts. Synthetic cohorts are constructed by stitching together snapshots of individuals born in the same year across different survey rounds. For example, to create a synthetic cohort for those born in 1970, we first identify people whose birth year was 1970 in the CPS surveys conducted in 1995, 1996, 1997, and so on, up to 2015. We then plot the data for this cohort as if we have been following the individuals born in 1970 throughout their ages, as shown in the inset of Figure 4.

It is important to note that this cohort is referred to as a “synthetic birth cohort” because it is not a real cohort in the traditional sense. The individuals surveyed by CPS in each round are different, even though they were all born in the same year. By following individuals born in the same year across multiple survey rounds, we can track changes in the behaviors or characteristics of interest as people age, albeit with different individuals representing the cohort at each point in time.

While synthetic cohorts do not provide the same level of individual-level consistency as true longitudinal studies, they offer a valuable tool for analyzing long-term trends and changes within a specific age group when long-running longitudinal data is not available. This approach allows researchers to leverage the representative nature of the CPS surveys to gain insights into the evolution of various social, economic, and demographic characteristics over time.

Acemoglu, D., & Autor, D. (2011). Skills, tasks and technologies: Implications for employment and earnings. *Handbook of Labor Economics*, 4(PART B), 1043–1171. [https://doi.org/10.1016/S0169-7218\(11\)02410-5](https://doi.org/10.1016/S0169-7218(11)02410-5)

Aeppli, C., & Wilmers, N. (2022). Rapid wage growth at the bottom has offset rising US inequality. *Proceedings of the National Academy of Sciences of the United States of America*, 119(42), 1–7. <https://doi.org/10.1073/pnas.2204305119>

Flood, S., King, M., Rodgers, R., Ruggles, S., Warren, J. R., & Westberry, M. (2022). *Integrated Public Use Microdata Series, Current Population Survey: Version 10.0 [dataset]*. IPUMS. <https://doi.org/https://doi.org/10.18128/D030.V10.0>

Handel, M. J. (2016). The O*NET content model: strengths and limitations. *Journal for Labour Market Research*, 49(2), 157–176. <https://doi.org/10.1007/s12651-016-0199-8>

Herme, S., Päällysaho, M., Seim, D., & Shapiro, J. M. (2022). Labor Market Returns and the Evolution of Cognitive Skills: Theory and Evidence. *The Quarterly Journal of Economics*, 137(4), 2309–2361. <https://doi.org/10.1093/qje/qjac022>

Jo, W. S., Park, J., Luhur, A., Kim, B. J., & Ahn, Y.-Y. (2020). Extracting hierarchical backbones from bipartite networks. <http://arxiv.org/abs/2002.07239>

Kambourov, G., & Manovskii, I. (2013). A cautionary note on using (March) current population survey and panel study of income dynamics data to study worker mobility. *Macroeconomic Dynamics*, 17(1), 172–194. <https://doi.org/10.1017/S1365100510000350>

Serrano, M. Á., Boguñá, M., & Vespignani, A. (2009). Extracting the multiscale backbone of complex weighted networks. *Proceedings of the National Academy of Sciences*, 106(16), 6483–6488. <https://doi.org/10.1073/pnas.0808904106>

- 1.3. In some areas, the paper is in a position to contribute more deeply to existing literatures than it currently does. I have a number of suggestions here. First, the cities analysis has a lot of potential, but is currently underdeveloped. To some extent, I think that the authors could withhold their analysis of the geography of nested skills from this paper, and write a separate paper about how their nestedness framework helps to weigh in on the diversity vs. specialization debate in urban economics and economic geography – that debate is still stuck on defining and measuring specialization and diversity, and I believe that the skill hierarchy produced in this paper would lend a lot of clarity to that literature. As it currently stands, the cities analysis currently confirms much of what urbanists already know about the skill composition of cities, and raises (but does not answer) a handful of interesting questions about cities, such as whether the high level of general skills in big cities is driven by a deep local division of labor, or that the professionals in the tradable sectors in big cities also have developed foundational, general skills. Moreover, I was curious to what extent the big-city results were driven by within-individual or across-individual variation. I would like to see a decomposition by cities based on their raw material / natural resource extraction. This has a nice feature because natural resources are the commodities that generally require the least human capital input to enhance their value. Again, I think it would take a full-length paper to bring out all of the contributions that this paper can make to our understanding of skill compositions in cities.

We are thankful to Reviewer 1 for the insightful comments on the analysis of the geographic distribution of skills. We side with the suggestions of reviewers 1 as well as 4 that heterogeneous and complex determinants of geographic concentration of skills require a more careful assessment, imposing significant changes to the current set of analyses to produce robust and insightful results. We agree with both reviewers that the task is likely beyond the core scope of the current paper. Therefore, we have moved the current analysis to the supplementary document, addressed the alignments of our skill structure with known facts in the literature, and will take on the suggested directions by reviewers in future work.

However, we felt compelled to follow the curiosity behind the suggestions by Reviewer 1 to the extent that our data and framework accommodate. Part of the economic geography and science of cities literature has found evidence of urban wage premiums, which it has associated with the complexity of the economic activities in larger and more populous areas (Glaeser, 1999, Wheeler, 2001, Rosenthal & Strange, 2004, Combes et al, 2010, Gomez-Lievano & Patterson-Lomba, 2021). Consistent with prior work, our skill hierarchy implies that more complex economic activities, which require the accumulation of nested and, most importantly, general skills, concentrate in larger cities (Lucas, 1988, Bacolod et al. 2009, Gomez-Lievano & Patterson-Lomba, 2016). While most workers with nested skills need general skills, the concentration of managerial and other supporting roles also needs high levels of general skills. Such prediction is supported by our descriptive analysis (Fig. 6 a,b, and d in the initial manuscript). However, as Reviewer 1 pointed out, we did not offer any analysis that examines whether the accumulation of general

skills indeed explains part of the value generated in cities. Here we test that hypothesis directly by utilizing the CBSA size (CBSASZ) variable from CPS microdata, which carries information about the size of the metropolitan area in which the surveyed individual resides (since 2004). The values range from 0: areas of <100,000 inhabitants that do not meet the threshold of a metropolitan area to 6: over 5 million inhabitants. We transform these brackets to cities below and above 1M population (Hong et al., 2020).

In the model (1) of Table S6., below, we first regressed the log-wage reported by individuals to CPS on the size of the metropolitan area in which they reside, obtaining partial correlations that signify the urban wage premiums (the baseline is areas of <1M inhabitants.) In the second model of the table, we add general skills of individuals (which we obtain from matching to O*NET the occupation associated with each individual in the CPS microdata). That means that large cities tend to have more people in occupations with general skills. This bias toward more general-skill-intensive activities explains over one-third of the urban wage premiums (Lucas, 1988, Glaeser & Maré, 2001, Bacolod et al., 2009, Gomez-Lievano & Patterson-Lomba, 2021). Adding nested and un-nested specific skills first without and then with general skills, in models 3 and 4, respectively, have similar effects. This analysis is made available as part of the SI Sec. 6.

Urban Wage Premiums and Skills				
	Dependent variable:			
	Log(Wage) OLS			
	(1)	(2)	(3)	(4)
Population > 1M	0.082*** (0.080, 0.084)	0.054*** (0.053, 0.056)	0.056*** (0.054, 0.058)	0.059*** (0.057, 0.060)
General Skills		0.269*** (0.268, 0.270)		0.281*** (0.278, 0.283)
Nested Specific Skills			0.248*** (0.246, 0.250)	0.007*** (0.005, 0.010)
Un-nested Specific Skills			-0.073*** (-0.074, -0.072)	0.026*** (0.025, 0.027)
Constant	4.671*** (4.670, 4.673)	3.787*** (3.783, 3.792)	4.471*** (4.469, 4.474)	3.709*** (3.702, 3.717)
Observations	635,554	635,554	635,554	635,554
R ²	0.012	0.200	0.141	0.203
Adjusted R ²	0.012	0.200	0.141	0.203
Residual Std. Error	0.368 (df = 635552)	0.331 (df = 635551)	0.343 (df = 635550)	0.331 (df = 635549)
F Statistic	7,845.032*** (df = 1; 635552)	79,439.180*** (df = 2; 635551)	34,872.520*** (df = 3; 635550)	40,451.410*** (df = 4; 635549)
Note:	*p<0.1; **p<0.05; ***p<0.01			

Bacolod, M., Blum, B. S., & Strange, W. C. (2009). Skills in the city. *Journal of Urban Economics*, 65(2), 136–153. <https://doi.org/https://doi.org/10.1016/j.jue.2008.09.003>.

Combes, P.-P., Duranton, G., Gobillon, L., & Roux, S. (2010). *Estimating Agglomeration Economies with History, Geography, and Worker Effects* (pp. 15–66). National Bureau of Economic Research, Inc.

Glaeser, E. L. (1999). Learning in Cities. *Journal of Urban Economics*, 46(2), 254–277. <https://doi.org/https://doi.org/10.1006/juec.1998.2121>.

Glaeser, E. L., & Maré, D. C. (2001). Cities and Skills. *Journal of Labor Economics*, 19(2), 316–342. <https://doi.org/10.1086/319563>.

Gomez-Lievano, A., & Patterson-Lomba, O. (2021). Estimating the drivers of urban economic complexity and their connection to economic performance. *Royal Society Open Science*, 8(9), 210670. <https://doi.org/10.1098/rsos.210670>.

Gomez-Lievano, A., Patterson-Lomba, O. and Hausmann, R., 2016. Explaining the prevalence, scaling and variance of urban phenomena. *Nature Human Behaviour*, 1(1), p.0012.

Hong, I., Frank, M. R., Rahwan, I., Jung, W. S., & Youn, H. (2020). The universal pathway to innovative urban economies. *Science Advances*, 6(34), 1–7. <https://doi.org/10.1126/sciadv.aba4934>

Lucas, R. E. (1988). On the mechanics of economic development. *Journal of Monetary Economics*, 22(1), 3–42. [https://doi.org/https://doi.org/10.1016/0304-3932\(88\)90168-7](https://doi.org/https://doi.org/10.1016/0304-3932(88)90168-7).

Rosenthal, S. S., & Strange, W. C. (2004). Chapter 49 - Evidence on the Nature and Sources of Agglomeration Economies. In J. V. Henderson & J.-F. B. T.-H. of R. and U. E. Thisse (Eds.), *Cities and Geography* (Vol. 4, pp. 2119–2171). Elsevier. [https://doi.org/https://doi.org/10.1016/S1574-0080\(04\)80006-3](https://doi.org/https://doi.org/10.1016/S1574-0080(04)80006-3).

Wheeler, C. H. (2001). Search, Sorting, and Urban Agglomeration. *Journal of Labor Economics*, 19(4), 879–899. <https://doi.org/10.1086/322823>.

- 1.4. The demographic analyses (gender, race, ethnicity) are very interesting, and without individual level on these attributes (they are crosswalked by occupation), I think it makes sense to keep these analyses within the current paper (for example, the authors do not have individual level data on women workers, so they cannot test whether the slow adoption of nested skills by women beyond age 30 is associated with shifting to motherhood).

We appreciate that Reviewer 1 recognized our attempt at offering a skill-based explanation for ethnic and gender socio-economic inequities and guide policy to mitigate them. We are also intrigued by what seems to be a substantial influence of children on individuals' careers. After some investigation, we found information in the Current Population Survey (CPS) that would allow us to investigate the role of children explicitly. CPS for the entire period of our sample records the number of children in the surveyed households. Using this information, we split the sample into individuals with and without children and replicate the analysis of synthetic birth cohorts (previous Fig. 3d-f and the new Fig. 4d-f): we track the skills manifested in the occupational compositions of birth cohorts as they age, splitting individuals based on their binary gender (Male: lower panels; Female: upper panels of the below figure) and their binary parental status (with child: square; without child: triangle in the below figure) at the time of survey.

The result of aggregating skills can be seen below. Each column shows the levels of a certain category of skills, while the rows show the results for men and women. The solid line (and triangles) show the pattern for people without children, while the dashed line (and squares) show the pattern for individuals with children. There is a pronounced gap in the general and nested skills between women with and without children. Please note that the later convergence is likely to arise from the fact that at higher ages, the "without category" will mix families who never had any children with families whose children have already left the household. In the latter families, caregivers may have been disadvantaged in their early careers, leading to lower skill levels at higher ages. Contrary to the negative correlation with general and nested skills, women with children appear to sort into jobs that require higher un-nested skills. Interestingly, men with children tend to do better. Especially men in jobs that require general and nested skills, tend to progress steadily and for longer periods compared to their counterparts without children. The latter pattern for men may arise from sample selection effects, from potential domestic division of labor, or from the fact that the cost of raising children incentivizes acquiring skills that lead to better paid careers. It is noteworthy that as Reviewer 1 alluded, synthetic birth cohorts are not the ideal data for this purpose, as it does not allow for tracking individuals over time. However, it is reasonable to believe this approach offers unbiased estimates of the population behavior.

Fig. S49. In the figure, we track the skills manifested in the occupational compositions of birth cohorts as they age, splitting individuals based on their binary gender (Male/Female) and whether they lived with children at the time of the survey, obtaining the below figure. Each column shows the levels of a certain category of skills, while the rows show the results for men and women. The solid line (and triangles) show the pattern for people without children, while the dashed line (and squares) show the pattern for individuals with children. There is a pronounced drop in the general and nested skills for women with children.

- 1.5. In the demographic analyses, the gap between the skill level for men and women in un-nested occupations was really remarkable (Figure 3F). This makes sense to me, in that we often think of applied/mechanical jobs that are high-paying to be “mens’ work” (think of oil well operators or deep sea fisherman). But I would like to see more discussion and analysis of this finding.

Thank you for bringing up this interesting point. In line with your prior comment, we were also intrigued by the diverging patterns of skill development between men and women, wherein women exhibit high levels of general skills, surpassing their male counterparts at certain ages (as shown in previous Fig. 3d-f and the new Fig. 4d-f) but do not manifest the high levels of nested skills observed for male workers of the same age. Lower levels of nested skills for women is also seen in the first column of table S7, below, which predicts the gender of workers based on their general and nested skills in our CPS sample (Female = 1): general skills are associated with greater, but nested skills with smaller shares of women in an occupation.

Indeed, a comprehensive treatment of this observation merits a project in its own right. However, one explanation for this pattern, at the center of the nobel-prize winning work of Claudia Goldin, is that women may avoid jobs with irregular or long working schedules (Bertrand et al., 2009, Goldin, 2015, Canon & Golan, 2016). Hence, despite their high levels of general skills and education, women may avoid jobs that require nested skills because of the working conditions of such jobs. To examine that hypothesis, we

examined whether adding descriptors of work schedules to the same regression diminishes the correlation between skills and the gender of the worker, as reported in column 1 below.

To implement this test, we matched individuals in the Current Population Survey (CPS) aged 18 to 55 who were in the workforce between 1980 and 2020 to the following information in the O*NET using their reported occupation code: skill information (namely, general and nested specific skills) and occupational work schedule (irregularity). The latter variable is collected as a part of the O*NET work context record as a categorical variable with three levels:

1. Regular (established routine, set schedule)
2. Irregular (changes with weather conditions, production demands, or contract duration)
3. Seasonal (only during certain times of the year)

This variable is reported for all occupations with weights associated with each category. For example, the Chief Executive has the majority of weight in category 1, as it is primarily a job with a regular schedule. A surgeon has more weight in comparison to the irregular category. Using the weights, we obtained an aggregated “schedule irregularity” score for each occupation, wherein a value closer to 1 denotes a regular schedule, and a value closer to 3 denotes a more irregular schedule. To create a proxy for long working hours, we follow Cha et al. (2014) to use the number of hours worked during the week in the CPS data and form a dummy variable that is 1 if the worker had worked more than 50 hours a week, and 0 otherwise.

Adding the descriptors of irregular schedules or long hours indeed diminishes the correlation estimated in the baseline. Per the baseline model of column 1, a unit increase in the nested specific skills required by a job decreases the chances of the worker being female by ~36%. Adding schedule descriptors reduces that relation by more than one-third, to about 26%.

Skills, Work Schedule, and Gender		
	Dependent variable:	
	Gender Dummy (Female = 1)	
	OLS	
	(1)	(2)
General Skills	0.203*** (0.201, 0.204)	0.150*** (0.148, 0.153)
Nested Skills	-0.357*** (-0.359, -0.355)	-0.258*** (-0.261, -0.256)
Irregular Schedule		-0.338*** (-0.342, -0.334)
Long Hours Dummy (>50 weekly)		-0.176*** (-0.178, -0.174)
Constant	0.097*** (0.092, 0.101)	0.629*** (0.620, 0.638)
Observations	1,493,142	1,096,362
R ²	0.072	0.108
Adjusted R ²	0.072	0.108
Residual Std. Error	0.463 (df = 1493139)	0.455 (df = 1096357)
F Statistic	57,942.160*** (df = 2; 1493139)	33,058.290*** (df = 4; 1096357)
Note:	*p<0.1; **p<0.05; ***p<0.01	

Table S7. Regression analysis of the correlation between gender, skills and irregular and long work schedule. The first column offers a baseline model that predicts the gender (Female = 1) of the worker based on general and nested skills showing a negative correlation with nested skills. Adding descriptors of irregular and long schedules in the second model explains away part of the predictive power of nested skills for workers’ gender. As such, part of the reason why women manifest high levels of general skills but

comparatively low levels of nested skills is that jobs that require the latter categories of skills likely impose long and irregular work conditions, which have been found to deter female workers.

Bertrand, M., Goldin, C., & Katz, L. F. (2009). *Dynamics of the gender gap for young professionals in the corporate and financial sectors.*

Canon, M., & Golan, L. (2016). Gender Pay Gap May Be Linked to Flexible and Irregular Hours. *The Regional Economist, July.*

Cha, Y., & Weeden, K. A. (2014). Overwork and the slow convergence in the gender gap in wages. *American Sociological Review, 79*(3), 457–484.

Goldin, C. (2015). Hours flexibility and the gender gap in pay. *Center for American Progress, 31.*

- 1.6. I am also interested in seeing the authors interpret the higher attainment of Hispanic/Latinx workers in unnested skills. Is there a tradeoff for unnessted skills, in that they can be attained relatively quickly and at lower cost, but offer fewer avenues toward entering high-income occupations; moreover, by being involved in primarily unnested skills, are Latinx and many male workers stuck in a sort of “middle income trap”, where they could reach moderate income levels relatively easily but have not built the foundational skills needed to achieve high income?

This is an excellent question that helped us dive deeper into how our framework helps explain some of the racial disparities in the labor market. We approach this in two ways. First, we suspect language skills are barriers to some Hispanic workers, particularly early on in their careers in the US. A lack of language skills will mostly hamper the acquisition of (language-related) general and of (the downstream) nested specific skills, but less so the acquisition of unnested specific skills. To test this, we split the sample of individuals from the CPS into four categories, ordered based on their likely level of English proficiency: Hispanics born outside of the US who immigrated less than a year before the survey, Hispanics born outside the US who have been in the US for more than a year, Hispanics born inside of the US, and White workers. We map the average skill levels of each of the above subgroups for each skill category in Fig. S18, below. As hypothesized, the foreign-born Hispanics who recently migrated to the US have the lowest levels of general and nested skills and have the highest unnested skills. The suspected ranking of English proficiency of each subgroup is consistent with their ranking in terms of general, nested, and unnested skills. We investigate the role of language skills directly in the next figure.

Figure S18: Comparison of the Skill Levels of Hispanic Immigrants and White Workers We distinguish between four groups of workers (i. foreign-born Hispanics who have migrated less than a year to the US from the time of survey, ii. foreign-born Hispanics who have been in the US for more than a year, iii. US-born Hispanics, and iv. the White workers) and map their average skill levels for each skill category. Recently migrated foreign-born Hispanics have the least levels of general and nested skills and most un-nested skills.

Our network allows us to directly identify which nested skills more closely depend on language general skills. To do so, we first identify six general skills as “language-related”: i. English Language, ii. Oral Expression, iii. Oral Comprehension, iv. Written Expression, v. Written Comprehension, and vi. Speaking. One can quantify the dependence of each nested skill, i , on each of the mentioned language skills, j , by deriving the arrival probability of a random walk starting from the mentioned language general skills, $P_{i,j}^{<arrival>}$. Aggregating these probabilities over the language general skills, we obtain $\bar{P}_i^{<arrival>} = \frac{1}{6} \sum_j P_{i,j}^{<arrival>}$. We flag nested specific skills at the top 25% of skills in terms of their average arrival probability, $\bar{P}_i^{<arrival>}$, obtaining the following skills: i. History & Archeology, ii. Management of Material Resources, iii. Management of Financial Resources, iv. Programming, v. Philosophy & Theology. Splitting general and nested skills by their language associations (general skills into *Language-related* and *Non-language* skills, and nested skills into *Language dependent* and *Language independent*), we obtain the average skill levels of individuals for the previously defined subgroups of workers (Hispanic and White based on their place of birth and time since immigration).

In Fig. S19, we show the ratios of skills levels for the different Hispanic subpopulation groups relative to White workers for the Language-related (i. English Language, ii. Oral Expression, iii. Oral Comprehension, iv. Written Expression, v. Written Comprehension, and vi. Speaking) and non-language general skills and Language-dependent (defined as the skills in the top 25% arrival probability to the mentioned language skills) and Language-independent nested skills (i. History & Archeology, ii. Management of Material Resources, iii. Management of Financial Resources, iv. Programming, v. Philosophy & Theology). The results show that the skill gaps between Hispanic subpopulations and White workers mimic the implied language gradient: the less proficient in English a subgroup will be, the larger the gap is to White workers in language-dependent nested skills but not in language-independent specific skills. This supports our hypothesis that the skill gaps for Hispanic workers as a whole are, at least in part, due to language barriers.

Finally, in response to your question about the possibility of a wage entrapment due to un-nested skills, we show “wage curves” that depict wages as a function of age (Argote & Eppel, 1990, Jovanovic et al., 1995, Jovanovic & Nyarko, 1997, Lange, 2007, Nagypál, 2007, Kahn & Lange, 2014, Nedelkoska et al., 2015), for individuals in the most nested and the most un-nested occupations. To do so, we averaged over the levels of nested and un-nested skills of each occupation in our sample and picked occupations at the top 20% of the nested skills as the most nested and occupations at the top 20% of the un-nested skills as the most un-nested. Matching these occupations to the individuals in the CPS, we can obtain estimates of wages for individuals in these occupations at different ages. To avoid conflating long-run economic factors, we show

the wage-age curves for four distinct periods of 5-years: 1983-1987, 1993-1997, 2003-2007, 2013-2017, in Fig. S30. In three of the four periods, un-nested jobs have an early wage lead, which quickly evaporates with age. The pattern is consistent with the notion that learning is steeper in occupations with more complex tasks (Jovanovic & Nyarko, 1997, Nedelkoska, et al., 2015). To arrive at a complete picture, one would need to account for the higher cost of education associated with nested occupations. Hence, the wage offsets observed in the figure may occur later in individuals' lives in terms of real earnings once the cost of education is accounted for.

Figure S19: Language Barriers Manifest in Lower Levels of Language-related Nested Skills for Hispanics. The figure depicts, for the Language-related and non-language general skills and Language-dependent (defined as the skills in the top 25% arrival probability to the mentioned language skills) and Language-independent nested skills, the ratios of skills levels for the different Hispanic subgroups relative to White workers. The results depict that the language-dependent nested skills vary significantly more across the Language-dependent subset, supporting our suspicion that Hispanic workers, at least in part, suffer from their language skills, which prevents them from acquiring/applying downstream skills.

Figure S30: Wage Curves for Occupations with Primarily Nested vs. Primarily Un-nested Skills. We average over the levels of nested and un-nested skills of each occupation in our sample, and pick occupations at the top 20% of the nested skills as the most nested, and occupations at the top 20% of the un-nested skills as the most un-nested. Matching these occupations to the individuals in the CPS, we can obtain estimates of wages for individuals in these occupations at different ages. To avoid conflating long-run economic factors, we show the wage-age curves for four 5-year periods: 1983-1987, 1993-1997, 2003-2007, 2013-2017. Un-nested jobs have an early wage lead which quickly evaporates with age.

Argote, L., & Epple, D. (1990). Learning Curves in Manufacturing. *Science*, 247(4945), 920–924. <http://www.jstor.org/stable/2873885>.

Jovanovic, B., Nyarko, Y., & Griliches. (1995). A Bayesian Learning Model Fitted to a Variety of Empirical Learning Curves. *Brookings Papers on Economic Activity. Microeconomics*, 1995, 247–305. <https://doi.org/10.2307/2534775>.

Jovanovic, B., & Nyarko, Y. (1997). Stepping-stone mobility. *Carnegie-Rochester Conference Series on Public Policy*, 46, 289–325. [https://doi.org/https://doi.org/10.1016/S0167-2231\(97\)00012-2](https://doi.org/https://doi.org/10.1016/S0167-2231(97)00012-2)

Kahn, L. B., & Lange, F. (2014). Employer Learning, Productivity, and the Earnings Distribution: Evidence from Performance Measures. *The Review of Economic Studies*, 81(4), 1575–1613. <https://doi.org/10.1093/restud/rdu021>

Lange, F. (2007). The speed of employer learning. *Journal of Labor Economics*, 25(1), 1–35. <https://doi.org/10.1086/508730>

Nagypál, É. (2007). Learning by doing vs. learning about match quality: Can we tell them apart? *Review of Economic Studies*, 74(2), 537–566. <https://doi.org/10.1111/j.1467-937X.2007.00430.x>

Nedelkoska, L., Patt, A., & Ederer, P. (2015). Learning by problem solving. *Available at SSRN*.

Norris, J. R. (1998). *Markov chains* (No. 2). Cambridge university press.

- 1.7. Rhetorically, I found the introduction to not fit the paper exactly. I appreciated the writing on the historical growth of complexity of the economy over time, but the actual lead of the article does not appear until line 62, when the authors (finally) state that “certain skills can be gained without any prerequisites . . .while other skills are not immediately attainable”. This is a great sentence and a good point, should be featured in the first paragraph. Moreover, first two paragraphs of the introduction currently present the paper as if it is about growth, but it is not.

Thank you for bringing this issue to our attention. It was a common concern among the reviewers. This was because we are coming from the field of economic complexity, which was the main motivation behind our project but may have led to unnecessary reasoning in the manuscript. In response to your feedback, we have revised the introduction to be more straight to the research question and the importance of skill dependencies. More specifically, we now bring the following up to the second paragraph with the key idea that occupational specialization is not an isolated phenomenon but is instead embedded within a complex network of interdependencies that span across teams, companies, and entire sectors. This network of dependencies implies that certain skills can be acquired without any prerequisites, while others are not immediately attainable and require a foundation of prior knowledge and expertise. By highlighting this central concept early in the introduction, we aim to provide a clearer and more focused context for the research that follows. We appreciate your valuable feedback, as it has helped us to improve the clarity and impact of our manuscript.

- 1.8. Finally, in attempting to get a sense of the extent of the contribution, I wondered if we can distinguish, using your framework, between the skills like Fine Art and Programming. I raise these skills because they are both are grouped as “nested skills” because they draw on a large range of more basic fundamental skills. But wages are strikingly different across these two skills. Ultimately, human capital theory is intended to explain things like productivity and wages. How can your framework help us to understand why programmers are paid more than people with expertise in the fine arts? Rhetorically speaking, what is your model’s R-squared in predicting wages?

This is an excellent question, which prompted us to develop a new result section dedicated to better highlighting the contribution of our nestedness framework to the value of human capital. We also add some elaboration that not every specific skill is dependent on general skills and the chains sometimes branch off and sometimes break. Therefore, we identify those specific skills that are dependent on general skills and those that are not. We operationalize these differentiations by employing nested contribution score to the overarching nested architecture, that is, c_s . With c_s , we define skills with $c_s \geq 0$ as nested skills and skills $c_s < 0$ un-nested skills because of negative contribution to the overall nested structure.

This method of the nestedness contribution of a skill c_s denotes the extent to which it facilitates complementarity (i.e., mutualism) across the occupation-skill network (Saavedra et al., 2011). To put that into context, removal of a node with a high nested contribution decreases the overall network nestedness, possibly suggesting a decline in the corresponding occupations' capacity to extract synergy from their skills. Hence, one expects using highly nested skills to contribute to occupations' wage premiums.

We test the above by estimating the value of a skill based on the weighted wage of occupations using each skill and comparing the skills' nestedness contribution c_s . The results are included in the new section, and shown below for your convenience.

Figure 3: Skill Nestedness Contributions Score.: skills' nestedness score is highly indicative of their generality (a), risk of automation (b), and their value (c-d). Skill Nestedness Contributions is measured following [15]. Generality is measured by Local Reaching Centrality, as in Fig. 2, Automation risk Index and Value for each skill is calculated, following [65–67]. We divide skills into *nested*, positive contributions, and *un-nested*, negative contributions toward the nested skill structure.

On your request, we also identified Programming and Fine Arts and found Programming has a higher contribution than Fine Arts, thus expected and also observed differential wages. This was an incredible finding and assured that the structural property of nestedness contains information on skills' rewards.

The result has an intuitive explanation based on the bargaining power theory (Davidson, 1898). *Skills "that conform to the architectural expectation of a nested network are subject to greater constraints than nodes that interact freely"* (Saavedra et al., 2011). If these constraints are costly, then the more a skill contributes to nestedness, the harder it is to acquire it, which explains its value based on the bargaining power it provides.

Finally, on the power of our models to predict occupational wages, we have included relevant statistics, including R-squared, for the models used in the paper in the SI Table S5, also included here for your convenience.

	log(Wage₂₀₁₉)					
	(1)	(2)	(3)	(4)	(5)	(6)
General	0.251*** (0.237,0.264)			0.135*** (0.109,0.161)		
Intermediate^{Nested}		0.260*** (0.245,0.274)			0.120*** (0.098,0.141)	
Intermediate^{Unnested}		0.015** (0.002,0.028)			0.034*** (0.020,0.047)	
Specific^{Nested}			0.199*** (0.182,0.215)			0.042*** (0.020,0.064)
Specific^{Unnested}			-0.051*** (-0.063,-0.039)			0.003 (-0.011,0.017)
Education				0.024*** (0.018,0.030)	0.034*** (0.029,0.039)	0.042*** (0.036,0.048)
Experience				0.014*** (0.008,0.021)	0.014*** (0.008,0.021)	0.023*** (0.016,0.030)
Training				0.038*** (0.029,0.048)	0.027*** (0.017,0.037)	0.042*** (0.031,0.052)
Constant	3.909*** (3.861,3.956)	4.069*** (4.017,4.121)	4.550*** (4.520,4.581)	3.969*** (3.908,4.031)	4.043*** (3.997,4.090)	4.223*** (4.182,4.264)
Observations	789	789	789	789	789	789
R ²	0.622	0.607	0.470	0.686	0.703	0.653
Adjusted R ²	0.622	0.606	0.469	0.684	0.701	0.651
Residual Std. Error	0.119	0.122	0.142	0.109	0.106	0.115
F Statistic	1,297.040***	607.527***	348.362***	428.084***	370.741***	294.761***

Note: OLS regressions are shown, with 95-percentile confidence intervals in parentheses (* $p < 0.1$; ** $p < 0.05$; *** $p < 0.01$). R², coefficient of determination, and adjusted R² is normalized for the models' number of variables.

SI Table S5: Wage Regression on Skill Endowment.

Davidson, J. (1898). *The bargain theory of wages ...* G. P. Putnam's Sons.

Saavedra, S., Stouffer, D. B., Uzzi, B., & Bascompte, J. (2011). Strong contributors to network persistence are the most vulnerable to extinction. *Nature*, 478(7368), 233–235. <https://doi.org/10.1038/nature10433>

1.9. -- Smaller points --

In the beginning of the results section, I found the language of how occupations

“demand” skills to be confusing. For me, this evoked the concept of demand in the economic sense, so my mind immediately started to think about a market for skills. This is not what you are describing --you’re just describing skill requirements in occupations – so I suggest simplifying the language.

We appreciate Reviewer 1 for pointing out ways we can clarify the exposition and methods. We recognize this is a potential mischaracterization of our work, but unfortunately we could not find a better alternative word/terminology yet. To address your concern, we have defined what we mean by demand for a skill as “the number of occupations that require a skill at a given level.” We have also carefully revised the mentioned section so that we do not utilize demand out of context. We have also attempted to state more clearly what we mean by “demand profiles”, by replacing them with “level distribution” of skills throughout the revised manuscript.

Your concern is in line with your previous comment. As explained before, by counting occupations that need a certain level of skill, we transform the quality to quantity as the number of occupations that need the skill at a certain level. By aggregating the differential frequencies that occupations require the skill, we distinguish between the shape of the demand profile of different skills, allowing us to categorize skills according to their demand distribution shape rather than their contexts. For example, some skills are ubiquitously required at high levels (such as English Language), while some skills are in need at high levels for a few occupations (such as Fine Arts). This was an important step as we are looking at the skills in *not* their context similarity but in their “demand” similarity.

In the initial manuscript, the term “demand distribution” for each skill denotes the number of occupations that use the skill at different levels of proficiencies. We concede that the term may be misleading, and have in the revised version, changed it to “Level Distribution”.

To address your concern here and in previous comments, we recognize this explanation was not enough in the text, and we have changed the text accordingly. We have clarified the notion of “Level Distribution” in the Data and Method section: “*We call the distribution of the number of occupations that require a skill at varying levels, the level distribution*” and in the Results section, upon introduction of the concept “...we classify skills into categories of generality based on the shape of their level distributions, the distribution of the number of occupations that require a skill at varying levels,” and have further unpacked it using the examples of the skills in the inset of revised Figure 1.

- 1.10. Some of the core writing of the intro is unclear. For example, the final sentence of the intro (line 100) refers to a widening gap between the two, but I do not know what “the two” refers to..

We appreciate Reviewer 1 for pointing out the vagueness in the introduction, particularly the statement mentioned. This was supposed to indicate the widening gap between the two parts of the network, where one primarily nested and the other primarily unnested skills reside. In the revised version, we have completely rewritten the Introduction, paying attention to the clarity of language and the terminology we used throughout the paper.

- 1.11. For median ages for occupations, the figure should say “the median age of workers in each occupation”. The current text suggests to me that you might be counting the numbers of years elapsed since the BLS introduced each occupation code to its classification system.

We are thankful to Reviewer 1 for pointing out the possible confusion caused by the title of Figure 3 (a-c). The mentioned figure is revised accordingly (currently Fig. 4), and the revised version is presented below. We have also ensured the figure's caption and the description in the Data and Method section reflect that the x-axis represents the median age of the employed labor in each occupation.

Figure 4: Skill Compositions with Occupational Ages and Career Trajectory. (a-c) Average skill levels of occupations (and 95% confidence intervals), segmented by occupations' employees' median ages. Levels of general and nested skills rise with an occupation's median age, while un-nested skills do not vary across median-age groups. (d-f) Average skill levels (and 95% confidence intervals) against age in synthetic birth cohorts. The insets isolate cohorts born in 1967, whereas the main figures average across all cohorts. Notably, general and nested skills rise markedly until around age 30, with declining un-nested skills. Moreover, gender gaps also become more pronounced around 30. (g-h) Average skill levels (and 95% confidence intervals) over identified job sequences as documented in resumes for general, nested, and un-nested skills. (i) Changes in skill levels in consecutive job transitions. Skill profiles are typically stabilized within the initial five jobs. The grey triangles indicate bootstrapped results where the sequences of jobs are randomized.

1.12. Why in the maps of skills by county (Figure 6 and in the supplement) do there appear to be state-level effects? For example, Wyoming is almost all the same shade of blue, as is Nevada? Is the occupational info classified uniquely by each state? I am trying to figure out why there would be any state-level grouping, given that the economies are porous between adjacent counties across state lines.

Thank you for pointing this out. The data's geographical unit is the Metropolitan Statistical Areas (MSA) and Counties. We suspect our geographical analysis lacks power and depth in dealing with a great deal of nuances that belie the geographic distribution of skills. In response to your and other reviewers' concerns, which we agree on, we moved the analyses in the SI to be an input for follow-up research in the future. In any case, we provide our methods in detail below.

First, the analysis employs occupational employment data by area (Occupational Employment and Wage Statistics for May 2019 produced by BLS) at the level of Metropolitan/micropolitan Statistical Areas (MSA). We obtain county-level occupational employment (that allows the geographic inference of skills in counties, visualized in the mentioned figures) through a crosswalk (also provided by BLS) between metropolitan/micropolitan statistical areas and FIPS code (representing counties). In the crosswalk, county-level FIPS can be obtained from concatenating FIPS and county codes. However, the matching from MSA to FIPS is many-to-many in some instances and likely washes out some of the signal in the data.

Second, occupational employment is correlated for counties in the same state. This is perhaps more intense for less populous counties. The data bears out this suspicion: for each state, we form a matrix of the employment for each occupation; then, calculate a correlation matrix based on the employment requirements of each pair of counties in the state; the mean and median of the upper triangle of such a correlation matrix is reported in Fig. 1.12.1 below for 20 *least* correlated states (sorted by median employment correlation). These figures capture the high levels of (average and median) similarity in the occupational employment of counties within the same state.

	State	Avg. Cor	Med. Cor
1	Nevada	0.6489000	0.6749303
2	North Dakota	0.7752025	0.7531286
3	Maine	0.7569847	0.7690763
4	Indiana	0.7724164	0.7711732
5	Massachusetts	0.7272225	0.7713588
6	Hawaii	0.7098591	0.7716991
7	Tennessee	0.7326185	0.7754035
8	Louisiana	0.7967349	0.7896386
9	Puerto Rico	0.8022018	0.7928329
10	New Jersey	0.7765288	0.7942969
11	Utah	0.7746658	0.7948648
12	Florida	0.7531122	0.8019303
13	Alabama	0.7939583	0.8046104
14	Missouri	0.8047595	0.8089002
15	Pennsylvania	0.8245821	0.8181735
16	Maryland	0.8196977	0.8184346
17	New Mexico	0.8288715	0.8214059
18	South Carolina	0.8080131	0.8220101
19	New York	0.8202575	0.8227055
20	South Dakota	0.8230878	0.8230289

Fig. 1.12.1. These figures capture the high levels of (average and median) similarity in the occupational employment of counties within the same state. To calculate these values for each state, we form a matrix of the employment for each occupation; then, calculate a correlation matrix based on the employment requirements of each pair of counties in the state; the mean and median of the upper triangle of such a correlation matrix is reported here.

2. Reviewer #2

This paper presents a detailed study of how different occupational skills precede and depend on one another within a network of dependencies. Its main data source is a database of occupations and the skills that each needs. By observing the conditional probability for an occupation to require some skill A, given that it requires skill B, the paper infers a network of skill dependencies showing which skills (e.g. "Mathematics") tend to precede which others (e.g. "Programming"). From this network and other analyses, the paper presents findings about skill acquisition and its relationship to variables such as wages and the geographic distribution of employment.

The strong positives of this paper are that it offers new ways of thinking about and analyzing human capital that are potentially richer than traditional conceptualizations, and that it presents intriguing findings on the nature of skills that appear to be enabled by this new approach. The results and methods are of interest to a broad array of researchers.

Some key claims include the following:

- Many skills are "nested". These skills come from a tree-like portion of the skill network that extends from root nodes (which can be understood as foundational skills) to leaf nodes (which represent specialized skills).
- Other skills fall in an "un-nested" portion of the network. Such skills (e.g. "Repairing") do not require foundational skills (to the same degree?) as the specialized skills in the nested portion.
- The analyses focuses on differences between these 3 large categories of skills above: (1) the un-nested skills, and the 2 types of nested skills: (2) "general" skills (which I call here "foundational" skills), and (3) "specific" or specialized skills.
- Over the course of a career, workers use more of *both* foundational skills and specialized skills, and less of un-nested skills. As the authors note, the increase in foundational skills is interesting because it suggests that these skills need to be "continuously enhanced regardless of how advanced we are in our career journeys". (Also interestingly, the authors find that this increased used of foundational skills is not driven by workers' tendency to move into management jobs over time, nor is it specific to those foundational skills that could be considered "social skills". Evidently foundational skills in general get enhanced as workers progress.)
- One dramatic result is the correlation with wages: Without controlling for other variables, wages rise with an occupation's skill level in specialized skills, but not with skill level in un-nested skills. However, this difference largely vanishes when an occupation's level of foundational skills is taken into account. Apparently, without strengthening foundational skills, higher level of ability in specialized skills doesn't yield higher wages.
- Economic demand may be increasing over time for general skills over specialized and un-nested skills.

The results and methods are interesting and the manuscript warrants publication in a revised form

Thank you so much for taking the time to review our manuscript and for providing a detailed summary along with constructive feedback. Your encouragement for us to move forward is greatly appreciated. As you will see in our responses below, your suggestions have significantly helped us clarify our exposition and enhance the quality of our work. Below, we address each point you raised:

My two main concerns are below.

2.1. (1) Analysis of un-nested skills / focus on nestedness

I feel unsatisfied by the analysis of un-nested skills, and more generally by the strong focus on the

concept of "nestedness". From what I can tell, "un-nested" nodes are not un-nested - they are less deeply nested, with a different set of root nodes/foundational skills. The authors have "backboned" the network in Figure 2, i.e. what appears there drops many links for the sake of visualization. This is not necessarily a problem, but after seeing the detailed representations of this network in the SI, I am not sure why they do this, and feel the one in the main text leaves out useful information. Fig S21-S22, for example, make it clear that the nested skills are overwhelmingly non-physical ones (e.g. thinking, social interaction) and that un-nested skills overwhelmingly emphasize physical activities (dexterity, depth perception, hearing sensitivity, arm-hand steadiness). Among the un-nested skills, one can see plausible nests / sequences of skills that build on one another, from general skills to specific ones, e.g. Visualization -> Perceptual Speed -> Quality Control Analysis -> Equipment selection. The data seems to distinguish many fewer skills in this "physical-and-related" category than in the "non-physical" category. Don't the "un-nested" skills just represent a different nest, for physical and related skills? (And that it is shallower than the other nest, because there are just fewer skills to construct dependencies between?)

The labels "nested" and "un-nested" themselves are a bit abstract and it matters for interpreting how novel the results are. For example, one finding is about how cities with different levels of manufacturing intensity differ in skill emphasis:

"Upon grouping cities by manufacturing employment... we find that cities highly specialized in manufacturing tend to exhibit lower levels of nested specialization but higher level of unnested specializations."

Written this way, the statement sounds more novel than if one had said

"Upon grouping cities by manufacturing employment... we find that cities highly specialized in manufacturing tend to exhibit lower emphasis on cognitive skills but higher emphasis on physical skills."

I do not mean to convey that the emphasis on nestedness is unjustified and that findings like the one above are not interesting or novel. It's rather that I find I'm left struggling to decide on my own whether the focus on nestedness teaches me something new (or an interesting new angle on something old). It would be great for the paper to provide more guidance.

From what I can tell, the results just hinge on there being "dependencies" between skills - to do "negotiation", a worker needs a sufficiently strong basis in a general skill such as "oral expression". Why the strong focus on "nestedness"? This is a specific phenomenon where observational units - a company or a location say - have portfolios of activities that are subsets of one another. Diversified companies not only make more products than un-diversified ones; they make supersets of the portfolios of un-diversified companies. As the authors note, this phenomenon is seen in ecosystems, e.g. a diverse continental ecosystem may contain all the species of less diverse island ecosystems, plus others. The text and the title call out this ecosystem analogy. But it is only mildly developed in the paper, and I'm ultimately not sure what the significance of nestedness is here over the simpler idea that some skills depend on other skills.

On a separate note: In Figure 4c, after controlling for the level of general skills, the slope flips and wages rise with the level of un-nested skills. I am just curious whether the authors think this clashes with the idea that un-nested skills do not depend on general skills.

This paper has a number of methods and findings worth communicating to the audience of this

journal, and I do not feel all questions of interpretation should or can be resolved here. But I would like in the main text a better discussion of why "nestedness" is important and a more complete / nuanced discussion about the un-nested skills.

We extend our sincere appreciation for their meticulous review and the insightful suggestions provided. Your comprehensive feedback has been a catalyst for significant revisions to our manuscript. Upon reviewing our manuscript with your comments, it became evident that our initial approach to defining and utilizing the concepts of nested and unnested skills was insufficiently clear, lacking in both definition and justification. This oversight rendered the concepts of nestedness and unnestedness somewhat abstract, detracting from the clarity and impact of our work. In response to your comments, we have undertaken a comprehensive rewrite of our manuscript, aimed not only at clarifying these concepts but also at articulating our theoretical contributions and analytical constructs more effectively. To this end, we have introduced a new section dedicated entirely to elaborating on the nestedness measures, accompanied by the addition of one new figure and the revision of two others. In the revised manuscript, we construct a logical progression, starting with the analysis of individual skill distribution shapes (narrow vs. broad) in Section 1, followed by an examination of cross-group asymmetric relations (examinations of skills' asymmetric relations across the narrower vs. broader groups) in Section 2. We then move into the differential contributions of skills to the overarching nested structure and the differential economic rewards corresponding to nested contributions in Section 3. The rest of the sections (Sec 4, 5, and 7) are devoted to examining and testing our rationale in temporal data and demographic data. This structured approach has enabled us to elaborate on the rationale behind our methodology and articulate the novelty of our approach compared to existing frameworks. This was a really useful exercise, and we enjoyed the process of building logic here. Your feedback was invaluable in this process, providing us with a clear direction to enhance the logical coherence and depth of our manuscript.

More specifically, in response to your comments, we have made the following changes:

1. Articulated rationale for using nested structures and defined nested/unnested skills based on their structural contributions to the skill structure's overarching nested architecture.
2. Created a new section devoted to providing rationale and empirical evidence supporting our assumptions and showing differential rewards for nested/unnested skills (new Fig 3).
3. Included the full network in the inset, accompanied by the backbone, and incorporated new analyses to support our case.
4. Discussed the need for future research to validate and extend our conclusions.

For your convenience, we elaborate on a few key points in the revision:

We first explain why we use nestedness as a structural measure of dependency when generalists and specialists coexist. Using nestedness, widely used in ecology, was useful as the metric and its properties are well-understood, and thus allows us to operationalize measures and reveal economic rewarding systems without relying on context-informed categories in social science (see below). In ecology, the interdependencies of specialists on generalists imply a progression in dynamics, unfolding individual progressions as well as recapitulating organizations (areas, companies, and countries) (Hong et al., 2020). In primary succession, for example, generalists often arrive in an area first to make the system function, and specialists cannot survive in the absence of generalists, showing the directionality of their dependencies. Therefore, we aim to understand how specialists interact with generalists and make the quantity more operationalized and whether these interactions themselves reveal economic rewarding systems without relying on semantic categories from our introspective insights, such as physical/cognitive/social/managerial skills. We group skills by their demand distributions (broad and narrow) and then determine which skills are more conditionally dependent on others. Again, these are purely structural information with no introspective insights involved and thus is an interdisciplinary

approach appropriate for a general-science journal. The nestedness measure is useful for operationalizing these results in a systemic way and uncovering the importance of structural properties in the economy's rewarding system. It also allows us to quantify not only the dependencies themselves but also how skills branch off from dependencies by measuring individual skills' contributions to the overarching nested structure (a way of measuring the dependency structure).

Therefore, we defined skills that are positively contributing to the overarching nested architecture (that is, $c_s > 0$) as nested skills and skills that are negatively contributing to the total nested structure (that is, $c_s < 0$) as unnested. In doing so, we also found that skills contributing to the overall nested structure ($c_s > 0$) are rewarded the most, while skills ($c_s < 0$) are least rewarded. New Figure 3 shows how these nested scores, solely relying on the network, without knowing the actual content or nature of skills, correlate with wages. These skills, as you rightly point out, are mostly physical skills. So, in that sense, our findings are not novel, as we know most physical skills are less rewarded than cognitive skills. What is novel here is that their relations are not only shallow but also deviating from the overall nested structure and property inherent in the structure of the network, which is independent of the number of nodes that manifest such patterns. For instance, the nestedness score lens can delineate the value of Fine Arts and Programming, both niche nested skills.

We find the difference between the dependency structures of nested and unnested components observed in the network intuitive, as it aligns with the disjointed bodies of work that have described the increasing length of education and investment necessary for well-paid jobs and education wage premiums in recent years (Goldin & Katz, 2008; Jones, 2009; Autor, 2014). This is indeed the case with our new analysis in section 3. The figure below shows that our network-based approach predicts various socioeconomic factors, ranging from education and wage to demographic disparities. The result also has an intuitive explanation based on the bargaining power theory (Davidson, 1898). Skills "that conform to the architectural expectation of a nested network are subject to greater constraints than nodes that interact freely" (Saavedra et al., 2011). If these constraints are costly, then the more a skill contributes to nestedness, the harder it is to acquire, which explains its value based on the bargaining power it provides.

All in all, our findings are aligned with our understanding of the world. Much work in economics, sociology, and management has examined changes in the returns to different types of skills in the labor market. These include the importance of technical skills in facilitating the transition of college graduates into the labor market (Deming and Noray, 2020); increasing returns to social skills (Deming, 2017); and the increase in returns to cognitive skills from 1978 to 1986 contributing significantly to the rising wage premium associated with higher education (Murnane, Willett, and Levy, 1995). These skill categories are eventually correlated with education, "low-skilled" versus "high-skilled", or "low-paid" versus "high-paid" skills (Rotundo and Sackett, 2004; Acemoglu and Restrepo, 2019; Cappelli et al., 2024).

But these types are mostly based on categories of common sense and context-informed judgment. In our study, we offer structural explanations of increasing returns to increasing nested human capital by which we integrate these insights with structural understanding and thus provide structural reasoning for job mobility and disparity. In particular, as alluded to earlier, having a nested branch structure implies a trade-off in choosing one path over another, as their branches are nested under different sets of cores, and some skills are independent events and not dependent. Our exploration into network models reveals the potential for these structures to define conditional pathways of individual career trajectories. Therefore, it stands to reason that topology plays a role in specializations and the reward system, which our framework shows even without economic and social variables and without any presupposed or introspective judgment.

For example, while our findings align with the understanding that physical skills are less rewarded than cognitive ones, the novelty lies in revealing that their relations to other skills are shallow and deviate from

the nested structure. Again, this suggests that skill value may be determined by their embeddedness in the progression of specializations and complexity.

Broadly speaking, there are a few advantages of using network-based measures over the presupposed labels, such as blue- vs. white-collar, low- vs. high-skilled, physical vs. cognitive skills, or routine versus non-routine. For example, they systematically capture changes in the occupation-skill network topology over time. The literature has documented the concerning patterns of job polarization in the US economy (Autor & Dorn, 2013; Alabdulkareem et al., 2018; Althobaiti et al., 2022). While other work has offered statistics to document such changes, these all rely on how we categorize the skills between blue- vs. white-collar, low- vs. high-, physical vs. cognitive, but the nestedness approach does not rely on them, but the way they are conditional on each other and thus the structural change in the intensity of dependencies (i.e., the growth of nestedness), which describes the observed labor market polarization based on changes in the nature of work (the analysis of Fig. 6 documented in the Results subsection Widening Gap in the Skill Structures). This topological explanation is significant because the structure of the occupation-skill network constrains individual career paths, can increase disparity, carry macroscopic consequences, and has been shown to influence the system's resilience and stability (Allesina & Tang, 2012; Rohr et al., 2014; Saavedra et al., 2016; Moro et al., 2021).

Furthermore, our approach offers a scalable metric for skill categories, readily extending arguments to higher levels of granularity, which is what we mean by operationalizable. The nestedness approach captures structural changes in dependency intensity, providing an in-depth description of polarization. Data on skills applied in the workplace are increasingly more available at finer-grained levels. Skills' nestedness contribution readily extends our arguments to skills at higher levels of granularity by capturing skill heterogeneities in skills' differential nestedness contributions. In contrast, the results based on labels of skills may require follow-up analysis. For example, O*NET distinguishes between three types of Mathematical expertise (i.e., Mathematical Skills: Using mathematics to solve problems; Mathematical Knowledge: Knowledge of arithmetic, algebra, geometry, calculus, statistics, and their applications; and Mathematical Reasoning: The ability to choose the right mathematical methods or formulas to solve a problem). As the content of these skills change over time, a nestedness metric can tease out such changes when they translate into different occupational skill requirements, while an introspective may not do so as readily.

As for labeling and categories, for a lack of better terminology, we have continued using "nested" and "unnested" as they are contributing positively and negatively toward the total nested structure, respectively. We debated between using the labels and using continuous variables (c_s) as is. At the moment, we concluded that using labels with clear quantifiable definitions seems to be a simpler exposition.

The observation in Fig 4c, where controlling for general skills diminishes the positive correlation between wage and nested skills and reverses the negative correlation between wages and unnested skills, is consistent with our main findings that unnested skills are weakly dependent on general skills. This is because occupations with high levels of nested skills are commonly conditional on general skills but rarely unnested skills at high levels. The negative unconditional slope in Fig. 2.1.1a (below) indicates that high levels of unnested skills are often associated with lower-wage occupations. However, unnested skills are still a form of human capital, and in theory, a plumber with higher Repairing skill should earn more than a counterpart with lower Repairing skill, despite Repairing being an unnested skill. The negative slope in the unconditional setting arises because general skills, which strongly predict wages, are more intensely applied in occupations with primarily nested skills, aligning with our main message that general and nested skills are interdependent. As expected, once the influence of general skills on wages is accounted for in the conditional setting (Fig. 2.1.1b, below), the slope for unnested skills becomes positive. This regression analysis reveals the unexpected pattern that general skills, rather than (nested) niche skills, are the strongest predictors of wages.

Figure 3: Skill Nestedness Contributions Score. skills' nestedness score is highly indicative of their generality (a), risk of automation (b), and their value (c-d). Skill Nestedness Contributions is measured following [15]. Generality is measured by Local Reaching Centrality, as in Fig. 2, Automation risk Index and Value for each skill is calculated, following [65–67]. We divide skills into *nested*, positive contributions, and *un-nested*, negative contributions toward the nested skill structure.

Fig. 2.1.1. Wage regression on skill levels of nested and un-nested skills. a. Shows the regression in an unconditional setting, depicting the negative (positive) slope for un-nested (nested) skills. b. first residualized wages on general skills, accounting for their wage premiums, and then show the regression lines on the levels of nested and un-nested skills. The latter conditional setting shows the reversal of the slope of un-nested skills and the diminished slope of nested skills.

Acemoglu, D., & Restrepo, P. (2019). Automation and New Tasks: How Technology Displaces and Reinstates Labor. *Journal of Economic Perspectives*, 33(2), 3–30. <https://doi.org/10.1257/jep.33.2.3>.

Alabdulkareem, A., Frank, M. R., Sun, L., AlShebli, B., Hidalgo, C., & Rahwan, I. (2018). Unpacking the polarization of workplace skills. *Science Advances*, 4(7), 1–10. <https://doi.org/10.1126/sciadv.aao6030>.

Allesina, S., & Tang, S. (2012). Stability criteria for complex ecosystems. *Nature*, 483(7388), 205–208. <https://doi.org/10.1038/nature10832>.

Althobaiti, S., Alabdulkareem, A., Shen, J. H., Rahwan, I., Frank, M., Moro, E., & Rutherford, A. (2022). *Longitudinal Complex Dynamics of Labour Markets Reveal Increasing Polarisation*.

Autor, D. H. (2014). Skills, education, and the rise of earnings inequality among the “other 99 percent”. *Science*, 344(6186), 843–851.

Autor, D. H., & Dorn, D. (2013). The growth of low-skill service jobs and the polarization of the US Labor Market. *American Economic Review*, 103(5), 1553–1597. <https://doi.org/10.1257/aer.103.5.1553>.

Cappelli, P., Schwartz, S., & Yang, Y. (2024). College Attributes and Career Advancement. *Wharton School Department of Management Working Paper*.

Davidson, J. (1898). *The bargain theory of wages ...* G. P. Putnam’s Sons.

Deming, D. J. (2017). The Value of Soft Skills in the Labor Market. *NBER*. <https://www.nber.org/reporter/2017number4/value-soft-skills-labor-market#9>.

Deming, D. J., & Noray, K. (2020). Earnings Dynamics, Changing Job Skills, and STEM Careers. *The Quarterly Journal of Economics*, 135(4), 1965–2005.

Goldin, C., & Katz, L. F. (2008). The Race Between Technology & Education. In *Harvard University Press*.

Hong, I., Frank, M. R., Rahwan, I., Jung, W. S., & Youn, H. (2020). The universal pathway to innovative urban economies. *Science Advances*, 6(34), 1–7. <https://doi.org/10.1126/sciadv.aba4934>

Jones, B. F. (2009). The burden of knowledge and the “death of the renaissance man”: Is innovation getting harder?. *The Review of Economic Studies*, 76(1), 283–317.

Moro, E., Frank, M. R., Pentland, A., Rutherford, A., Cebrian, M., & Rahwan, I. (2021). Universal resilience patterns in labor markets. *Nature Communications*, 12(1), 1972. <https://doi.org/10.1038/s41467-021-22086-3>.

Murnane, R. J., Willett, J. B., & Levy, F. (1995). The Growing Importance of Cognitive Skills in Wage Determination. *The Review of Economics and Statistics*, 77(2), 251–266. <https://doi.org/10.2307/2109863>.

Rohr, R. P., Saavedra, S., & Bascompte, J. (2014). On the structural stability of mutualistic systems. *Science*, 345(6195), 1253497. <https://doi.org/10.1126/science.1253497>.

Rotundo, M., & Sackett, P. R. (2004). Specific versus general skills and abilities: A job level examination of relationships with wage. *Journal of Occupational and Organizational Psychology*, 77(2), 127–148.

Saavedra, S., Stouffer, D. B., Uzzi, B., & Bascompte, J. (2011). Strong contributors to network persistence are the most vulnerable to extinction. *Nature*, 478(7368), 233–235. <https://doi.org/10.1038/nature10433>

Saavedra, S., Rohr, R. P., Olesen, J. M., & Bascompte, J. (2016). Nested species interactions promote feasibility over stability during the assembly of a pollinator community. *Ecology and Evolution*, 6(4), 997–1007. <https://doi.org/https://doi.org/10.1002/ece3.1930>.

2.2. (2) Presentation

Overall I was not thrilled with how this material has been presented. The paper contains a lot of passages and phrases that seems needlessly complex or vague:

- "Modern economies... operate through globally interconnect networks. As economies become more complex, so do these networks..." (Are we discussing social networks? Trade networks? What are these sentences ultimately telling me? Could they be reduced to just "As economies become more complex, they coordinate more diverse portfolios of knowledge"?)

Thank you for bringing this issue to our attention. The introduction was a common concern among the reviewers, possibly due to our initial emphasis on economic complexity and network science leading to some tangential reasoning in the text. In response to your feedback, we've restructured the introduction to directly address the research question, aimed at driving home the importance of the skill dependencies structure.

- ## 2.3.
- "Nevertheless, the role of foundational skills for such ascent remains pivotal; without reinforcing them, the anticipated wage premiums may vanish." (Is this referring to a prior result, or are the authors referring to their own results on wages?)

Thank you for pointing out this confusion. Yes! This was one of our findings. To clarify this, we've now included "we find" in the sentence to indicate the origin of the statement. Additionally, we have revised other sections to clearly delineate which aspects are our findings. We greatly appreciate you bringing this to our attention.

- ## 2.4.
- "By differentiating general human capital into a structured spectrum, from the most foundational and general to the most specialized, human capital are comparable at different scales of organizations, which is essential for policy implications." (What does "comparable at different scales of organizations" mean?)

- "The hierarchical structure and its inherent directionalities add a new dimension to the rising field of economic complexity, providing a deeper understanding of how knowledge is accumulated within a population and how it is expressed in the economic activities of a firm, city, region, or country." (This paper is not about adding "hierarchy" and "directionality" in general to economic complexity. This feels over-wrought. The paragraph it appears within generally seems like it could be reduced to a sentence or two.)

- The comparison to ecosystems is possibly interesting but again too vague for me to gain anything from it. These examples focus on individually confusing sentences but the problem is more pervasive; the text could use a work over. The abstract is particularly convoluted. I spent much more time parsing the paper than should be necessary. The authors have interesting results and they could present them much more simply.

Thank you for bringing these confusions to our attention. Upon further reflection, we agree that our writing was overly complex, which likely obscured the message we intended to convey. Our goal was to emphasize the implications of our findings for understanding human capital in various organizational contexts (first

point) and to discuss how economic complexity, focusing on capabilities and the concept of a directed capability network, extends beyond just labor (second point), and through nested hierarchical skill structure in human capital (third point). However, we recognize that we did not communicate this effectively, leading to potential confusion in the discussion. In addition, human capital management seems a bit tangential to our findings. As a result, we've decided to remove some of these intricate and unnecessary text, and rewrite our reasoning behind the nested hierarchy in human capital. Please refer to the response above for these elaborations.

- 2.5. The paper comes with a huge SI - 90 pages - that is difficult to map to the main text. References to the SI are a bit chaotic; SI sections/figures/tables are not arranged in an obvious way (e.g., sequentially) where these components are needed, and quite a number of figures do not seem to be referenced at all. The SI doesn't compensate with its own separate intro or summary to offer guidance. The result is that it is very hard for anyone to practically review all that has been done or for interested specialists to find things they are curious about. It also triggers the worry that some SI analyses are really essential for the core arguments and may be more appropriate for the Methods section of the main text.

We are thankful to Reviewer 2 for bringing this issue to our attention. In compiling the Supplementary Information (SI), our aim was to be comprehensive, incorporating a variety of analyses, results, and intermediate findings, including crosswalks, to enable a detailed review of our work. As noted by Reviewer 2, the extensive supplementary material primarily serves specialists in our field. Nonetheless, we acknowledge that the considerable volume of the SI may inadvertently obscure our commitment to transparency and thoroughness. To address this, we have streamlined the SI, starting with an introductory section that quickly guides the reader through its contents in less than a page.

Despite the revisions, we believe it is crucial to retain the detailed analyses within the SI. Our experience has shown that extensive supplementary materials are often vital for enabling students and researchers to replicate and extend findings. Our goal is to create a resource that future researchers can use to replicate our results. As a navigational aid, we've included a table of contents (below) that outlines the technical sections, such as parameters, thresholds, robustness tests, and the logic behind them.

This doesn't mean the redundancy is acceptable. Therefore, we went through SI to make it more concise than before. We hope the changes made facilitates an examination of our work by the interested readers.

Here are content:

1. Skill Groups: we explain different grouping algorithms and parameters and provide robustness tests.
2. Skill Nestedness: we explain how the nested structure is constructed from raw data to the network structure. We provide complications in the data and parameters/thresholds and offer robustness tests.
3. Skill Categories in Career Trajectories: we explain the data structure of the raw resume data and survey data, and how to slice and weave them together to construct longitudinal data structure.
4. Skill Investment and Returns: we explain how to estimate wages and educational variables in different data sources.
5. Geographic distributions: We describe the analysis of geographic variation and the data sources we used to do so.
6. Demographic distributions: We describe the construction and use of demographic variables and robustness tests using different parameters.
7. Historical patterns: We explain how we match two datasets as they use slightly different occupation categories. We explain here how we matched them.
8. Robustness Checks with Management and Admin Occupations and Social Skills include all the other variables in the literature to ensure our findings are robust to using different categorizations.

- 2.6. It is a little hard to find information about the datasets. It is there but dispersed. It would help to have a clearly-labeled places in the main text where the 3 datasets / data-derived objects are

covered:

- The database of occupations based on US BLS surveys 2005-2019.
- The pseudo data of cohorts constructed with data from the Current Population Survey.
- The corpus of resumes from Burning Glass (70 million job transitions).

This was an excellent suggestion. Thank you so much. In response, we have compiled and aggregated the information about datasets under a separate subsection "Datasets", where we described the sources and broad information about the collection and content of each dataset. The following subsections in the Data and Method describe how we derived measures from the information in each dataset.

3. Reviewer #3

The study presented in this paper is intriguing and utilizes a unique dataset of 70 million CVs and career transitions. The methodological approach employed in the research is well-developed, and the derivation of relevant metrics is executed with precision. Furthermore, the concept of nested and un-nested skills is a novel and insightful perspective on human capital, and the organization of skills in a directed network based on their pair-wise dependencies is a powerful approach with significant implications for this and future investigations.

However, I am compelled to advocate for a major revision before this work can be considered for publication.

Thank you for dedicating your time to providing constructive feedback on our manuscript. We greatly appreciate your efforts in reviewing our work and acknowledging the merits of our research, especially your insightful suggestions which have presented us with a valuable chance to improve the manuscript's quality.

- 3.1. My primary criticism is the absence of a clear theoretical research question that is addressed by the findings of this study. The results explore various distributional issues related to skills across geographical locations, gender disparities, and life trajectories, but none of these insights seem to be driven by a theory-based research question. The paper reads more like a methodologically-savvy data expedition, covering diverse societal angles without offering a conclusive framework to close gaps in existing theory.

One potential way to address this deficiency would be to develop a research question grounded in theory. For instance, the paper could investigate whether earnings differentials between men and women can be explained by systematic differences in the distribution of nested versus un-nested skills. The data presented in Figure 3 could prompt an exploration into whether women experience disruptions in the accumulation of nested skills, possibly due to motherhood, and whether these imbalances subsequently contribute to wage differences. This would not only provide a stronger theoretical foundation but also enhance the relevance and impact of the research.

We're grateful for your constructive feedback and the insightful recommendations you provided. Upon revisiting our manuscript, it became clear that our initial exposition of the methodological and empirical frameworks lacked clarity and depth, particularly regarding the definition of key concepts and the explanation of their usage and underlying theories. Such shortcomings may have led to the manuscript being seen merely as an assortment of methodological and empirical observations lacking in theoretical depth. To address these concerns, we have rewritten the entire manuscript, adding a new section with figures from additional analyses to elaborate on our conceptualization of skill distributions and conditional probabilities. This new section is dedicated to expanding upon the theoretical underpinnings of our research as well as empirical justification, providing a clearer explanation of our approach and its significance within a broader theoretical framework. Our aim with these revisions has been to integrate our findings more cohesively with existing scholarship and to highlight our study's contributions to the academic discourse. Through this process, we hope to present our work not just as empirical findings but as a meaningful theoretical advancement in economic complexity.

More specifically, we have made the following changes:

- Articulated the conceptualization of skill dependency and the rationale for using nested structures and defined nested/unnested skills based on their structural contributions to the skill structure's overarching nested architecture.

- Created a new result section (Section 3: nested contribution) with a new figure (Fig 3, which we included at the end of this response, see below), dedicated to the nested skill structure, to spell out our conceptual framework and methodology.
- Added theoretical implications of having nested structures in human capital.
- Elaborated on theoretical constructs of skill application and acquisition and their implications in the nested structure. In particular, we provide additional analyses with case studies to support our theoretical/methodological constructs, demonstrating how our framework based on the conditional probability of skill distributions, observed in occupational attributes, infers individuals' acquisitions, which are unobserved workers' attributes (SI Sec. 3.4-3.5).
- Articulated theoretical contributions of structural analysis to existing economic and social theories.
- Conducted follow-up inquiries to reveal the intriguing skill divergence observed between men and women, examining the role of parenthood and work schedule (Parenthood Analysis and Job Sorting and Work Schedule Analysis). Our framework captures nuances that conventional approaches, such as metrics based on education or existing direction-agnostic skill networks, would not have revealed (SI Secs. 7.1-7.2), but in line with prior findings that a noticeable extent of life-long learning occurs post-school (Heckman et al., 1998), and perhaps surprisingly, reveal that comparable skill development patterns emerge for workers without college degrees. The latter analysis is collected in SI Sec. 4.3., and partially presented below (*Skill Development and Education Analysis*). Our SI Table S5 shows that each subset of our skill categories explains occupational wages equally as well as education and withstands other controls, such as experience and training.

For your convenience, we elaborate a few key points in the revision:

We first acknowledge that our paper is indeed predominantly grounded in empirical and methodological analysis rather than theoretical exploration. This was a reason for us to aim for a journal with an empirical focus and broad readership. Our approach places our work alongside studies by Jia et al. (2017), Alabdulkareem et al. (2018), Yun et al. (2019), Wu et al. (2019); and Moro et al. (2021a, 2021b), McNerney et al. (2022), which primarily focus on sharing methodological and quantitative assessments, with a secondary focus on theoretical development.

However, we also recognize the critical importance, as pointed out by Reviewer 3, of contributing to and developing a theoretical foundation that goes beyond methodological and empirical findings. This involves not only bridging the gap between theoretical frameworks and empirical evidence but also challenging the conventional view of human capital theory, which often relies on context-informed judgments and assumes a linear and uniform approach to skill investment. In this context, our contribution is the development of a structural framework that re-conceptualizes human capital through a quantitative lens, revealing a nested hierarchical structure that reframes the discourse on skill investment and economic returns. We believe this nested structure not only presents a new perspective on the growing returns to certain skills, but also challenges the traditional human capital theory that assumes a linear and singular investment approach. In this revision, we elaborate on the theoretical implications of the nested skill structure, demonstrating how it can account for the observed patterns of career progression and wage premiums.

More specifically, in the economic and sociological literature, skills and knowledge are often categorized using context-oriented and ad hoc labels, arranging them into simplistic, one-dimensional categories such as blue-collar vs. white-collar, low-skilled vs. high-skilled, or physical vs. cognitive. This tendency persists even in discussions that recognize the value of skills (Mincer, 1958; Card, 1994) and addresses challenges like the skill hold-up problem (Klein et al., 1978; Rogerson, 1992; Balmaceda, 2005; Kessler & Lülfesmann, 2006), critiquing the underinvestment in general human capital (Bovenberg & Jacobs, 2005; Gathmann & Schönberg, 2010), and conceptually distinguish between general and specialized skills (Becker, 1962;

Poletaev & Robinson, 2008; Gathmann & Schönberg, 2010). This approach remains prevalent in contemporary literature, including recent works by Rotundo and Sackett (2004), Acemoglu and Restrepo (2019), and Cappelli et al. (2024).

In addition, much work in economics, sociology, and management has examined changes in the payoffs to different types of skills in the labor market. Thus, identifying relevant types, such as social skills and technical skills, and the expected effects from our insights is crucial for these studies. To cite a few examples, Deming and Noray (2020) have suggested the importance of technical skills in facilitating the transition of college graduates into the labor market; Deming (2017) shows increasing returns to social skills by comparing workers who entered the labor market in the 1980s and early 1990s with those who did so in the early 2000s; and Murnane, Willett, and Levy (1995) find that the increase in returns to cognitive skills from 1978 to 1986 contributed significantly to the rising wage premium associated with higher education. While these studies focus on individual skills, they do not provide an overall framework to explain the value of different skills in today's labor market. We address this discussion through the lens of nested structure in skill dependency.. Increases in the nestedness over time can describe payoffs to cognitive and technical skills that are primarily nested, as well as the growing importance of social skills (which belong to the general skills category). Although we acknowledge that we do not provide the ultimate causation for this trend, we offer a possible reasoning for it, which is at the more inquisitorial stage, far from complete. Nevertheless, to our knowledge, our finding is the first to show that the values of skills can be attributed to the way skills are structured and embedded in the larger structure. This knowledge further reshapes theoretical and empirical developments on skill holdup and policy on instilling general human capital (Klein et al., 1978; Rogerson, 1992; Balmaceda, 2005; Bovenberg & Jacobs, 2005; Kessler & Lulfesmann, 2006; Stantcheva, 2017).

Our framework, in which the value and properties strictly arise from the topology of the skill network, might initially seem unorthodox and unconventional in social science given its detachment from traditional context-informed analysis, such as the consideration of socioeconomic factors like wages and education or the presumption of a skill acquisition and application model. Instead, we simply operationalize this conceptual shift through a methodological framework that (1) measures skill distributions across occupations (narrow vs. broad), (2) identifies the existence of asymmetric conditional probabilities in skills' cross-group, and (3) quantifies a nested contribution score where nested (unnested) skills are defined as those positively (negatively) contributing to the overarching nestedness in human capital with $c_s > 0$ ($c_s < 0$). In the new section with Fig 3, we show these values c_s , which only use information on the network topology, predict wages and education well, suggesting that the required skill set has its own structure, and higher-value skills are only accessible after prerequisite skills have been acquired or utilized. This has societal consequences, including social mobility and labor disparity.

This analysis relies on computational, statistical, and mathematical insights, devoid of direct socioeconomic interpretations, positioning our paper as suitable for an interdisciplinary journal, as we want to get our methodological applications and empirical statistics right first. Our further explorations into demographic disparities serve as proof of concepts, indicating the path for future research to fully understand the implications of our findings. Obviously we need more subsequent studies to truly grasp what we find.

The structural explanation goes beyond the conceptualization and measurement of workplace skills. Our approach offers scalability in conceptualizing skills, capabilities, and tools, readily extending arguments to higher levels of granularity, building toward the future of work literature. Skills' nestedness contribution captures skill heterogeneity's structural changes in dependency and provides an in-depth description of polarization.

Of course, this structural perspective and its merit are not entirely new, but align with complexity theory. Complexity theory examines products, capabilities, and skills as network structures (Anderson, 2017,

Alabdulkareem et al., 2018., Neffke, 2019, del Rio-Chanona et al., 2021, Althobaiti et al., 2022). Indeed, understanding the network architecture in complex economic systems—spanning technology, input-output, supply-chain, trade, products, and skills—has yielded insights into socio-economic phenomena. These insights both corroborate and contest established theoretical frameworks, including the underlying causes of economic disparities between countries and their potential developmental trajectories by analyzing trade networks; the pace of technological innovation and economic growth through technology networks; differences in labor productivity and resilience through the lens of skill and occupation networks and economic network resistance and persistence using business network. These models and methods translate a range of structural properties into quantifiable and actionable insights

Therefore, our empirical study aims to add a new layer to these structural properties, illustrating how connections within these networks are directed and how structures become increasingly nested as complexity and specialization grow. By bridging the divide between traditional economic models and complexity theory, our study contributes to viewing skill development, application, and the structural underpinnings of labor market dynamics as the growth of nestedness. This topological explanation is useful because the structure of the occupation-skill network constrains individual career paths, can increase disparity, carry macroscopic consequences, and influence the system's resilience and stability (Allesina & Tang, 2012; Rohr et al., 2014; Saavedra et al., 2016; Moro et al., 2021a).

More broadly, our theoretical contribution is to shift research attention towards understanding human capital through the lens of structural dependencies and their implied dynamics, laid upon individuals as possible choices conditional on previous choices, shaped by the structure of society that grows with complexity and specializations. Therefore, our interpretation of human capital transcends methodological preferences, employing conditional probabilities of skill distribution and nestedness scores as illustrative tools rather than definitive measures, perhaps addressing career contingency: one's future career options may be broadly influenced by the skills required in their previous positions. In that direction, Our interpretation of human capital transcends methodological preferences, employing conditional probabilities of skill distribution and nestedness scores as illustrative tools rather than definitive measures. We acknowledge that hierarchical dependencies within human capital can be depicted through various methodologies and empirics, not strictly tied to the ones we've chosen. Our ambition is to explore this framework's viability, admitting that while our method may not be flawless, it offers a pragmatic approach given the constraints of our data and analytical scope, emphasizing the importance of moving beyond conventional metrics like education to examine the structural underpinnings of skill dependencies.

To showcase how our approach captures specialization paths, we conducted additional case studies, (i) comparing registered nurses with nurse practitioners, which highlight how our skill hierarchies capture deepening branches of skills as individuals progress (SI Sec. 3.4), and (ii) revealing how language deficiencies across Hispanic immigrants mount barriers to dependent downstream specialized skills, causing skill entrapment in unnested human capital (SI Sec. 3.5). By doing so, we propose our framework as a theoretical springboard for further investigations into how skills are developed, applied, and valued within the complex fabric of modern society.

We reiterate that our study's value doesn't depend on its success in one narrow explanation but on the structural explanation for these socioeconomic attributes and their structural implications. Although our methodological question is, "Can we infer skill hierarchies from just the topology of the co-occurrence network?" and shows feasibility, our research question is, "Skill dependencies are important structural properties that offer profound insights into the nature of human capital, insights that have been largely recognized in discussion but not empirically quantified until now. Our findings indicate that the structural properties of skill dependencies can explain wage and education outcomes without relying on conventional socioeconomic labels/content of skills (mostly coming from common sense or introspective judgments).

Finally, we are receptive to your suggestion for a more in-depth analysis of gender differences in developing and applying skills. We have conducted follow-up inquiries to reveal the intriguing diverging skill patterns observed between men and women and their timelines, examining the role of parenthood and work schedules. Our approach captures nuances that conventional approaches, such as metrics based on education or a direction-agnostic view of skill networks, would not have revealed. Our approach's descriptive capacity extends beyond the demographic to the geographic distribution of skills, where we find our skill categorization can account for a significant part of urban wage premiums (Glaeser, 1999; Wheeler, 2001; Rosenthal & Strange, 2004; Combes et al., 2010; Gomez-Lievano & Patterson-Lomba, 2021). We also corroborate prior findings that a noticeable extent of life-long learning occurs post-school (Heckman et al., 1998), and surprisingly, comparable skill development patterns emerge for workers without college degrees (SI Sec. 4.3, Skill Development and Education Analysis). SI Table S5 shows that each subset of our skill categories explains occupational wages equally as well as education and withstands other controls, such as experience and training.

Below we provide key findings in the follow-up analysis for your convenience.

Figure 3: Skill Nestedness Contributions Score.: skills' nestedness score is highly indicative of their generality (a), risk of automation (b), and their value (c-d). Skill Nestedness Contributions is measured following [15]. Generality is measured by Local Reaching Centrality, as in Fig. 2, Automation risk Index and Value for each skill is calculated, following [65–67]. We divide skills into *nested*, positive contributions, and *un-nested*, negative contributions toward the nested skill structure.

1. Parenthood

Analysis:

We are intrigued by what seems to be a substantial influence of children on individuals' careers, given the age in which diverging patterns of skill development appear between men and women

(initial manuscript Fig. 3). Pursuant to your suggestion and similar remarks from other reviewers; we have found information in the Current Population Survey (CPS) that would allow us to investigate the role of children explicitly. This analysis, now presented in SI Sec. 7.1. and succinctly captured in Fig. S49 below, reveals that parenthood indeed has a differential influence on the careers of men and women. Compared to women without children, mothers experience subpar growth of general and nested skills. In contrast, compared to men without children, fathers have faster growth of skills, perhaps due to an elevated need to earn a sufficient income. The contrasting patterns for men and women with children could also emerge as a result of a domestic division of labor in which men have been more likely to take up the role of economic provider.

The implementation of the analysis is as follows: for the entire period of our sample, CPS records the number of children in the surveyed households. Using this information, we split the sample to individuals with and without children and replicate the analysis of synthetic birth cohorts (previous Fig. 3d-f and the new Fig. 4d-f): we track the skills manifested in the occupational compositions of birth cohorts as they age, splitting individuals based on their binary gender (Male: lower panels; Female: upper panels of the below figure) and their binary parental status (with child: square; without child: triangle in the below figure) at the time of survey. Fig. S49 captures the result of aggregating skills for the named subgroups. Each column shows the levels of a certain category of skills, while the rows show the results for men and women. The solid line (and triangles) show the pattern for people without children, while the dashed line (and squares) show the pattern for individuals with children.

Fig. S49. In the figure, we track the skills manifested in the occupational compositions of birth cohorts as they age, splitting individuals based on their binary gender (Male/Female) and whether they lived with children at the time of the survey, obtaining the below figure. Each column shows the levels of a certain category of skills, while the rows show the results for men and women. The solid line (and triangles) show the pattern for people without children, while the dashed line (and squares) show the pattern for individuals with children. There is a pronounced drop in the general and nested skills for women with children.

2. Job Sorting and Work Schedule Analysis:

We further inquired about the diverging patterns of skill development between men and women, wherein women exhibit high levels of general skills, surpassing their male counterparts at certain ages (as shown in previous Fig. 3d-f and the new Fig. 4d-f) but do not manifest the high levels of nested skills observed for male workers of the same age. Lower levels of nested skills for women is also seen in the first column of the table S7, below, that predicts the gender of workers based on their general and nested skills in our CPS sample (Female = 1): general skills are associated with greater, but nested skills with smaller shares of women in an occupation.

Indeed, a comprehensive treatment of this observation merits a project in its own right. However, one explanation for this pattern, at the center of the nobel-prize winning work of Claudia Goldin, is that women may avoid jobs with irregular or long working schedules (Bertrand et al., 2009, Goldin, 2015, Canon & Golan, 2016). Hence, despite their high levels of general skills and education, women may avoid jobs that require nested skills because of the working conditions of such jobs. To examine that hypothesis, we examined whether adding descriptors of work schedules to the same regression diminishes the correlation between skills and the gender of the worker, as reported in column 1 below. This analysis, described in detail in the revised SI Sec. 7.2., produced two proxies for irregular working schedules (higher denotes more irregular) and long working hours (a binary variable that is 1 if the occupation requires more than 50 hours of weekly work, following Cha et al. (2014)) from O*NET. To test how work schedule correlates with the gender of workers, we matched individuals in the Current Population Survey (CPS) aged 18 to 55 who were in the workforce between 1980 and 2020 to the descriptors of irregular schedule or long hours and included the result in the second column of table S7, below. The result shows that, indeed, irregular work schedules and long working hours account for part of the skill-gender correlations estimated in the baseline. Per the baseline model of column 1, a unit increase in the nested specific skills required by a job decreases the chances of the worker being female by ~36%. Adding schedule descriptors reduces that relation by more than one-third, to about 26%.

Skills, Work Schedule, and Gender		
	Dependent variable:	
	Gender Dummy (Female = 1)	
	OLS	
	(1)	(2)
General Skills	0.203*** (0.201, 0.204)	0.150*** (0.148, 0.153)
Nested Skills	-0.357*** (-0.359, -0.355)	-0.258*** (-0.261, -0.256)
Irregular Schedule		-0.338*** (-0.342, -0.334)
Long Hours Dummy (>50 weekly)		-0.176*** (-0.178, -0.174)
Constant	0.097*** (0.092, 0.101)	0.629*** (0.620, 0.638)
Observations	1,493,142	1,096,362
R ²	0.072	0.108
Adjusted R ²	0.072	0.108
Residual Std. Error	0.463 (df = 1493139)	0.455 (df = 1096357)
F Statistic	57,942.160*** (df = 2; 1493139)	33,058.290*** (df = 4; 1096357)
Note:		* p<0.1; ** p<0.05; *** p<0.01

Table S7. Regression analysis of the correlation between gender, skills, and irregular and long work schedules. The first column offers a baseline model that predicts the gender (Female = 1) of the

worker based on general and nested skills, showing a negative correlation with nested skills. Adding descriptors of irregular and long schedules in the second model explains away part of the predictive power of nested skills for workers' gender. As such, part of the reason why women manifest high levels of general skills but comparatively low levels of nested skills is that jobs that require the latter categories of skills likely impose long and irregular work conditions, which have been found to deter female workers.

3. Skill Development and Education Analysis:

While there is no doubt about the importance of education as a source of skill acquisition and as a proxy for measuring skills, there is much to examine about the dynamics of skills and wages beyond education. Figs. S49, below, shows evidence consistent with that claim.

In Fig. S49 we replicate the evolution of skill and age depicted in the initial manuscript Fig. 3 d-f, in addition to the education attainment and the fraction of individuals who attend school as functions of age directly taken from the Current Population Survey (CPS). The education attainment variable ranges from 02 (i.e., no schooling) to 125 (i.e., doctorate degree). To obtain the fraction of the sample attending school, we utilized the information in CPS variable SCHOOLCOL that documents attending high school (1 or 2) or college/university (3 or 4) or not attending school (5). We transformed the information so that if an individual attends school (1,2,3, or 4), it receives a value of 1, and if not attending, it has a value of 0. Even though by the age of 30, education plateaus and school attendance drops significantly, skill growth continues, manifesting the presence of other mechanisms for skill accumulation apart from education.

Figure S28: Evolution of skill, age and education. To measure education, we have used educational attainment and the fraction of individuals who attend school as functions of age, both taken from the Current Population Survey (CPS). The education attainment variable ranges from 2 (i.e., no schooling) to 125 (i.e., doctorate degree). To obtain the fraction of the sample attending school, we utilized the information in CPS variable SCHOOLCOL that documents attending high school (1 or 2) or college/university (3 or 4) or not attending school (5). We transformed the information so that if an individual attends school (1,2,3, or 4), it receives a value of 1, and if not attending, it has a value of 0. Even though by the age of 30 education plateaus and school attendance drops significantly, skill growth continues, manifesting the presence of other mechanisms for skill accumulation apart from education.

Acemoglu, D., & Restrepo, P. (2019). Automation and New Tasks: How Technology Displaces and Reinstates Labor. *Journal of Economic Perspectives*, 33(2), 3–30. <https://doi.org/10.1257/jep.33.2.3>.

Alabdulkareem, A., Frank, M. R., Sun, L., AlShebli, B., Hidalgo, C., & Rahwan, I. (2018). Unpacking the polarization of workplace skills. *Science Advances*, 4(7), 1–10. <https://doi.org/10.1126/sciadv.aao6030>.

Allesina, S., & Tang, S. (2012). Stability criteria for complex ecosystems. *Nature*, 483(7388), 205–208. <https://doi.org/10.1038/nature10832>.

Althobaiti, S., Alabdulkareem, A., Shen, J. H., Rahwan, I., Frank, M., Moro, E., & Rutherford, A. (2022). *Longitudinal Complex Dynamics of Labour Markets Reveal Increasing Polarisation*.

Anderson, K. A. (2017). Skill networks and measures of complex human capital. *Proceedings of the National Academy of Sciences*, 114(48), 12720–12724.

Balmaceda, F. (2005). Firm-Sponsored General Training. *Journal of Labor Economics*, 23(1), 115–133. <https://doi.org/10.1086/425435>

Becker, G. S. (1962). Investment in Human Capital : A Theoretical Analysis. *Journal of Political Economy*, 70(5), 9–49.

Bertrand, M., Goldin, C., & Katz, L. F. (2009). *Dynamics of the gender gap for young professionals in the corporate and financial sectors.*

Bovenberg, A. L., & Jacobs, B. (2005). Redistribution and education subsidies are Siamese twins. *Journal of Public Economics*, 89(11), 2005–2035. <https://doi.org/https://doi.org/10.1016/j.jpubeco.2004.12.004>

Canon, M., & Golan, L. (2016). Gender Pay Gap May Be Linked to Flexible and Irregular Hours. *The Regional Economist*, July.

Card, D. (1994). Earnings, Schooling, and Ability Revisited. *Research in Labor Economics*, 14. [https://doi.org/10.1108/S0147-9121\(2012\)0000035031](https://doi.org/10.1108/S0147-9121(2012)0000035031)

Cappelli, P., Schwartz, S., & Yang, Y. (2024). College Attributes and Career Advancement. *Wharton School Department of Management Working Paper.*

Cha, Y., & Weeden, K. A. (2014). Overwork and the slow convergence in the gender gap in wages. *American Sociological Review*, 79(3), 457–484.

Combes, P.-P., Duranton, G., Gobillon, L., & Roux, S. (2010). *Estimating Agglomeration Economies with History, Geology, and Worker Effects* (pp. 15–66). National Bureau of Economic Research, Inc.

del Rio-Chanona, R. M., Mealy, P., Beguerisse-Díaz, M., Lafond, F., & Farmer, J. D. (2021). Occupational mobility and automation: a data-driven network model. *Journal of The Royal Society Interface*, 18(174), 20200898. <https://doi.org/10.1098/rsif.2020.0898>.

Deming, D. J. (2017). The Value of Soft Skills in the Labor Market. *NBER*. <https://www.nber.org/reporter/2017number4/value-soft-skills-labor-market#9>.

Deming, D. J., & Noray, K. (2020). Earnings Dynamics, Changing Job Skills, and STEM Careers. *The Quarterly Journal of Economics*, 135(4), 1965–2005.

Gathmann, C. & Schönberg, U. How general is human capital? A task-based approach. *Journal of Labor Economics* 28, 1–49 (2010).

Glaeser, E. L. (1999). Learning in Cities. *Journal of Urban Economics*, 46(2), 254–277. <https://doi.org/https://doi.org/10.1006/juec.1998.2121>.

Goldin, C. (2015). Hours flexibility and the gender gap in pay. *Center for American Progress*, 31.

Gomez-Lievano, A., & Patterson-Lomba, O. (2021). Estimating the drivers of urban economic complexity and their connection to economic performance. *Royal Society Open Science*, 8(9), 210670. <https://doi.org/10.1098/rsos.210670>.

Heckman, J. J., Lochner, L., & Taber, C. (1998). Tax Policy and Human-Capital Formation. *The American Economic Review*, 88(2), 293–297. <http://www.jstor.org/stable/116936>

Jia, T., Wang, D., & Szymanski, B. K. (2017). Quantifying patterns of research-interest evolution. *Nature Human Behaviour*, 1(4), 1–7. <https://doi.org/10.1038/s41562-017-0078>

Kessler, A. S., & Lülfsmann, C. (2006). The Theory of Human Capital Revisited: On the Interaction of General and Specific Investments. *The Economic Journal*, 116(514), 903–923. <https://doi.org/10.1111/j.1468-0297.2006.01116.x>

Klein, B., Crawford, R. G., & Alchian, A. A. (1978). Vertical Integration, Appropriable Rents, and the Competitive Contracting Process. *The Journal of Law & Economics*, 21(2), 297–326. <http://www.jstor.org/stable/725234>

McNerney, J., Savoie, C., Caravelli, F., Carvalho, V. M., & Doyne Farmer, J. (2022). How production networks amplify economic growth. *Proceedings of the National Academy of Sciences of the United States of America*, 119(1). <https://doi.org/10.1073/pnas.2106031118>

Mincer, J. (1958). Investment in Human Capital and Personal Income Distribution. *Journal of Political Economy*, 66(4), 281–302. <http://www.jstor.org/stable/1827422>

Moro, E., Frank, M. R., Pentland, A., Rutherford, A., Cebrian, M., & Rahwan, I. (2021a). Universal resilience patterns in labor markets. *Nature Communications*, 12(1), 1972. <https://doi.org/10.1038/s41467-021-22086-3>.

Moro, E., Calacci, D., Dong, X., & Pentland, A. (2021b). Mobility patterns are associated with experienced income segregation in large US cities. *Nature Communications*, 12(1). <https://doi.org/10.1038/s41467-021-24899-8>

Murnane, R. J., Willett, J. B., & Levy, F. (1995). The Growing Importance of Cognitive Skills in Wage Determination. *The Review of Economics and Statistics*, 77(2), 251–266. <https://doi.org/10.2307/2109863>.

Neffke, F. M. H. (2019). The value of complementary co-workers. *Science Advances*, 5(12). <https://doi.org/10.1126/sciadv.aax3370>.

Poletaev, M., & Robinson, C. (2008). Human capital specificity: evidence from the Dictionary of Occupational Titles and Displaced Worker Surveys, 1984–2000. *Journal of Labor Economics*, 26(3), 387–420.

Rogerson, W. P. (1992). Contractual Solutions to the Hold-Up Problem. *The Review of Economic Studies*, 59(4), 777–793. <https://doi.org/10.2307/2297997>

Rohr, R. P., Saavedra, S., & Bascompte, J. (2014). On the structural stability of mutualistic systems. *Science*, 345(6195), 1253497. <https://doi.org/10.1126/science.1253497>.

Rosenthal, S. S., & Strange, W. C. (2004). Chapter 49 - Evidence on the Nature and Sources of Agglomeration Economies. In J. V. Henderson & J.-F. B. T.-H. of R. and U. E. Thisse (Eds.), *Cities and Geography* (Vol. 4, pp. 2119–2171). Elsevier. [https://doi.org/https://doi.org/10.1016/S1574-0080\(04\)80006-3](https://doi.org/https://doi.org/10.1016/S1574-0080(04)80006-3).

Rotundo, M., & Sackett, P. R. (2004). Specific versus general skills and abilities: A job level examination of relationships with wage. *Journal of Occupational and Organizational Psychology*, 77(2), 127–148.

Saavedra, S., Rohr, R. P., Olesen, J. M., & Bascompte, J. (2016). Nested species interactions promote feasibility over stability during the assembly of a pollinator community. *Ecology and Evolution*, 6(4), 997–1007. <https://doi.org/https://doi.org/10.1002/ece3.1930>.

Stantcheva, S. (2017). Optimal Taxation and Human Capital Policies over the Life Cycle. *Journal of Political Economy*, 125(6), 1931–1990. <https://doi.org/10.1086/694291>

Wheeler, C. H. (2001). Search, Sorting, and Urban Agglomeration. *Journal of Labor Economics*, 19(4), 879–899. <https://doi.org/10.1086/322823>.

Wu, L., Wang, D., & Evans, J. A. (2019). Large teams develop and small teams disrupt science and technology. *Nature*, 566(7744), 378–382. <https://doi.org/10.1038/s41586-019-0941-9>

Yun, J., Lee, S. H., & Jeong, H. (2019). Early onset of structural inequality in the formation of collaborative knowledge in all Wikimedia projects. *Nature Human Behaviour*, 3(2), 155–163.

- 3.2. As a minor point, I noted that some findings related to skill nestedness and wages appear to overlap with a previous study (<https://doi.org/10.101/j.respol.2023.104898>) that is not cited in the present work.

Thank you for directing us to a reference related to our topic. Unfortunately, the link provided seems to be inactive. The closest match we encountered was Stephany & Teutloff study (2024) with the DOI <https://doi.org/10.1016/j.respol.2023.104898> with an additional digit "6". Recognizing the relevancy to our paper, we've taken the liberty to include the reference in the updated version of our manuscript. Should this reference not match the one you intended, we would greatly appreciate additional details to ensure accuracy. It's noteworthy to mention that the Stephany & Teutloff study was published after our original submission (our paper has been reviewed unusually long, for 6 months, as we submitted July 2023—a time in which the field has undoubtedly evolved). To reflect this progression in our revised manuscript, we have duly incorporated this relevant content into our revised manuscript. However, if this does not align with the intended reference, we kindly request further clarification.

Stephany, F., & Teutloff, O. (2024). What is the price of a skill? The value of complementarity. *Research Policy*, 53(1), 104898. <https://doi.org/https://doi.org/10.1016/j.respol.2023.104898>

- 3.3. In conclusion, while "Nested Skills in Labor Ecosystems: A Hidden Dimension of Human Capital" is an engaging and methodologically sound study with novel concepts and an impressive dataset, it is crucial to address the lack of a clear theoretical research question. The addition of a well-defined theoretical framework and the exploration of questions such as gender disparities in skill accumulation could significantly strengthen this work.

Thank you so much for the encouragement, constructive feedback, and insightful suggestions. Pushing us to make a more theoretical contribution has been immensely helpful for us to articulate our work and make good progress to improve the quality of the research. We are very much excited about our new research and cannot wait to publish it to shift our perspective of human capital and economic complexity!

4. Reviewer #4

This is a thoughtful paper taking on a creative question: how do skills co-occur? They take advantage of a novel idea, which is the direction of conditional probabilities, to assess which skills are more general and important. They document that the value of general skills has not diminished, and higher wages are predicted by higher specialization. They document large differences in this skill specialization across race, location, and gender.

I think the methods are solid and presents a novel contribution to nature human behavior. However, there are a lot of limitations in what is claimed and what is documented.

Thank you for taking the time to review our work and recognizing the merits of our research. Your insightful feedback and critiques have been invaluable. In the responses that follow, we list how your guidance has sharpened our thinking and improved the overall quality. We acknowledge the limitations you have pointed out and have addressed them accordingly. In response to your feedback, we have elaborated on the foundations and premises of our research, explained the theoretical background of our use of nestedness, and included follow-up analyses in the revision. Specifically, we rewrote our introduction (response 4.1 below) and detailed our assumptions and theoretical background and layout limitations (response 4.2). In fact, in response to your feedback, we have rewritten the entire manuscript to better articulate our conceptualization (response 4.3), created an additional section with additional analysis to clarify our measures of nestedness, and put them in the context of human capital (responses 4.4 and 4.6). Moreover, we tidy up the paper by removing unnecessary tangential results (response 4.5), straightening our analyses (response 4.8), and clarifying our references and theoretical contribution (response 4.9).

This process has been incredibly valuable for us to sharpen our thoughts and logic. Your feedback has been instrumental in helping us think hard to refine our work and present a more coherent and well-supported argument. We are grateful for the time and effort you have invested in reviewing our research and providing such constructive criticism.

- 4.1. The introduction reads more like a Scientific American article or an op ed than an academic paper. I would prefer it to talk about skill specialization and acquisition. There is a lot of good work speaking about task specialization, including even Adam Smith and Emile Durkheim, but also a lot of economic and sociology work.

The authors nail the complexity literature, which they know well. But unless the authors can also completely re-write the intro to really speak to the academic literature on task specialization and productivity, I don't think this paper achieves a relationship to the academic discussion on the topic. You have a lot of the right papers in citations 17 to 31; you may want to actually build around them more than simply citing them.

I believe this change is very important. If the authors cannot make a paper that can speak to the academic literature, I would strongly encourage rejection.

Thank you for bringing this issue to our attention. The main authors come from the fields of economic complexity, urban scaling, science of science, and network science, which was the primary motivation behind our project, and our manuscript was written as such. Our aim is to draw attention to our conditional probability and nested structure framework, encouraging its broad application and testing within these fields of more computational focus. However, upon your feedback, we realized that our previous introduction, while aligned with our core interests, read as tangential and lacked clarity, definition, and justification of our research questions, not to mention it was poorly written. Your feedback helped us

recognize the merit of initiating an engaging conversation within the traditional academic discourse. In response, we've rewritten the introduction to directly address the research question, emphasizing the importance of the skill dependencies structure (skill specialization and acquisition). We have removed unnecessary arguments and highlighted the key idea that occupational specialization is embedded within a complex network of interdependencies. By doing so, we aim to provide a clearer and more focused context for the research that follows. Furthermore, we have added more paragraphs to integrate complexity science, aligned with our work, with more conventional social science literature. We appreciate your valuable feedback, as it has helped us improve the clarity and impact of our manuscript.

Your feedback on the Introduction, along with other suggestions (4.4 and 4.6, therefore you will notice some of our response will naturally be reiterated), has prompted us to rewrite the entire manuscript, getting rid of tangential points and introducing more focused sections (Sec 3) featuring additional analyses and case studies (Fig 3, and Figs. S17, S18, S30, S49, and S50) to make a more cohesive point, underpinning our argument. We have also elaborated on our conceptualizations and their theoretical contributions to sociology and economics throughout the manuscript. This involves integrating our work with existing scholarship and challenging the conventional framework of studying human capital, which often relies on contextually informed linear conceptualizations of skills.

Through this process, we were able to flesh out the theoretical contribution of our finding in a productive way: economic factors can be understood as purely structural properties in how skills co-exist in occupations. We find this not only interesting/surprising but also potentially profound, a shift in our view of skills and knowledge. In the economic and sociological literature, skills and knowledge are often categorized using context-informed or ad hoc labels, often dichotomous categories or simplistic, one-dimensional arrangements, resulting in the presupposed conceptualization of skills (e.g., blue- vs. white-collar, low- vs. high-skilled, or physical vs. cognitive skills), even when recognizing the value of skills (Mincer, 1958; Card, 1994), addressing challenges like the skill hold-up problem (Klein et al., 1978; Rogerson, 1992; Balmaceda, 2005; Kessler & Lülfsmann, 2006), critiquing the underinvestment in general human capital (Bovenberg & Jacobs, 2005; Gathmann & Schönberg, 2010), and conceptually distinguishing between general and specific skills (Becker, 1962; Poletaev & Robinson, 2008; Gathmann & Schönberg, 2010). This approach remains surprisingly prevalent even in contemporary literature, including recent works by Rotundo and Sackett (2004), Acemoglu and Restrepo (2019), and Cappelli et al. (2024).

In addition, much work in economics, sociology, and management has examined changes in the payoffs to different types of skills in the labor market. Identifying relevant types, such as social and technical skills, and expected effects from our insights is crucial for these studies. To cite a few examples, Deming and Noray (2020) have suggested the importance of technical skills in facilitating the transition of college graduates into the labor market; Deming (2017) shows increasing returns to social skills by comparing workers who entered the labor market in the 1980s and early 1990s with those who did so in the early 2000s; and Murnane, Willett, and Levy (1995) find that the increase in returns to cognitive skills from 1978 to 1986 contributed significantly to the rising wage premium associated with higher education. While these studies provide incredible insights into relevant skills, they do not provide an overall framework to explain the value of different skills in today's labor market.

We hope our structural properties relax the economic and sociological literature's categories by addressing this discussion through the ecological lens of nestedness in skill dependency. Basically, we look at the systems where skills are embedded, which is an ecological perspective. As the final section of our result alludes to, increases in nestedness over time can describe payoffs to cognitive and technical skills that are primarily nested, as well as the growing importance of social skills (which belong to the general skills category). Although we acknowledge that we do not provide the ultimate causation for this trend in the current paper, we offer a possible reasoning for it (perhaps unconventional reasoning), which is at an inquisitorial stage, far from complete. Nevertheless, to our knowledge, our finding is the first to show that

the values of skills can be attributed to the way skills are structured and embedded in the larger structure. This knowledge further reshapes theoretical and empirical developments on skill holdup and policy on instilling general human capital (Klein et al., 1978; Rogerson, 1992; Balmaceda, 2005; Bovenberg & Jacobs, 2005; Kessler & Lülfsmann, 2006; Stantcheva, 2017). However, we try to be careful here: we do not think our framework is a substitute for those existing theories but more of a complementary framework to understand where we are all heading when the economy becomes more complex and increasingly more specialized, manifested as nested long paths.

We believe this nested structure not only presents a new perspective on the growing returns to certain skills but also challenges the traditional human capital theory that assumes a linear and singular investment approach. In this revision, we elaborate on the theoretical implications of the nested skill structure, demonstrating how it can account for the observed patterns of career progression and wage premiums. This topological explanation is significant because the structure of the occupation-skill network constrains individual career paths, can increase disparity, carry macroscopic consequences, and has been shown to influence the system's resilience and stability (Allesina & Tang, 2012; Rohr et al., 2014; Saavedra et al., 2016; Moro et al., 2021).

Finally, the merit of structural explanation goes beyond the conceptualization and measurement of workplace skills. First, our methodologies offer scalability in conceptualizing skills, capabilities, and tools, readily extending arguments to higher levels of granularity, building toward the future of work literature. Skills' nestedness contribution captures skill heterogeneities and structural changes in dependency and provides an in-depth description of polarization. Second, our framework accommodates the increasing availability of fine-grained data on skills applied in the workplace, meaning that we can readily apply this method to other contexts.

Our framework, in which value and properties strictly arise from the topology of the skill network (especially topological properties used in ecology), might initially seem unorthodox and unconventional in social science, given its detachment from traditional context-oriented analysis. This was part of the reason that we were a bit timid in writing the previous version and wanted to stay in our comfort zone. Nevertheless, this topological approach is not entirely new. On the contrary, this approach is well-aligned with the philosophy behind the trend in complexity economics, urban scaling, and science of science literature (Neffke, 2019; Frank et al., 2019; Saavedra et al., 2011; Uzzi, 2013; I. Hong et al., 2020; Anderson, 2017; Hausmann & Hidalgo, 2011; Alabdulkareem et al., 2018; and E. Moro et al., 2021). These structural approaches have successfully addressed the systematic understanding of socioeconomic factors, including economic complexity, urban premiums, the role of novelty in innovation, and so on. Rewriting our manuscript in that direction was a very useful exercise for us, as we were able to flesh out the theoretical contribution of our findings in a productive way. Therefore, we reiterate our gratitude to the reviewer for pushing us to go a little bit further. We do hope to contribute to developing a structural framework that reconceptualizes human capital through a quantitative lens and reveals a nested hierarchical structure that reframes the discourse on skill investment and economic returns.

Allesina, S., & Tang, S. (2012). Stability criteria for complex ecosystems. *Nature*, 483(7388), 205–208. <https://doi.org/10.1038/nature10832>.

Alabdulkareem, A., Frank, M. R., Sun, L., AlShebli, B., Hidalgo, C., & Rahwan, I. (2018). Unpacking the polarization of workplace skills. *Science Advances*, 4(7), 1–10. <https://doi.org/10.1126/sciadv.aao6030>.

Anderson, K. A. (2017). Skill networks and measures of complex human capital. *Proceedings of the National Academy of Sciences*, 114(48), 12720–12724.

Balmaceda, F. (2005). Firm-Sponsored General Training. *Journal of Labor Economics*, 23(1), 115–133. <https://doi.org/10.1086/425435>

Bovenberg, A. L., & Jacobs, B. (2005). Redistribution and education subsidies are Siamese twins. *Journal of Public Economics*, 89(11), 2005–2035. <https://doi.org/https://doi.org/10.1016/j.jpubeco.2004.12.004>

Frank, M. R., Autor, D., Bessen, J. E., Brynjolfsson, E., Cebrian, M., Deming, D. J., ... & Rahwan, I. (2019). Toward understanding the impact of artificial intelligence on labor. *Proceedings of the National Academy of Sciences*, 116(14), 6531-6539.

Gathmann, C. & Schönberg, U. How general is human capital? A task-based approach. *Journal of Labor Economics* 28, 1–49 (2010).

Hausmann, R., & Hidalgo, C. A. (2011). The network structure of economic output. *Journal of economic growth*, 16, 309-342.

Hong, I., Frank, M. R., Rahwan, I., Jung, W. S., & Youn, H. (2020). The universal pathway to innovative urban economies. *Science Advances*, 6(34), 1–7. <https://doi.org/10.1126/sciadv.aba4934>

Kessler, A. S., & Lülfesmann, C. (2006). The Theory of Human Capital Revisited: On the Interaction of General and Specific Investments. *The Economic Journal*, 116(514), 903–923. <https://doi.org/10.1111/j.1468-0297.2006.01116.x>

Klein, B., Crawford, R. G., & Alchian, A. A. (1978). Vertical Integration, Appropriable Rents, and the Competitive Contracting Process. *The Journal of Law & Economics*, 21(2), 297–326. <http://www.jstor.org/stable/725234>

Moro, E., Frank, M. R., Pentland, A., Rutherford, A., Cebrian, M., & Rahwan, I. (2021). Universal resilience patterns in labor markets. *Nature Communications*, 12(1), 1972. <https://doi.org/10.1038/s41467-021-22086-3>.

Neffke, F. M. H. (2019). The value of complementary co-workers. *Science Advances*, 5(12). <https://doi.org/10.1126/sciadv.aax3370>.

Poletaev, M., & Robinson, C. (2008). Human capital specificity: evidence from the Dictionary of Occupational Titles and Displaced Worker Surveys, 1984–2000. *Journal of Labor Economics*, 26(3), 387-420.

Rogerson, W. P. (1992). Contractual Solutions to the Hold-Up Problem. *The Review of Economic Studies*, 59(4), 777–793. <https://doi.org/10.2307/2297997>

Rohr, R. P., Saavedra, S., & Bascompte, J. (2014). On the structural stability of mutualistic systems. *Science*, 345(6195), 1253497. <https://doi.org/10.1126/science.1253497>.

Saavedra, S., Stouffer, D. B., Uzzi, B., & Bascompte, J. (2011). Strong contributors to network persistence are the most vulnerable to extinction. *Nature*, 478(7368), 233–235. <https://doi.org/10.1038/nature10433>

Stantcheva, S. (2017). Optimal Taxation and Human Capital Policies over the Life Cycle. *Journal of Political Economy*, 125(6), 1931–1990. <https://doi.org/10.1086/694291>

Uzzi, B., Mukherjee, S., Stringer, M., & Jones, B. (2013). Atypical combinations and scientific impact. *Science*, 342(6157), 468-472.

4.2. I am not sure that your assumption that general skills are pre-requisite to specialized is valid since more specialized skills are also more applied. Taking your negotiations example, I think it is more likely that developing the negotiations skill *in turn* develops critical thinking and perspective taking, than the other way around. Similarity, being a soldier *develops* leadership

skills.

Overall, I would agree that skills co-occur and develop together, but I do not think you can draw a casual arrow in any way. Can we do the analyses without implying such causality?

Excellent point! This is indeed the nuanced perspective that we intended to convey in our original text, but perhaps did so too subtly. You are absolutely right: developing negotiation skills (dependent) further strengthens one's critical thinking (general), a reverse direction in our method. This explains why general skills still have a powerful impact on wage premiums (Fig 5 in the revision). However, this doesn't necessarily mean negotiation skills are not dependent in terms of career sequences. On the contrary, our statistics show that critical thinking is a prerequisite for transitioning into roles requiring new skills like negotiation, manifested as an asymmetric conditional probability of negotiation skills being contingent upon critical thinking across jobs. In other words, new roles demanding negotiation tend to follow those with critical thinking in career progressions, suggesting people acquire dependent skills in that order. We operate under the assumption that individuals are most likely to be rational, choosing not to pursue skills that are unnecessary or unrewarding, particularly in today's complex economy with a wide range of specialized skills. As our conditional probabilities are derived from occupation-level data of national surveys, this assumption of bounded rationality is likely valid at the population level.

This introduces a nuanced picture. While the sequence of acquiring/applying these skills may appear directional, their development actually occurs hand-in-hand along rewarding career pathways, as illustrated in Fig 5 (previously Fig. 4). The asymmetric order reflects both the sequence of acquiring new skills and an individual's career advancement, while skills are developed through nested paths in the tree structure (Fig. 2). We recognize that our previous manuscript did not clearly articulate this significant insight. We had indirectly noted the importance of foundational skills for career progression and the potential for wage growth to stagnate without continuous general skill development.

This perspective is even more complicated when our thinking is so ingrained in linear progression. Indeed, human capital development is often portrayed as a preordained linear progression from general to specialized skills in mainstream economics, sociology, and psychology (Jovanovic 1997, Klein et al., 1978; Rogerson, 1992; Balmaceda, 2005; Kessler & Lülfesmann, 2006).

In response to your comments, we made our conceptualizations more upfront and articulated our findings to the extent that we do not overwhelm the reader, especially since many readers are likely to be unfamiliar with the conditional probability framework and the resulting branches in the tree structure in the context of human capital. We also explicitly acknowledge the limitations of our assumptions and their potential impact on our findings. The data is too crude to tease out all sorts of detailed processes, such as acquisition, development, and applications, in a clear way.

Indeed, we regard this as a promising direction for future work. For instance, machine learning algorithms may uncover latent structures from larger, more diverse datasets. Nevertheless, understanding these structures in a human-recognizable way is crucial as we move towards a machine learning, prediction-oriented approach, often developed in industries like LinkedIn, Revelios, and Burning Glass Technology, where machines predict individuals' job transitions. Our research provides an important foundation before industries dive into these predictive exploitations of data mining through recommendation systems, dominating our understanding.

To address your comments, we conducted additional follow-up analyses and included them in the revised manuscript. These support our original premise and help readers recognize the nuanced interplay between the sequence of acquiring new skills and their development along that sequence (one cannot neglect fundamental skill development even when progressing to specialized skills). We showcase how our

approach captures skill progression through: (i) a case study comparing registered nurses and nurse practitioners to examine how our skill hierarchies reflect the deepening of skill branches as individuals advance (SI Sec. 3.4), and (ii) an example illustrating how language deficiencies among Hispanic immigrants create barriers to dependent downstream niche skills (SI Sec. 3.5). We propose our framework as a theoretical springboard for further investigations into how skills are developed, applied, and valued, potentially enabling causal identification of the order in which general and specialized skills are acquired.

However, even then, fully understanding the directionality/bidirectionality of skill development/acquisition/application requires a more detailed examination of underlying skill changes, and our approach merely scratches the surface. One way to disentangle the order of development is through worker surveys, asking individuals about the sequence in which they acquire/apply skills from different skill groups. In response to this review, we conducted a limited survey of 1,000 individuals and found it aligned with our temporal career analysis (initial manuscript Fig. 3), nurse case studies (SI Sec. 3.4), and possibly Fig. 5, suggesting a simultaneous growth of general and specialized skills along nested specialization paths. While promising, incorporating a comprehensive survey into an academic paper introduces additional complexities associated with human subjects research (e.g., psychological methods including controlling priming, manipulation, and mediation) that are beyond our current scope. We plan to pursue a more thorough survey in future research to provide stronger empirical evidence. However, we also recognize that identifying the causal arrow remains an open question that may require more qualitative methods, such as interviews or experiments. This research area opens up new perspectives on the system, and we are excited about its potential. We have discussed the need for future research using more suitable data and methods.

We appreciate your feedback, as it has helped us refine our assumptions and present a more nuanced perspective on skill development. We hope our revisions address your concerns and provide a clearer understanding of the limitations and potential of our approach. This research area is opening up exciting new ways of looking at the system, and we look forward to further advancing this line of inquiry.

Balmaceda, F. (2005). Firm-Sponsored General Training. *Journal of Labor Economics*, 23(1), 115–133. <https://doi.org/10.1086/425435>

Jovanovic, B., & Nyarko, Y. (1997). Stepping-stone mobility. *Carnegie-Rochester Conference Series on Public Policy*, 46, 289–325. [https://doi.org/https://doi.org/10.1016/S0167-2231\(97\)00012-2](https://doi.org/https://doi.org/10.1016/S0167-2231(97)00012-2)

Kessler, A. S., & Lulfesmann, C. (2006). The Theory of Human Capital Revisited: On the Interaction of General and Specific Investments. *The Economic Journal*, 116(514), 903–923. <https://doi.org/10.1111/j.1468-0297.2006.01116.x>

Klein, B., Crawford, R. G., & Alchian, A. A. (1978). Vertical Integration, Appropriable Rents, and the Competitive Contracting Process. *The Journal of Law & Economics*, 21(2), 297–326. <http://www.jstor.org/stable/725234>

Rogerson, W. P. (1992). Contractual Solutions to the Hold-Up Problem. *The Review of Economic Studies*, 59(4), 777–793. <https://doi.org/10.2307/2297997>

- 4.3. I disagree conceptually that specialization would correlate closely with age beyond education. Can you be clearer on this?

I think this section can also use more referencing to social science:

- a. I am surprised specialization plateaus after age 30, because typically we see wages go up until age 40. Why are these two different and what can we learn from your measures?
- b. The main underlying unobserved variable, of course, is education. How do your measures vary across education? Can you show, for example, that if you take people who didn't study college, your patterns also track?

My worry is that what you have is just a poorer measure of education and that I would learn a lot more by simply looking at the degrees that people have, years studied, and the time they study them as a measure of skill development. Such would also explain why men and women follow different paths in skill acquisition, for example.

I think doing this well is critical for this paper to be a contribution..

Thank you for sharing your insights. Before we proceed to your comment 4.3, we'd like to address a potential discrepancy between two of your comments. Specifically, comment 4.3 expressed surprise that "specialization plateaus after age 30, given that wages usually increase until age 40," while comment 4.7 noted that "...Men continue to grow their general and nested skills until well 238 into their 40s" but it seems to me your measure plateaus for men much beyond their 40s in fig 3e?..." We want to clarify that the observation in comment 4.7 aligns with our findings—Figure 3e indeed shows that skill growth for men extends into their 40s, albeit at a diminishing rate. The reason for this marginalized increase may stem from various factors. When skills and knowledge reach the end of the category spectrum, they suffer from a lack of resolution to capture the frontiers due to two main reasons: (1) the limited number of individuals at the frontier does not incentivize government agencies to create and measure new categories because of the associated costs; and (2) the evolving nature of frontier skills and knowledge makes it challenging to capture their appropriate vocabulary, as it takes time for us to recognize their use and create new terms in our dictionary (a phenomenon studied in urban settings by H. Youn et al., 2016). Given these complexities, we wouldn't focus too heavily on underpinning the exact age at which skill increases plateau, as various factors influence our measurements at the tails of the distribution.

What we can argue here is that our empirical analysis shows that skill development at least continues past the age of 30 and into the 40s, suggesting there is something more than formal education offers in skill acquisitions as statistically few individuals older than 30 still attend school, which we discuss and offer evidence of in Fig. S49, below. As you suggest, "typically we see wages go up until age 40," possibly indicating post-school learning, which may occur through experience, training, learning-by-doing, and spillovers (Becker, 1962; Irwin & Klenow, 1994; Jovanovic & Nyarko, 1997; Levitt et al., 2013; Bessen, 2015; Nedelkoska et al., 2015).

To address your concern, beyond including necessary references to the relevant work in economics, sociology, psychology, and education (Mincer, 1958; Murnane et al., 1995; Autor, 2013, 2014; Goldin & Katz, 2008; Pritchett, 2001; Kerckhoff, 2001), we conducted two additional analyses that highlight the role of education. First, in Fig. S49, we replicate the evolution of skill and age as depicted in the initial manuscript Fig. 3d-f, alongside data on educational attainment and the proportion of individuals attending school as a function of age sourced directly from the Current Population Survey (CPS). Notably, the educational attainment variable in CPS ranges from 02 (indicating no schooling) to 125 (indicating a doctorate degree). To gauge the fraction of the sample attending school, we utilized the CPS variable SCHOOLCOL, which distinguishes between individuals attending high school (coded as 1 or 2) or college/university (coded as 3 or 4), or those not attending school (coded as 5). We transformed this information such that individuals attending school receive a value of 1, while non-attendees are assigned a value of 0. Despite educational attainment plateauing by the age of 30 and a significant drop in school attendance, our analyses reveal that skill growth persists (voluntarily or involuntarily). This observation suggests the existence of alternative mechanisms for skill accumulation beyond formal education. Second, in response to your comment, we replicated the skill-age analysis in Fig. S50 for individuals with no more than a high school diploma (values of less than or equal to 073 on the CPS education attainment variable), revealing that similar skill accumulation patterns, albeit in smaller magnitudes, emerge even among individuals without college degrees. The patterns resemble the skill accumulation across the population, even though the levels of general and nested skills are lower compared to the population-level estimates.

In fact, our approach aims to complement, rather than replace, traditional views on education within human capital contexts. For example, in Fig 2, we embedded skills not only in their reachable centrality (y-axis) but also associated education (x-axis) to show that the most specific skills (on the leaves of the tree) are spreading across various levels of education, suggesting that every specialization (from top to bottom of the tree) is associated with education in their direction, and can have lasting impacts on career trajectories, influenced by initial educational choices. This can lead to skill entrapment across intra-generations. In Fig. S30, we show the wage-age curves for four distinct periods of 5 years to avoid conflating long-run economic factors (Argote & Epple, 1990, Nedelkoska et al., 2015). In three of the four periods, un-nested jobs have an early wage lead, which quickly evaporates with age. It becomes harder to develop nested specialized skills, as contingencies become stricter, thereby increasing barriers for someone without the contingency skills, emphasizing the potential opportunity costs tied to early educational decisions. Together with Fig 7, where the skill structure becomes more nested over the decade, the result suggests that the more nested the human capital structure becomes, the more skill entrapment takes place for intra-generational (within one's life).

This also has implications for labor disparity not only intra-generationally but also for other demographic factors. For example, Education does not always track wages; our data shows education and wage, for instance, do not correlate across gender lines. Women are on average more or equally educated; yet, wages do not follow. Without direct skill data, one could argue that this is due to extremes of the wage distributions; but we do see skill differentials more closely track wages than education does for gender. This skill-based framework allows more nuanced analysis, such as what we show in SI Sec. 7.1 and 7.2 that further investigate the influence of parenthood on male and female workers' skills by slicing data by those with and without children as well as the influence of sorting into jobs based on schedule and working hours, respectively.

Together, empirical evidence based on gender and the temporality of skill acquisition corroborate the conceptualization of education as one possible input to skill acquisition, without too much undermining its vital role. While education is an input to skills, skill acquisition continues on the job with experience, training, and spillover. Heckman et al. (1998) estimate that half of lifetime learning is obtained post-school, and skills are also the driving force (Jovanovic & Nyarko, 1997). For instance, Murnane, Willett, and Levy (1995) find that the increase in returns to cognitive skills from 1978 to 1986 contributed significantly to the rising wage premium associated with higher education, but educational components (supply side) may also deviate from the skills needed in the workplace (demand sides) (Borner et al., 2018). Indeed, given the increasing availability of finer-grained data, using education to approximate skills has come under scrutiny (Pritchett, 2001), and we expand education's capacity to capture the multidimensionality, heterogeneity, and variation in the quality of the instilled skills (Pritchett, 2001). In fact, our SI Table S5, below, shows that each subset of our skill categories explains occupational wages equally as well as education and withstands other controls, such as experience and training. In that sense, we are of the view that our framework enables more in-depth investigation of education as a topic in its own right. The emphasis on education as a sole source for skill has perhaps hampered our inquiry into when and where it is an appropriate investment for skill development.

As such, we reiterate our view is not substitutional but complementary to the existing view. While there is no doubt about the importance of education as a source of skill acquisition and as a powerful proxy for measuring skills, there is much to examine about the dynamics of skills and wages beyond education. This suggests that there is more than formal education that plays out in skill acquisition in occupations. Thus, we thank the reviewer for pushing us to go one step further to flesh out our standpoint.

Argote, L., & Epple, D. (1990). Learning Curves in Manufacturing. *Science*, 247(4945), 920–924.
<http://www.jstor.org/stable/2873885>.

Autor, D. H., & Handel, M. J. (2013). Putting tasks to the test: Human capital, job tasks, and wages. *Journal of Labor Economics*, 31(S1), S59-S96.

Autor, D. H. (2014). Skills, education, and the rise of earnings inequality among the "other 99 percent". *Science*, 344(6186), 843-851.

Becker, G. S. (1962). Investment in Human Capital: A Theoretical Analysis. *Journal of Political Economy*, 70(5), 9-49.

Bessen, J. (2015). Learning by doing: The real connection between innovation, wages, and wealth. In *Learning by Doing: The Real Connection between Innovation, Wages, and Wealth* (Vol. 52, Issue 12). Yale University Press. <https://doi.org/10.5860/choice.191886>.

Börner, K., Scrivner, O., Gallant, M., Ma, S., Liu, X., Chewning, K., Wu, L., & Evans, J. A. (2018). Skill discrepancies between research, education, and jobs reveal the critical need to supply soft skills for the data economy. *Proceedings of the National Academy of Sciences*, 115(50), 12630-12637. <https://doi.org/10.1073/pnas.1804247115>.

Goldin, C., & Katz, L. F. (2008). The Race Between Technology & Education. In *Harvard University Press*.

Heckman, J. J., Lochner, L., & Taber, C. (1998). Tax Policy and Human-Capital Formation. *The American Economic Review*, 88(2), 293-297. <http://www.jstor.org/stable/116936>

Irwin, D. A., & Klenow, P. J. (1994). Learning-by-Doing Spillovers in the Semiconductor Industry. *Journal of Political Economy*, 102(6), 1200-1227. <http://www.jstor.org/stable/2138784>

Jovanovic, B., & Nyarko, Y. (1997). Stepping-stone mobility. *Carnegie-Rochester Conference Series on Public Policy*, 46, 289-325. [https://doi.org/https://doi.org/10.1016/S0167-2231\(97\)00012-2](https://doi.org/https://doi.org/10.1016/S0167-2231(97)00012-2)

Kerckhoff, A. C. (2001). Education and Social Stratification Processes in Comparative Perspective. *Sociology of Education*, 74, 3-18. <https://doi.org/10.2307/2673250>

Levitt, S. D., List, J. A., & Syverson, C. (2013). Toward an Understanding of Learning by Doing: Evidence from an Automobile Assembly Plant. *Journal of Political Economy*, 121(4), 643-681.

Mincer, J. (1958). Investment in Human Capital and Personal Income Distribution. *Journal of Political Economy*, 66(4), 281-302. <http://www.jstor.org/stable/1827422>.

Murnane, R., Willett, J. B., & Levy, F. (1995). The growing importance of cognitive skills in wage determination.

Nedelkoska, L., Patt, A., & Ederer, P. (2015). Learning by problem solving. *Available at SSRN*.

Pritchett, L. (2001). Where Has All the Education Gone? *World Bank Economic Review*, 15(3), 367-391.

Youn, H., Bettencourt, L. M., Lobo, J., Strumsky, D., Samaniego, H., & West, G. B. (2016). Scaling and universality in urban economic diversification. *Journal of The Royal Society Interface*, 13(114), 20150937.

Figure 6: Skill Disparity in Demographic Distribution of race/ethnicity and gender (a) The relative average skill level, education level, and weekly wages for Asian, Black, and Hispanic/Latinx workers compared to White workers (expressed as a ratio). **(b)** The relative average skill level, education level, and weekly wages for female workers compared to male workers. 95% confidence intervals for each estimated ratio are calculated by bootstrapping subsamples (see Methods). These differentials are robust to measurement (SI Fig. S48), follow similar age trends seen in Fig. 4, and are robust to time-variant economic factors (Fig. S53.) SI Figs. S50 and S51, further show the gaps have narrowed over time.

	log(Wage₂₀₁₉)					
	(1)	(2)	(3)	(4)	(5)	(6)
General	0.251*** (0.237,0.264)			0.135*** (0.109,0.161)		
Intermediate^{Nested}		0.260*** (0.245,0.274)			0.120*** (0.098,0.141)	
Intermediate^{Unnested}		0.015** (0.002,0.028)			0.034*** (0.020,0.047)	
Specific^{Nested}			0.199*** (0.182,0.215)			0.042*** (0.020,0.064)
Specific^{Unnested}			-0.051*** (-0.063,-0.039)			0.003 (-0.011,0.017)
Education				0.024*** (0.018,0.030)	0.034*** (0.029,0.039)	0.042*** (0.036,0.048)
Experience				0.014*** (0.008,0.021)	0.014*** (0.008,0.021)	0.023*** (0.016,0.030)
Training				0.038*** (0.029,0.048)	0.027*** (0.017,0.037)	0.042*** (0.031,0.052)
Constant	3.909*** (3.861,3.956)	4.069*** (4.017,4.121)	4.550*** (4.520,4.581)	3.969*** (3.908,4.031)	4.043*** (3.997,4.090)	4.223*** (4.182,4.264)
Observations	789	789	789	789	789	789
R ²	0.622	0.607	0.470	0.686	0.703	0.653
Adjusted R ²	0.622	0.606	0.469	0.684	0.701	0.651
Residual Std. Error	0.119	0.122	0.142	0.109	0.106	0.115
F Statistic	1,297.040***	607.527***	348.362***	428.084***	370.741***	294.761***

*Note: OLS regressions are shown, with 95-percentile confidence intervals in parentheses (*p<0.1; **p<0.05; ***p<0.01). R², coefficient of determination, and adjusted R² is normalized for the models' number of variables.*

SI Table S5: Wage Regression on Skill Endowment. Importantly, the correlation and predictive power of skills withstand controls such as Education, Training, and Experience.

Fig. S49. Evolution of skill, age and education. To measure education, we have used educational attainment and the fraction of individuals who attend school as functions of age, both taken from the Current Population Survey (CPS). The education attainment variable ranges from 2 (i.e., no schooling) to 125 (i.e., doctorate degree). To obtain the fraction of the sample attending school, we utilized the information in CPS variable SCHOOLCOL that documents attending high school (1 or 2) or college/university (3 or 4) or not attending school (5). We transformed the information so that if an individual attends school (1,2,3, or 4), it receives a value of 1, and if not attending, it has a value of 0. Even though by the age of 30 education plateaus and school attendance drops significantly, skill growth continues, manifesting the presence of other mechanisms for skill accumulation apart from education.

Fig. S50. replicated the skill-age analysis for the subset of individuals who have obtained no more than a high school diploma (values of less than or equal to 073 on the CPS education attainment variable.) The patterns resemble the skill accumulation across the population, even though the levels of general and nested skills are lower compared to the population-level estimates.

Figure S30: Wage Curves for Occupations with Primarily Nested vs. Primarily Un-nested Skills. We average over the levels of nested and un-nested skills of each occupation in our sample, and pick occupations at the top 20% of the nested skills as the most nested, and occupations at the top 20% of the un-nested skills as the most un-nested. Matching these occupations to the individuals in the CPS, we can obtain estimates of wages for individuals in these occupations at different ages. To avoid conflating long-run economic factors, we show the wage-age curves for four 5-year periods: 1983-1987, 1993-1997, 2003-2007, 2013-2017. Un-nested jobs have an early wage lead which quickly evaporates with age.

4.4. A central assumption of this paper that “nestedness is good”, but I needed a lot more clarity on why this was the case from a theoretical perspective. In a sense, I could find many things that also predict income: height, home ZIP Code, physical image, number of children, job role, country of birth and culture, etc. I would like to understand why nestedness is important on its own, rather than simply some measure with predictive power. In particular because, in a way, nested skills are also somewhat less specialized?

We appreciate this question. Together with comments 4.1 and 4.6, you have highlighted our failure to justify and position the lens of nestedness in the context of our investigation of human capital. At the risk of some repetition, we must clarify that our work develops a structural view of the interdependencies between skills. The main premise is that the skill interdependencies can explain skills’ socio-economic implications, including necessary investment to develop and apply skills, value, and resilience of skills (see above in 4.1 and also 4.6).

Therefore, we wouldn’t say ‘nestedness is good’. Our goal is not to replace the context-informed (such as cognitive, physical, technical, social, Zip code, height) and semantic categories of skills, but to provide a complementary way to explain socio-economic properties of interest from interactions between skills. The core contribution of our work is a conceptual shift toward conceptualizing the conditional and directional skill interdependencies by distinguishing general and specialized skills, and elucidating the resulting hierarchy of skills. In that context, we found nestedness as a quantity that suits the purpose (but we are receptive to alternatives.)

Our rationale for using nestedness arose from the inherent similarity of our structural lens to the frameworks prevalent in structural ecology, where nestedness was first conceived to investigate a preferential interaction propensity between generalist and specialist species (Wright & Reeves, 1992,

Saavedra, 2011). With the observation of similar interdependence patterns between general and specialized skills in our data, nestedness emerged as a natural conceptual candidate.

We found nestedness offers empirical utility in describing skill value and automation risk, captured in the new result section and the new main Fig. 3. Critical to our work, we show that nestedness scores are capable of discerning skills that sit on nested and unnested specialization paths. We have further illustrated skill progression in such pathways by a case study of Registered Nurse and Nurse Practitioner (Fig. S17.) Both figures are included below for your convenience.

But beyond such empirical utilities, nestedness offers conceptual advantages. First, with the increasing availability of fine-grained data on workplace skills, nestedness contribution readily extends our arguments to skills at higher levels of granularity by capturing similar topological patterns. It is also flexible in conceptualizing heterogeneous entities, such as skills and technologies, within a unified framework. In the manuscript, we have further expanded on the implications of using the nestedness approach for social sciences, including a topological explanation of US job polarization (Fig. 7), providing an intuitive explanation of skill value based on Bargaining theory (Davidson, 1898), and unpacking and expanding the growing value of social skills (Deming, 2017). Finally, we outline that it provides a vast repertoire of ecological tools to study vital avenues in human capital theory, including resilience, stability, growth, and diversity (Allesina & Tang, 2012; Rohr et al., 2014; Saavedra et al., 2016; Moro et al., 2021).

Figure 3: Skill Nestedness Contributions Score. skills' nestedness score is highly indicative of their generality (a), risk of automation (b), and their value (c-d). Skill Nestedness Contributions is measured following [15]. Generality is measured by Local Reaching Centrality, as in Fig. 2, Automation risk Index and Value for each skill is calculated, following [65–67]. We divide skills into *nested*, positive contributions, and *un-nested*, negative contributions toward the nested skill structure.

Figure S17: Transition between Registered Nurses (RNs) and Nurse Practitioners (NPs). (a) uses resume data from Burning Glass Technology to capture the transition statistics between RNs and NPs. We restrict the analysis to individuals with at least five listed occupations in their resume and define their early career occupations as the most appeared occupation in the first three jobs, similarly late career occupations as the most appeared in the fourth jobs and onward. We disregard individuals whose early and late careers are neither RN nor NP. Including these individuals would not change the result but significantly complicate the exposition. One expects that higher wages for NPs would attract RNs (e). Indeed, most NPs were RNs early on. However, only a subset of RNs progresses to NP jobs, suggesting barriers to entry, summarized in higher experience and educational requirements (c-d). (b) captures the skill requirements of RNs and NPs, highlighting the advantage of integrating the conceptual distinction between general and niche skills with a structural network approach to studying skills in revealing pathways of progress (also known as “specialization”). The structure of our skill hierarchy also implies that progress entails co-development in certain niche skills and the prerequisite, often more general skills.

Allesina, S., & Tang, S. (2012). Stability criteria for complex ecosystems. *Nature*, 483(7388), 205–208. <https://doi.org/10.1038/nature10832>.

Davidson, J. (1898). *The bargain theory of wages ...* G. P. Putnam’s Sons.

Deming, D. J. (2017). The Value of Soft Skills in the Labor Market. *NBER*. <https://www.nber.org/reporter/2017number4/value-soft-skills-labor-market#9>.

Moro, E., Frank, M. R., Pentland, A., Rutherford, A., Cebrian, M., & Rahwan, I. (2021). Universal resilience patterns in labor markets. *Nature Communications*, 12(1), 1972. <https://doi.org/10.1038/s41467-021-22086-3>.

Rohr, R. P., Saavedra, S., & Bascompte, J. (2014). On the structural stability of mutualistic systems. *Science*, 345(6195), 1253497. <https://doi.org/10.1126/science.1253497>.

Saavedra, S., Stouffer, D. B., Uzzi, B., & Bascompte, J. (2011). Strong contributors to network persistence are the most vulnerable to extinction. *Nature*, 478(7368), 233–235. <https://doi.org/10.1038/nature10433>

Wright, D. H., & Reeves, J. H. (1992). On the meaning and measurement of nestedness of species assemblages. *Oecologia*, 92(3), 416–428. <https://doi.org/10.1007/BF00317469>.

4.5. I would recommend completely removing the geographical analyses. Overall, your results suggest that there is a gap in skill specialization across regions, but there are fundamentally different economic clusters across regions, and these depend on regional comparative advantage. In other words, the reason Laredo, Texas is not specialized is because its main economic function is to be the entry point of land trade into the U.S.

In general, I do not think differences in specialization are nothing more than a consequence of other local economic forces, and certainly not a new measure of regional inequality..

We are thankful to reviewer 4 for insightful comments on the analysis of geographic distribution of skills. We side with suggestions of reviewers 4 and 1 that heterogeneous and complex determinants of geographic concentration of skills require a more careful assessment, imposing significant changes to the current set of analyses to produce robust and insightful results. We agree with both reviewers that the task is likely beyond the core scope of the current paper. Therefore, we have moved the current analysis to the supplementary document, addressing the alignments of our skill structure with known facts in the literature, and we will take on the suggested directions by reviewers in future work.

However, we felt compelled to explore the curiosity behind several suggestions from reviewers to the extent that our data and framework allow. The economic geography, economic complexity, and urban scaling literature has found evidence of urban wage premiums, which it has associated with the complexity of economic activities in larger and more populous areas (Glaeser, 1999; Wheeler, 2001; Rosenthal & Strange, 2004; Combes et al, 2010; L. Bettencourt et al., 2007; L. Bettencourt et. al., 2014, H. Youn et al., 2016; I Hong et al., 2020; Gomez-Lievano & Patterson-Lomba, 2021). These evidences have been amounting to support our skill hierarchy—more complex economic activities require the accumulation of nested and, most importantly, general skills, which are concentrated in larger cities (Lucas, 1988; Bacolod et al., 2009; Gomez-Lievano et al., 2016), and thus individual dynamics are recapitulating (I. Hong et al., 2020), implying the nested nature of the capability structure. While most workers with nested skills need general skills, the concentration of managerial and other supporting roles also requires high levels of general skills. Such prediction is supported by our descriptive analysis (Fig. 6 a,b, and d in the initial manuscript). However, as Reviewer 1 pointed out, we did not offer any analysis that examines whether the accumulation of general skills indeed explains part of the value generated in cities. Here, we test that hypothesis directly by utilizing the CBSA size (CBSASZ) variable from CPS microdata, which carries information about the size of the metropolitan area in which the surveyed individual resides (since 2004). The values range from 0 (areas of <100,000 inhabitants that do not meet the threshold of a metropolitan area) to 6 (over 5 million inhabitants). We transform these brackets to cities below and above 1M population, following Hong et al. (2020).

In model (1) of Table S6 below, we first regressed the log-wage reported by individuals to CPS on the size of the metropolitan area in which they reside, obtaining partial correlations that signify the urban wage premiums (the baseline is areas of <1M inhabitants). In the second model of the table, we add general skills of individuals (which we obtain from matching O*NET to the occupation associated with each individual in the CPS microdata). This indicates that large cities tend to have more people in occupations with general skills. This bias toward more general-skill-intensive activities explains over one-third of the urban wage premiums (Lucas, 1988; Glaeser & Maré, 2001; Bacolod et al., 2009; Gomez-Lievano & Patterson-Lomba, 2021). Adding nested and un-nested specific skills first without and then with general skills, in models 3 and 4, respectively, have similar effects. This analysis is made available as part of the SI Sec. 6.

The results are inspiring, and we are excited about them, but we find that this requires more tests and understanding to be suitable for academic publications and thus beyond our current scope. Nevertheless, these results provide us with confidence that our metric and conceptualization are useful for

understanding not only the socioeconomic structure of individual career trajectories but also individual regional economies, which we would like to explore further in the future.

Urban Wage Premiums and Skills				
Dependent variable:				
Log(Wage)				
OLS				
	(1)	(2)	(3)	(4)
Population > 1M	0.082*** (0.080, 0.084)	0.054*** (0.053, 0.056)	0.056*** (0.054, 0.058)	0.059*** (0.057, 0.060)
General Skills		0.269*** (0.268, 0.270)		0.281*** (0.278, 0.283)
Nested Specific Skills			0.248*** (0.246, 0.250)	0.007*** (0.005, 0.010)
Un-nested Specific Skills			-0.073*** (-0.074, -0.072)	0.026*** (0.025, 0.027)
Constant	4.671*** (4.670, 4.673)	3.787*** (3.783, 3.792)	4.471*** (4.469, 4.474)	3.709*** (3.702, 3.717)
Observations	635,554	635,554	635,554	635,554
R ²	0.012	0.200	0.141	0.203
Adjusted R ²	0.012	0.200	0.141	0.203
Residual Std. Error	0.368 (df = 635552)	0.331 (df = 635551)	0.343 (df = 635550)	0.331 (df = 635549)
F Statistic	7,845.032*** (df = 1; 635552)	79,439.180*** (df = 2; 635551)	34,872.520*** (df = 3; 635550)	40,451.410*** (df = 4; 635549)
Note:				*p<0.1; **p<0.05; ***p<0.01

Bacolod, M., Blum, B. S., & Strange, W. C. (2009). Skills in the city. *Journal of Urban Economics*, 65(2), 136–153. <https://doi.org/https://doi.org/10.1016/j.jue.2008.09.003>.

Bettencourt, L. M., Lobo, J., Helbing, D., Kühnert, C., & West, G. B. (2007). Growth, innovation, scaling, and the pace of life in cities. *Proceedings of the national academy of sciences*, 104(17), 7301-7306.

Bettencourt, L., Samaniego, H. & Youn, H. Professional diversity and the productivity of cities. *Sci Rep* 4, 5393 (2014). <https://doi.org/10.1038/srep05393>

Combes, P.-P., Duranton, G., Gobillon, L., & Roux, S. (2010). *Estimating Agglomeration Economies with History, Geology, and Worker Effects* (pp. 15–66). National Bureau of Economic Research, Inc.

Glaeser, E. L. (1999). Learning in Cities. *Journal of Urban Economics*, 46(2), 254–277. <https://doi.org/https://doi.org/10.1006/juec.1998.2121>.

Glaeser, E. L., & Maré, D. C. (2001). Cities and Skills. *Journal of Labor Economics*, 19(2), 316–342. <https://doi.org/10.1086/319563>.

Gomez-Lievano, A., & Patterson-Lomba, O. (2021). Estimating the drivers of urban economic complexity and their connection to economic performance. *Royal Society Open Science*, 8(9), 210670. <https://doi.org/10.1098/rsos.210670>.

Gomez-Lievano, A., Patterson-Lomba, O. and Hausmann, R., 2016. Explaining the prevalence, scaling and variance of urban phenomena. *Nature Human Behaviour*, 1(1), p.0012.

Hong, I., Frank, M. R., Rahwan, I., Jung, W. S., & Youn, H. (2020). The universal pathway to innovative urban economies. *Science Advances*, 6(34), 1–7. <https://doi.org/10.1126/sciadv.aba4934>

Lucas, R. E. (1988). On the mechanics of economic development. *Journal of Monetary Economics*, 22(1), 3–42. [https://doi.org/https://doi.org/10.1016/0304-3932\(88\)90168-7](https://doi.org/https://doi.org/10.1016/0304-3932(88)90168-7).

Rosenthal, S. S., & Strange, W. C. (2004). Chapter 49 - Evidence on the Nature and Sources of Agglomeration Economies. In J. V. Henderson & J.-F. B. T.-H. of R. and U. E. Thisse (Eds.), *Cities and Geography* (Vol. 4, pp. 2119–2171). Elsevier. [https://doi.org/https://doi.org/10.1016/S1574-0080\(04\)80006-3](https://doi.org/https://doi.org/10.1016/S1574-0080(04)80006-3).

Wheeler, C. H. (2001). Search, Sorting, and Urban Agglomeration. *Journal of Labor Economics*, 19(4), 879–899. <https://doi.org/10.1086/322823>.

Youn, H., Bettencourt, L. M., Lobo, J., Strumsky, D., Samaniego, H., & West, G. B. (2016). Scaling and universality in urban economic diversification. *Journal of The Royal Society Interface*, 13(114), 20150937.

SMALLER COMMENTS

4.6. I don't understand all the references to ecology and what value do they bring. I would remove them, personally.

Thank you for bringing this to our attention. This comment connects with earlier comments in 4.1 and 4.4, clearly indicating that our initial explanation and justification of the ecological concept of nestedness—a key ecological framework—was unclear and poorly written. This realization led us to substantially revise the entire manuscript to better articulate why and how nestedness serves as an effective framework for quantifying the complex dependency patterns observed between specialized and general skills in the context of our research.

The most significant changes in response to this feedback are: (1) rewriting the manuscript to drive home, by explaining conceptually and empirically, why we chose the nestedness framework and applied metrics based on it; (2) introducing a new section (Sec 3), accompanied by further analyses (Fig 3), to substantiate our methodological choices; (3) adding case studies to showcase skill hierarchies in career trajectories; and (4) detailing the broader implications of our findings throughout the manuscript.

A brief summary for your convenience follows:

The nestedness toolkit emerged as a natural candidate for our analysis, and main motivations of the research, given the presence of preferential interaction between general and specialized skills in our skill network. This choice is primarily motivated by the structural ecology literature that has observed and investigated the preferential interaction of generalist and specialist species under the nestedness framework. In addition, the metric has been tested and established in this literature with well-understood properties, and has also recently found utility in the social sciences and economic complexity (Saavedra et al., 2011; König et al., 2014; Borge-Holthoefer et al., 2017).

By adopting the nestedness lens, we aim to operationalize measures and unveil economic systems, progression pathways, and reward mechanisms, detaching the context-informed categories. The new result section (Sec 3: Nestedness Contribution) elaborates how these nested interdependence patterns reveal skills' value and resilience to automation. In ecology, the interdependencies of specialists on generalists imply a progression in dynamics, called primary succession, unfolding individual progressions as well as describing organizations (areas, companies, and countries) (whose urban settings are shown in Hong et al., 2020). For example, in primary succession, generalists often arrive in an area first to make the system function, and specialists cannot survive in the absence of generalists, showing the directionality of their dependencies.

To showcase how our approach captures specialization paths, we conducted additional case studies, (i) comparing registered nurses with nurse practitioners, which highlight how our skill hierarchies capture deepening branches of skills as individuals progress (SI Sec. 3.4), and (ii) revealing how language deficiencies across Hispanic immigrants mount barriers to dependent downstream specialized skills, causing skill entrapment in unnested human capital (SI Sec. 3.5). By doing so, we propose our framework

as a theoretical springboard for further investigations into how skills are developed, applied, and valued within the complex fabric of modern society.

In addition, this approach allows us to analyze the interactions between specialists and generalists based on structural information, rather than semantic classifications such as physical, cognitive, social, and managerial skills, or white vs. blue collar jobs. We simply group skills by their demand distributions (broad and narrow) and then determine which skills are more conditionally dependent on others. Then, we classify skills by their conformity to the overarching nestedness (c_s , each skill's nested contribution score). These are purely structural information with no introspective insights involved, making it an interdisciplinary approach appropriate for a general-science journal. The nestedness measure is useful for operationalizing these results in a systemic way and uncovering the importance of structural properties in the economy's rewarding system. It also enables a systematic quantification of the dependencies as well as how skills branch off from dependencies by measuring individual skills' contributions to the overarching nested structure (a way of measuring the dependency structure).

Moreover, nestedness offers a scalable metric for skill descriptions, allowing the extension of our arguments to finer levels of granularity or newly created datasets, replication studies using other skill topologies, and incorporation of new entities like technologies. By capturing topological interactions, the nestedness framework seamlessly integrates skills and technologies required by occupations, enabling the quantification of their interactions and development of testable implications, including value and economic resilience (Allesina & Tang, 2012; Rohr et al., 2014; Saavedra et al., 2016; Moro et al., 2021). This feature will be extremely useful to understand the future of work. In contrast, the semantic labels and even the analytical models developed based on skills may require foundational rework to accommodate such changes.

Our revisions aim to clarify the application and significance of nestedness within the occupation-skill network, emphasizing its value in dissecting the complex dynamics of skill interdependencies. While open to alternative quantification methods, we believe that presenting our findings in a broad-audience journal catalyzes further investigation into the methodological and theoretical facets of human capital research, underscoring our work's foundational shift towards recognizing asymmetric information in skill dependencies.

Allesina, S., & Tang, S. (2012). Stability criteria for complex ecosystems. *Nature*, 483(7388), 205–208. <https://doi.org/10.1038/nature10832>.

Borge-Holthoefer, J., Baños, R. A., Gracia-Lázaro, C., & Moreno, Y. (2017). Emergence of consensus as a modular-to-nested transition in communication dynamics. *Scientific reports*, 7(1), 41673.

König, M. D., Tessone, C. J., & Zenou, Y. (2014). Nestedness in networks: A theoretical model and some applications. *Theoretical Economics*, 9(3), 695–752.

Moro, E., Frank, M. R., Pentland, A., Rutherford, A., Cebrian, M., & Rahwan, I. (2021). Universal resilience patterns in labor markets. *Nature Communications*, 12(1), 1972. <https://doi.org/10.1038/s41467-021-22086-3>.

Rohr, R. P., Saavedra, S., & Bascompte, J. (2014). On the structural stability of mutualistic systems. *Science*, 345(6195), 1253497. <https://doi.org/10.1126/science.1253497>.

Saavedra, S., Rohr, R. P., Olesen, J. M., & Bascompte, J. (2016). Nested species interactions promote feasibility over stability during the assembly of a pollinator community. *Ecology and Evolution*, 6(4), 997–1007. <https://doi.org/10.1002/ece3.1930>.

Saavedra, S., Stouffer, D. B., Uzzi, B., & Bascompte, J. (2011). Strong contributors to network persistence are the most vulnerable to extinction. *Nature*, 478(7368), 233–235. <https://doi.org/10.1038/nature10433>

4.7. You have the following sentence “Men continue to grow their general and nested skills until well 238 into their 40s” but it seems to me your measure plateaus for men much beyond their 40s in fig 3e?

We thank the reviewer for brining this to our attention. We agree with the review that both general and nested skills for men plateau around the ages of 45 and 50. Consequently, we have revised the sentence to accurately reflect this finding: "Men continue to develop their general and nested skills until their 50s."

4.8. You have the sentence “The development of general skills is perhaps 298 instrumental to accruing absorptive capacity [61], enhancing further skill accumulation in 299 later periods.” But absorptive capacity (at least in cohen and levintal) is an organizational construct. Please remove this sentence.

Thank you for pointing out our broad interpretation of the term. Here, we would like to provide our reasoning, but we also recognize the issue of unorthodox use of the term. Therefore, we removed any problematic use of the term in the current manuscript. Nevertheless, we would like to share with you our rationale below to move the field forward.

First, we used the concept of absorptive capacity in the Abstract, Wage Premiums subsection of the Results, and the Discussion in our previous manuscript. This is based on our perceived resemblance between the contingent development of nested niche skills on the development of the basic general skills in our framework and the Cohen & Levinthal’s conceptualization of absorptive capacity, in which “basic skills” enable the assimilation and application of more applied type, and *that* certain prior expertise facilitate learning and performing related and subsequent skills:

“...At the most elemental level, this prior knowledge includes basic skills or even a shared language but may also include knowledge of the most recent scientific or technological developments in a given field. Thus, prior related knowledge confers an ability to recognize the value of new information, assimilate it, and apply it to commercial ends. These abilities collectively constitute what we call a firm’s “absorptive capacity.” [Cohen & Levinthal, 1990, p. 128]

“... prior knowledge facilitates the learning of new related knowledge...”. [Cohen & Levinthal, 1990, p. 130]

In our interpretation, Cohen & Levinthal, recognize that the expertise of an organization stems from the collective knowledge of its members, hinting at the emergence of organizational absorptive capacity from the dynamics of individuals’ skills, described by our framework. In fact, a potential implication of the nested

structure of the skill hierarchy is the formation of a "shared language" among workers possessing skills along similar vertical pathways in our hierarchy, who also share common prerequisites. It stands to reason that individuals following similar skill progression paths, such as a chemist and a biochemist, would likely have a greater degree of shared understanding compared to, for instance, a chemist and a janitor. Additionally, to us, organizational theory has borrowed capabilities from the skill concept often, and we are just doing the reverse. Also, absorptive capacity has been used for countries as well. So why not go down a level of aggregation? Nevertheless, we concur with the reviewer's perspective and acknowledge that using such terminology can be speculative, and too broad use, particularly in the abstract. Consequently, we have removed the term from our revised manuscript. Thank you for the guidance.

Cohen, W. M., & Levinthal, D. A. (1990). Absorptive capacity: A new perspective on learning and innovation. *Administrative science quarterly*, 35(1), 128-152.

4.9. To summarize, I think this paper has potential but it needs a better framing, clearer conceptualization of specialization, and a focus on the age and race dimensions, taking out geography.

Good luck.

Thank you once again for your valuable input. This was incredibly helpful.

Response to the reviewers

Thank you for providing insightful comments and the opportunity to improve our paper. We feel incredibly fortunate to have had the thoughtful review team and are impressed by their deep intellectual engagement with our manuscript. This is, especially so, given our unconventional approaches. We found the reviewers' comments in this round to be greatly helpful in improving our work. There were no objections to the reviewers' comments, and we have fully revised the manuscript accordingly.

Once again, we deeply appreciate your time and consideration.

Below, we include our point-by-point response.

Reviewer 1:

I am deeply impressed by the efforts the authors have taken to respond to my comments and improve their manuscript. The paper is now much more legible and its contributions are more clear. However, I have a few minor outstanding comments:

- 1.1. The complexity framing still does not work for me. The contribution of the paper is that it shows that the value of specialized knowledge is contingent on the existence of foundational knowledge. However, the first two sentences of the abstract and the first two paragraphs of the main text are about another topic entirely. I suggest these 2 sentences and these 2 paragraphs should be stricken. The intro section is excellent from the third paragraph onward, and is sufficient to begin the article. I would make a similar comment about the second paragraph of the Discussion section. That paragraph distracts from the actual contributions to of the paper to human capital theory (which I think are very substantial!)

Thank you for the incredibly insightful feedback. It took some time to find the appropriate flow, but we now realize that the initial motivations, as well as the opening and discussion sections, were not effective. We have fully rewritten them accordingly. Thank you so much for your patient and constructive suggestions!

- 1.2. Figure 1, I think the figure should be bumped back a page. Also, the headers for Figure 1A are confusing: there are specific skills, general skills, and "intermediate" skills, but it is unclear what intermediate refers to. How about High Specificity, Moderate Specificity, and Low Specificity? That would allow you to strike the term "General" from the paper, reducing the number of concepts your readers have to remember.

Thank you for the suggestion. We have updated the figures' labels to "high specificity" and "low specificity" to clarify alongside "general" and "specific." Initially, we considered using "specific," "moderate specificity," and "low specificity," but we realized that aligning with the

tradition of labor economics (Becker, 1962) would better convey the implications of how skills are required by occupations, our categorization of skills, and the hierarchical nature of human capital. The new specificity arrow should make it easier for readers to understand the terms "general" and "specific" skills. Additionally, we have fully rewritten the text adding more rationale in the main text to explain how they work with the skill generality measure.

Becker, G. S. (1962). Investment in Human Capital: A Theoretical Analysis. *Journal of Political Economy*, 70(5), 9–49.

- 1.3. Page 8, the term “foundational” in the sentence “Skills with a high nestedness contribution (*cs*) are foundational to ...” is confusing, because foundational/bedrock knowledge is used to refer to the general skills at the bottom of these skill pyramids, but now it is being used to refer to the skills at the top of the pyramids. Also, in the discussion, some terms (“skills like Programming exhibit a positive impact on nestedness”, imply causation, but you have no way to determine whether nested skills emerged on their own (and those drove the increase in nestedness), or were anticipated by the existence of the lower-level more fundamental skills.

Thank you for your excellent feedback. We agree that the wording can be misleading due to the lack of clarity regarding "impacts" and the "pyramid analogy" as well as foundational and nestedness contributions. We have revised the manuscript to better define nestedness and avoid unclear terms and inconsistent analogy. Additionally, we have clarified that by a skill's "impact on nestedness," we refer to its conformity with the nestedness pattern of interaction, which is also commonly referred to as "nestedness contribution", which is quantified by comparison against simulations.

- 1.4. I like the new Figure 3, but it seemed a bit out of order and that it broke down the nicely-flowing organization of the manuscript. I wonder if it can be moved around or if the sections organization can be redesigned.

Thank you for the insightful comments. Along with Reviewer 2's suggestions on skill nestedness and dependencies, this issue was attributed to our lack of a clear definition. Your feedback has prompted us to clarify the concept of nestedness and improve the flow of our manuscript. We have rewritten the previous two sections to better set up the expectations for Section 3 and have revised Section 3 to ensure a smoother transition. Thank you again for your valuable input.

- 1.5. With regard to the additional analyses you performed in response to my earlier comments, the results on all of these are very interesting, especially with regard to family dynamics (Figure S49 and Table S7) and race/ethnicity (Figure S18). Thank you for engaging so thoroughly with my comments!

On that note, we would like to express our appreciation for your thoughtful suggestions. They were incredibly instrumental in deepening our understanding of our research and strengthening the logic throughout our analysis. We feel lucky to have you as a reviewer!

Reviewer 2:

2.1. Overall, I am happy about the changes in the new version. My concerns about presentation have largely been addressed. The new text is clearer, and better explains the logic behind the various analyses. The Methods and SI gained helpful text to understand the datasets and navigate what is a large SI.

On the analysis of un-nested skills / focus on nestedness, the authors have also responded in a couple ways to concerns I raised. They argue that the labels "nested" and "un-nested", while a bit of a simplification, are useful expositionally, and derive from a clear quantitative criterion in the paper. In addition, they acknowledge that "nested" and "un-nested" skills closely correlate with cognitive and physical skills, which for me raised the question whether we learn something new from statements like (for example) un-nested skills are associated with lower wages. The authors note that "cognitive" and "physical" skills are arbitrary categories applied ex post. In contrast, the network quantities they observe are completely derived from the skill dependency network. I agree on all the above points.

In general, I remained convinced that the paper's results are fundamentally sound and represent an important contribution.

Thank you for the encouragement and constructive suggestions. In addition, thank you for the detailed comments! Your feedback, along with the additional comments below, has been truly developmental, helping to deepen our understanding of our research and strengthen the logic throughout our analysis. We feel fortunate to have you as a reviewer!

2.2. I have a small technical concern from text that appears to be new. In addition, I still have concerns about the discussion of nestedness. Many of these are about how this concept is discussed in the paper versus ecology and economic complexity, fields this paper is connecting itself with. More generally, I am also still trying to understand why nestedness and "ecosystems" are used in framing this paper at all.

Thank you so much for highlighting this and encouraging us to take a deeper look. After revisiting our text and reflecting on your comments 2.2 and 2.4, we realize that connecting our work with the ecosystem is too far-fetched, even though it was our initial intellectual motivation. While this analogy has shaped our research up to this point, we agree that it's time to move on. As a result, we have removed references to the ecosystem from our title, abstract, and main text while expanding the logics and structural implication of nestedness in response to your comment 2.4 below.

2.3. Independence of skills A and B:

At the top of page 6, the authors discuss what it means for the presence of two different skills in an occupation to be independent events. The text says if skill A is independent of skill B, we should see

$$p(\text{skill_A} \mid \text{skill_B}) = p(\text{skill_A}) p(\text{skill_B}).$$

But this seems to confuse conditional and joint probabilities. Presumably, the argument needed here is

$$p(\text{skill_A} \mid \text{skill_B}) = p(\text{skill_A}).$$

If I am correct then this worries me (slightly) how this assumption might be entering the analysis.

Thank you for bringing this issue to our attention. It stems from a combination of a typo and an oversight during the process of simplifying the algorithm (Jo et al., 2020) to be more intuitively understood.

First of all, the reviewer is correct: $p(A|B)=p(A)$, not $p(A)p(B)$. This is our overlook. When two events are independent, it should have been $p(A|B)=p(A \cap B)/p(B)=p(A)p(B)/p(B)$, and thus $p(A)$. The original intention was to explain that our method filters out co-occurrences that happen even when the events are independent, with $p(A)p(B)$, simply because A and B are widely distributed, that is, high $p(A)$ and $p(B)$. To get statistically significant asymmetric relations, therefore, we need to remove any statistical noise co-appearances of skills that can be explained by pure chance, using the threshold z_{th} . We now elaborate and articulate this method and logic in the main text.

2.4. "Nestedness" and connections to ecology, economic complexity

I greatly appreciate the effort the authors put into responding on these issues. However, in spite of significant changes, I am still asking myself, "Why frame these results in terms of nestedness and ecosystems at all? Aren't these results just about 'dependencies' between skills and how they matter for career and economic outcomes?"

It's evident from lived experience that some skills serve as prerequisite foundation for others. Yet (as the authors rightly hone in on) there is nowhere one can look for a comprehensive quantitative description of these dependencies. The paper offers this, making many interesting observations about both the network structure itself, and crucial correlations with economic variables and career trajectories.

Now, somehow, these interesting observations get entangled with an ecosystem analogy. This begins from the title, and is repeated throughout, explicitly, and is also reflected in the focus on the phenomenon of "nestedness" (which also occurs in ecology). The analogy originally struck me as a connection that could readily be dropped. Yet the new draft and the author responses seem to elevate its importance. In addition, there are also connections to the field of economic complexity that somehow a part of all this.

I think the connections to these fields are confusing. I first review what I think I know about other literatures (on which the authors should freely correct me). In ecology,

nestedness is a property of a set of ecosystems. It can be computed from a matrix that has the dimensions

species x places [ecology].

"Nestedness" here is the observation that less diverse ecosystems are subsets of ("nested within") more diverse ones. So, large island "A" has all the (generalist) species of small island "B", plus some extra specialist species. In economic complexity, the corresponding matrices have dimensions of economic activities x places. The current paper deals with "occupations", so let's take these as the activities:

occupations x places [economics]

"Nestedness" also comes up in economic complexity, where it has a nearly identical meaning, as the observation that less-diverse places not only have fewer occupations (by definition), but the occupations they have are subsets of those of more diverse places.

In the current paper, the relevant matrix has these dimensions:

skills x occupations [this paper].

Here, nestedness is posed as a property of a skill. So, a more nested skill here is one that is deeper along the branches that stem from root, foundational skills.

In all these contexts, there is discussion about possible mechanisms underlying the nestedness. It sounds like ecologists have a mechanism that involves mutualistic interactions of generalist species with specialists. In economic complexity, it's often thought that nestedness results from economic "capabilities". I.e. some activities need more capabilities and some places have more capabilities, and nestedness of economies occurs (it's thought) because having more capabilities means an economy can do any of the things less complex economies can do plus more. In this paper, the proposed mechanism is not quite fleshed out (nor needs to be as far as I'm concerned) but the authors have in mind that making complex things means combining many skills, but cognitive limits constrain how many skills one human can learn, requiring us to develop careers in which we specialized in particular skill sets.

I find the discussion about ecosystems, economic complexity, and nestedness confusing for a few reasons:

First, what exactly is the analogy to ecosystems? A "labor ecosystem" at first blush sounds like occupations are the "species". But the paper instead takes skills the "species"; it makes this correspondence:

skill <-> species

occupation <-> place

In the usual analogy between economic complexity and ecology, we have occupations as the species:

occupation <-> species

place <-> place

The usual analogy is natural and "nestedness" has essentially the same meaning in both economic complexity as in ecology. (I.e. not only are some places less occupationally diverse, but the occupations they do have are subsets of the occupations of more diverse economies.) The analogy of the paper feels odd or forced, and it seemingly leads to the issues below.

"Nestedness" clashes with its meaning elsewhere. If skills are the "species", then given how nestedness is discussed in ecology and economic complexity, nestedness should be a property of an occupation, not a skill. (Or really, a collection of occupations.) I.e. skills should not be the thing that's more or less nested but occupations, with some occupations requiring narrow skill sets, which are subsets of the skills of occupations requiring broad skill sets. (To some extent this issue gets confusing because many metrics of the nestedness of a matrix give the same score whether applied to a matrix or its transpose.)

Discussion of mechanisms: The paper says nestedness of skills is akin to ecological interactions where specialist species interact mutualistically with generalist species. This comment gets mentioned several times, but it's unclear why the paper is emphasizing this. Is this just reiterating something ecologists think about the cause of nestedness in ecology? If so, why do we care? Is the implication that an analogous mechanism could underly skills and occupations? Is this fleshing out the analogy itself? Is it just a random tidbit?

At the same, economic complexity has another explanation of nestedness that readers might naturally try to apply to this paper. (The capabilities story above.) But again, it is unclear what mapping the authors have in mind here, or what significance it would have. Also, note that, in the economic complexity discussion of nestedness (instead of ecology), one deals with the extra concept of "capabilities". I.e. we don't just have the bipartite network of places and the "species" that express the makeup of these places, but a tripartite network where the expression is mediated by capabilities. Nestedness is something that has been observed in the expression of activities by different places (i.e. in the sectors or products), but conceptually, "skills" correspond much better to "capabilities". And connecting "skills" to "capabilities" (instead of to economic sectors) makes all the more sense here given the authors' point that diverse skills need to be combined to make complex goods, while each human is limited in how many skills they can learn, leading to career specialization.

Making analogies is in principle fine, but the way it is handled currently will cause confusion. I am not quite sure what to suggest, because, while I don't see how these connections as necessary (given the importance and strength of the findings), the authors may regard these connections as important. I ultimately find myself back to my

original question- should these results just be discussed as an investigation of inter-dependencies among skills? "Nestedness" and the ecosystem analogy, while interesting topics, don't seem like necessary connections to make the findings important.

Thank you for your insightful feedback and detailed explanations. After reflecting on your comment 2.2 and revisiting our text, we find that our attempt to connect our work with nestedness in an ecological context was indeed a stretch, despite it being our initial intellectual inspiration. We have now chosen to use the term in a more conventional sense, focusing on nested distribution and its implication for the dependency direction, as you suggested. This shift makes our definition of nestedness more neutral, allowing us to focus on the structure we seek to measure, streamline our argument, and make the manuscript more coherent. Building on your feedback, we have revised the text as follows:

1. We first changed the title from "Nested Skills in Labor Ecosystems: A hidden dimension of Human capital" to "Skill Dependencies Uncover Nested Human Capital", emphasising dependencies while dropping ecosystems analogies.
2. More importantly, we removed our statements such as "nestedness of skills is akin to ecological interactions..." Meanwhile we elaborate and articulate more on 'dependency structure.' Thank you so much for suggestions.
3. We define nestedness as a subset of distributions, and expand how this measure indicates dependency chains. In addition, we elaborate your point that skills are not always nested because occupations, unlike islands or countries, are limited in what they can embody, is the reason why not every skill contributes to the nested structure. We added that the skill-occupation matrix is not perfectly upper-triangular for this reason, but there are a few small fuzzy blocks of upper-triangles. We categorize skills that align with this nested structure (portion of upper triangle filling in the skill-occupation matrix) and those that do not. We therefore identify those skills that belong to these small fuzzy upper-triangle matrices by calculating the nestedness score c_s for each skill. Skills with positive c_s are part of these upper-triangle patches, while skills with negative c_s are misaligned. We find that skills with positive c_s have higher wages and require more education, which makes sense to us.

Additionally, while it may not be directly relevant, we wanted to share our perspective that nestedness in ecology can appear in two ways. First, less diverse islands are nested within more diverse ones based on species composition; species present on less diverse islands are typically a subset of those on more diverse islands. Rare or specialist species are only found on the more diverse islands, so while less diverse islands share species with more diverse ones, the reverse is not true, resulting in asymmetric conditional probabilities. Second, rare species are nested within ubiquitous species regarding habitat distribution; the islands where rare species exist are a subset of those where more common species are found, but the opposite does not hold. This dual projection doesn't seem work very well with skill-occupation matrix as you describe for the constraints that ecology didn't have. It'd be interesting to study this constraint a little bit more in the future.

2.6. Minor comments

- Page 8, "Figure 2 (d-f)": No panel f in figure 2.

Thank you for pointing out the mistake. We have corrected the error and updated the references to Figure 2 (c-e) to point to the corresponding parts of the figure.

- Page 9: "translated"  "translate"

Thank you so much. This is fixed.

- Pages 18-19: "Future research could benefit from surveys targeting employees... etc." Repeated text with next paragraph.

Thank you for your thorough review. You were right about the repetition of phrases. We have removed the last two paragraphs and tightened the discussion section. In addition, we make the discussion section more concise.

- Page 21: Reference to Fig. 2 (b-f) but no panel f.

Thank you for pointing out the mistake. We have corrected the error.

Finally, once again, we sincerely appreciate your insights and guidance with the detailed and constructive feedback. They have greatly helped us articulate our study more clearly.

Reviewer 4:

4.1. The paper has advanced quite a bit from my review, and I think it is well on its way to publication in Nature Human Behavior, congratulations to the authors on these changes.

I have two important remaining notes:

Thank you for the compliments and encouragement! Your feedback was challenging to be honest but immensely helpful in deepening our understanding and strengthening the logic throughout our analysis. We feel fortunate to have had you as a reviewer!

4.2. The biggest one, which I don't know exactly how to say, is that while the authors have answered all my comments directly, the paper is throughout still hard to follow.

a. This issue stems from something in the writing that I cannot quite state: it is too wordy, too many long sentences, and too flowery, rather than the clear and concise language that make for good academic writing. Each sentence is long, complex, and jargony, rather than short and to the point.

b. The most egregious example of this are the extremely long responses to my review. I found the document very hard to follow.

c. And, while the reviewer document is less important, the issues also show up in the paper. For example, the paper starts with the following sentences: “Complexity and specialization are foundational to the narrative of economic growth and innovation [3{6]. As society advances, creating and maintaining sophisticated goods, services, and infrastructure, these socio-economic complexities have surpassed what individuals can embody and manage on their own [7, 8]. It is no longer feasible for individuals to master universal expertise across all areas. For economies, this means developing deep divisions of labor and knowledge that first distribute knowledge across people and then coordinate this distributed knowledge in teams, firms, and value chains [9{12].”

For an introductory paragraph, is this really jargon heavy and hard to read.

A simpler version of basically the same could be:

Complexity and specialization are central to economic growth and innovation [3{6]. As society advances, socio-economic complexities have surpassed what individuals can embody and manage on their own [7, 8]. Because individuals cannot master expertise across all areas, developing deep divisions of labor and knowledge that both distribute and coordinate this knowledge is important for all economies [9{12].”

I am not a good writer, but I do think the above is better. To me, the first order of business is getting this paper to be much simpler, potentially working with a copy editor.

A big area where this could improve is on the explanation of nestedness. I finally learned clearly what it is from a simple Wikipedia page. I think just slightly more clarity on writing this section will make a big difference.

Great work overall. I hope this helps.

Thank you for the thoughtful suggestions. We agree that the wording was indeed unnecessarily complex and filled with jargon, and also a bit logically incoherent (other reviews pointed out them). We have removed problematic sentences like 4.2-(c) and, most importantly, hired a professional editor to enhance the readability of our paper. We have also clarified the definition of nestedness and tailored it to skill-occupations, following Reviewer 2’s suggestion. By the way, it’s really interesting how skills ($c_s > 0$) always better rewarded than skills ($c_s < 0$). Thank you so much!